# CPathAgent: An Agent-based Foundation Model for Interpretable High-Resolution Pathology Image Analysis Mimicking Pathologists' Diagnostic Logic

**Yuxuan Sun**[1,2,†], **Yixuan Si**[2,†], **Chenglu Zhu**[2], **Kai Zhang**[3],
**Zhongyi Shui**[1,2], **Bowen Ding**[1,2], **Tao Lin**[2,‡], **Lin Yang**[2,‡]
[1]College of Computer Science and Technology, Zhejiang University, China
[2]Research Center for Industries of the Future and School of Engineering, Westlake University, China
[3]Department of Computer Science and Engineering, The Ohio State University, USA

## Abstract

Recent advances in computational pathology have led to the emergence of numerous foundation models. These models typically rely on general-purpose encoders with multi-instance learning for whole slide image (WSI) classification or apply multimodal approaches to generate reports directly from images. However, these models cannot emulate the diagnostic approach of pathologists, who systematically examine slides at low magnification to obtain an overview before progressively zooming in on suspicious regions to formulate comprehensive diagnoses. Instead, existing models directly output final diagnoses without revealing the underlying reasoning process. To address this gap, we introduce CPathAgent, an innovative agent-based approach that mimics pathologists' diagnostic workflow by autonomously navigating across WSI through zoom-in/out and move operations based on observed visual features, thereby generating substantially more transparent and interpretable diagnostic summaries. To achieve this, we develop a multi-stage training strategy that unifies patch-level, region-level, and WSI-level capabilities within a single model, which is essential for replicating how pathologists understand and reason across diverse image scales. Additionally, we construct PathMMU-HR², the first expert-validated benchmark for large region analysis. This represents a critical intermediate scale between patches and whole slides, reflecting a key clinical reality where pathologists typically examine several key large regions rather than entire slides at once. Extensive experiments demonstrate that CPathAgent consistently outperforms existing approaches across benchmarks at three different image scales, validating the effectiveness of our agent-based diagnostic approach and highlighting a promising direction for computational pathology.

## 1 Introduction

Pathology serves as the gold standard for diagnosing numerous diseases, particularly cancer. The advent of digital pathology has enabled the digitization and computational analysis of histopathological slides, opening transformative opportunities for AI-assisted diagnostics. Recently, numerous pathology foundation models have emerged to automate and enhance diagnostic workflows.

These foundation models (Figure 1, left) adopt two primary architectural paradigms. The first employs encoder-based models such as DINO[1; 2] or CLIP [3] to extract patch-level representations from

---

[†]Equal contribution.
[‡]Corresponding author.

39th Conference on Neural Information Processing Systems (NeurIPS 2025).

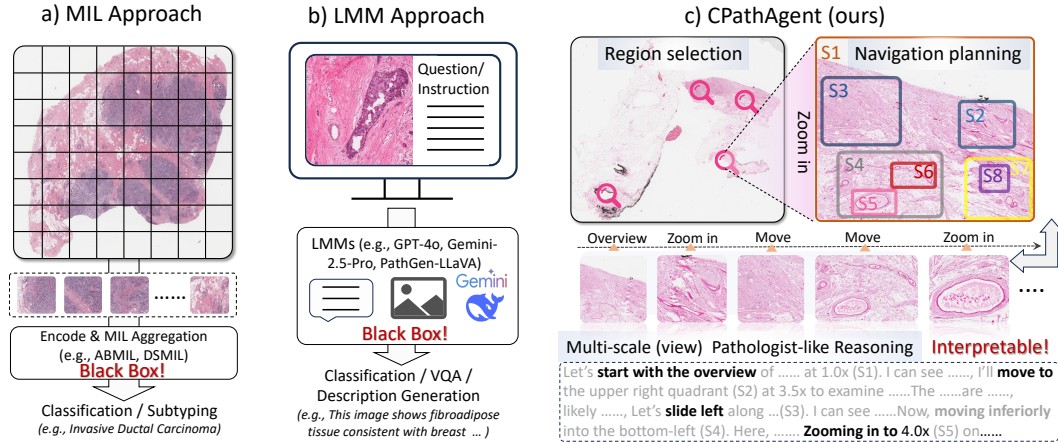

Figure 1: Comparison of the "black box" traditional MIL approach, LMM approach, and our proposed CPathAgent for analyzing pathology images. CPathAgent interpretably mimics pathologists' reasoning by performing operational actions (e.g., zooming in, moving the view) while describing analytical logic.

WSIs, which are then aggregated via Multi-Instance Learning (MIL) [4; 5; 6] for diagnostic predictions. The second utilizes large multimodal models (LMMs) that jointly process visual data and text instructions to generate pathology reports or structured outputs [7; 8; 9; 10]. While these approaches achieve impressive benchmark performance, they share a fundamental mismatch with clinical practice. In real-world diagnostics, pathologists employ a systematic multi-scale examination strategy. They begin by scanning slides at low magnification (2-4×) to assess overall tissue architecture and identify regions of interest, then strategically navigating to atypical regions while progressively increasing magnification (10×, 20×, 40×) to examine fine-grained details. Throughout this process, pathologists continuously integrate observations across scales while conducting step-by-step diagnostic reasoning.

However, current models bypass this systematic sequential process by directly aggregating information across the WSI or processing visual inputs in a single forward pass, leading to two critical limitations. First, **weak supervision:** relying solely on slide-level labels without explicit guidance on diagnostically important regions or when to examine specific magnifications makes them vulnerable to shortcut learning [6], exploiting spurious correlations (staining artifacts, background patterns) rather than genuine pathological features. Second, **lack of interpretability and verifiability:** producing only final predictions without providing natural language descriptions of the intermediate reasoning prevents pathologists from understanding what the AI actually discovered and validating whether its findings are based on clinically relevant features.

To address these limitations, we introduce CPathAgent, a novel agent-based framework inspired by recent advances in AI agent systems that have achieved remarkable success from strategic gameplay [11] and web navigation [12] to task automation [13] and multimodal content generation [14; 15; 16; 17]. The key to their success lies in decomposing complex tasks into sequential reasoning steps with dynamic decision-making. Following this paradigm, CPathAgent bridges the gap between clinical practice and AI-assisted diagnosis by rethinking pathology diagnosis as adaptive visual reasoning. Unlike existing black-box models that treat diagnosis as single-shot classification, CPathAgent emulates pathologists' examination strategy through WSI navigation with dynamic magnification adjustment and multi-view synthesis, conducting explicit step-by-step reasoning through natural language to enable interpretable and verifiable diagnostic assessments.

Specifically, CPathAgent performs a three-stage diagnostic workflow (Figure 1). In the **global screening** stage, CPathAgent analyzes a low-resolution WSI overview to identify suspicious regions that warrant detailed examination. In the **navigation planning** stage, CPathAgent plans an exploration strategy for each selected region, determining a sequence of view coordinates and magnification levels to investigate, mimicking how pathologists mentally map their examination before diving into details. Finally, in the **multi-scale reasoning** stage, the planned navigation path is executed through systematic analysis of fields of view at different magnifications (e.g., zooming in for cellular details, moving laterally across areas, zooming out for context), and synthesizes observations via step-by-step diagnostic reasoning to reach a conclusion. Overall, this work makes the following contributions:

1) **CPathAgent framework:** We propose an agent-based diagnostic system that performs diagnosis through strategic WSI navigation and multi-scale visual reasoning, mirroring pathologists' systematic examination process while providing interpretable step-by-step rationales.

2) **PathMMU-HR² benchmark**: We develop the first visual question answering benchmark specifically designed for high-resolution pathology image regions (16000×16000 pixels), rigorously validated by three professional pathologists to ensure clinical relevance and diagnostic accuracy.

3) **Multi-stage training strategy:** We design a progressive training approach that equips CPathAgent with perceptual and reasoning capabilities across different image scales within a unified framework.

4) **Comprehensive evaluation:** We conduct experiments at multiple scales, from patches and large regions to WSIs, highlighting CPathAgent's strong performance and practical clinical utility.

## 2    Related Work

### 2.1    Pathology Foundation Models

Recent years have seen significant progress in developing foundation models for computational pathology, broadly categorized into encoder-based models and large multimodal models (LMMs).

Encoder-based models focus on extracting rich patch-level representations from WSIs. Early efforts in this area relied on ImageNet-pretrained CNNs [18; 19] as feature extractors, combined with multi-instance learning (MIL) [4; 5; 6] to aggregate these features for slide-level prediction. More recently, the field has shifted toward pathology-specific foundation models that adopt self-supervised learning paradigms. Architectures such as DINO [1; 2] and CLIP [3], when trained on large-scale pathology datasets, have demonstrated superior feature extraction capabilities [20; 8; 21; 22; 23]. These improved representations, integrated via MIL or incorporated into pretrained WSI transformers [20; 9], have significantly advanced the WSI-level classification performance. LMMs for pathology have emerged in parallel, enabling joint processing of pathology images and text instructions to generate textual diagnostic outputs. These models integrate visual and linguistic information through various designs. Patch-level LMMs [7; 24; 8; 25; 26] adapt general vision-language frameworks like LLaVA [27] for patch-level pathology image interpretation, while WSI-level models [9; 28; 29] aim to process entire WSIs for generating reports or answering diagnostic questions. Recent unified frameworks [10] attempt to handle both patches and WSIs within a single model architecture.

Despite these advances, current models typically process pathology inputs in a static manner, contrasting with expert pathologists who dynamically navigate regions of interest, examine tissues at multiple magnifications, and integrate contextual information to formulate diagnoses.

### 2.2    Agent-based Models

Agent-based models represent an emerging paradigm in AI research, characterized by systems that can perceive their environment, make decisions, and take actions to achieve specific goals [30]. Recent advances in large language models (LLMs) have catalyzed significant progress in this domain, enabling more sophisticated reasoning and planning capabilities. Notable examples include WebGPT [12] and WebShop [31], which navigate web environments to complete information-seeking and shopping tasks; ReAct [32], which interleaves reasoning and action steps for improved task performance; AutoGPT [33] and AutoGen [34], which decompose complex tasks into manageable subtasks with minimal human supervision; and task-specific agents for code development [14], creative content generation [35], scientific research [36], and clinical decision-making and discovery that simulate physician workflows in hospital environments [37; 38; 39; 40]. These agent models typically incorporate several key components [15]: perception modules to process environmental inputs, reasoning modules to interpret observations and plan actions, and execution modules to implement planned actions. This architecture enables agents to adapt to dynamic environments and accomplish complex, multi-step tasks through iterative observation-reasoning-action cycles.

Despite success in various domains, agent-based approaches remain largely unexplored in computational pathology. PathFinder [41] represents a successful early attempt at introducing agent-based systems for WSI analysis. The system segments WSIs into fixed 512×512 patches and employs a multi-agent framework that iteratively generates importance maps to select diagnostically relevant regions, produces natural language descriptions of these patches, and then regenerates subsequent

importance maps conditioned on previous descriptions. However, PathFinder primarily operates through region selection at a fixed magnification level within predefined 512×512 patches, lacking the dynamic multi-scale navigation capability that characterizes pathologist workflows.

To address this gap, CPathAgent takes a step further by incorporating multi-scale navigation capabilities that mimic pathologist behavior. It autonomously decides where to examine, when to zoom in or out, and how to navigate across WSIs, closely mimicking the iterative diagnostic workflow of expert pathologists. Notably, CPathAgent provides verbalization of its diagnostic reasoning, effectively simulating the internal thought processes that pathologists experience during slide examination (specific examples can be found in Appendix Figures A12 to A16). This approach enables enhanced interpretability through traceable reasoning paths and improved diagnostic performance.

### 2.3 Pathology Benchmarks and Datasets

Benchmark datasets provide standardized evaluation frameworks crucial for advancing computational pathology. Traditional WSI benchmarks like Camelyon16 [42] and TCGA [43] focused primarily on slide-level classification tasks. More recently, datasets like WSICaption[44] and WSI-VQA[45] have expanded the scope to more complex tasks, including WSI description generation and VQA.

In parallel, several patch-level benchmarks have emerged for evaluating multimodal models in pathology. PathVQA [46] introduced the first VQA tasks for pathology images, drawing on textbook image-caption pairs. QuiltVQA [47] expanded this approach by extracting image-caption pairs from pathology lectures on YouTube, resulting in higher-quality VQA pairs due to the educational nature of these materials. PathMMU [48] marks a more recent advance, offering a comprehensive benchmark across diverse pathology sources validated by expert pathologists to ensure clinical relevance.

Despite this progress, existing benchmarks predominantly focus on either localized patch-level tasks or broad WSI analysis, overlooking a critical intermediate scale: the **huge region**. This scale is particularly important because it mirrors real-world diagnostic workflows. In routine practice, pathologists typically begin by scanning the WSIs to identify suspicious areas, then focus their diagnostic attention on these large, yet manageable regions. It is often within these huge regions that key diagnostic decisions are made, as they provide sufficient context for interpretation.

To address this limitation, we introduce PathMMU-HR², a benchmark specifically designed for huge region analysis in pathology. By targeting this intermediate scale between isolated patches and WSIs, PathMMU-HR² better aligns with how pathologists actually examine and interpret tissue. It provides a more controlled, context-rich, and diagnostically meaningful setting for evaluating multimodal models, especially those incorporating agent-based reasoning across different scales. To ensure clinical relevance, we also engaged three board-certified pathologists to validate the dataset, making PathMMU-HR² an expert benchmark for intermediate-scale pathology evaluation.

## 3 Methods

In this section, we detail the construction of CPathAgent-Instruct training data, the PathMMU-HR² benchmark, and the CPathAgent model architecture with its multi-stage training process.

### 3.1 Reasoning Workflow of CPathAgent

To simulate expert-level diagnostic reasoning, we design CPathAgent as an agent-based system capable of dynamic region selection, strategic navigation planning, and multi-scale reasoning. As illustrated in Figure 2, the entire diagnostic process is decomposed into three key stages:

**1) Global Screening:** To address the computational challenge of gigapixel WSIs, CPathAgent employs a coarse-to-fine region selection that mirrors pathologists' systematic screening protocol.

Given a WSI $\mathcal{I} \in \mathbb{R}^{H \times W \times 3}$, we first generate a thumbnail $\mathcal{I}_{\text{thumb}}$ via $32\times$ downsampling and partition it into a grid of $N$ regions with 5% overlap to ensure boundary continuity: $\mathcal{G} = \{g_1, g_2, \ldots, g_N\}$, where each $g_i$ corresponds to a $16000 \times 16000$ pixel region at $40\times$ magnification. CPathAgent $f_\theta$ takes the annotated thumbnail as input and generates structured text outputs: *(1) Region grouping*: CPathAgent clusters regions with similar pathological characteristics into $K$ diagnostic regions $\mathcal{C} = \{C_1, \ldots, C_K\}$, where each cluster $C_k = \{g_{i_1}, \ldots, g_{i_{|C_k|}}\}$ is assigned a descriptive semantic

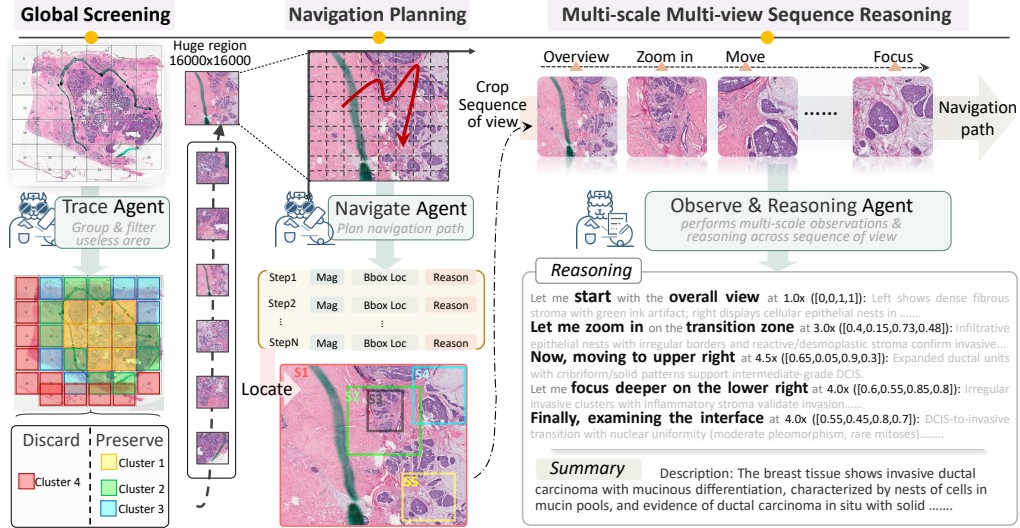

Figure 2: Illustration of CPathAgent framework, which mimics the diagnostic workflow of pathologists via global screening, navigation planning, and multi-scale reasoning across sequential views.

label $\ell_k$ (e.g., "Core Tumor Regions," "Lymph Node Assessment," "Background Adipose and Stromal Tissue"); *(2) Region prioritization*: For each cluster $C_k$, CPathAgent predicts a severity level $s_k \in \{0, 1, \ldots, 5\}$ where higher values indicate greater clinical importance and inspection priority, along with a binary decision $d_k \in \{0, 1\}$ indicating whether $C_k$ requires high-magnification review ($d_k = 1$) or can be skipped ($d_k = 0$). This stage produces a structured output $\mathcal{R} = \{(\ell_k, C_k, s_k, d_k)\}_{k=1}^{K}$ that filters uninformative regions ($s_k = 0$), reducing computational burden for subsequent analysis.

**2) Navigation Planning:** For each preserved huge region $r_i$, the model generates a dynamic navigation plan as a sequence of viewing steps. Each step is specified by a tuple $(x, y, m, o)$, where $(x, y)$ denotes spatial coordinates in normalized space $[0, 1]$, $m$ represents the magnification level relative to the original huge region (where $1.0\times$ corresponds to the full view), and $o$ describes the diagnostic focus for that position (i.e., what features need to be observed). The navigation plan $\mathcal{P}_i = \{(x_1, y_1, m_1, o_1), \ldots, (x_T, y_T, m_T, o_T)\}$ is generated autoregressively, where each step is conditioned on the visual content of the region and WSI source information (e.g., lung, colon, etc.).

**3) Multi-scale Multi-view Sequence Reasoning:** Following the planned navigation path $\mathcal{P}_i$, the model receives the complete sequence of cropped images corresponding to all viewing steps $\{(x_1, y_1, m_1, o_1), \ldots, (x_T, y_T, m_T, o_T)\}$ as input at once. Given this multi-view image sequence along with their associated diagnostic focus descriptions and spatial coordinates, the model performs holistic reasoning across the entire viewing trajectory in a first-person perspective similar to a pathologist's examination process. The reasoning process systematically observes features across multiple scales and positions, establishes and refines diagnostic hypotheses by cross-referencing evidence, and maintains logical continuity throughout the navigation sequence. Finally, the model synthesizes all observations into a coherent pathological report that summarizes the diagnostic findings.

Illustrative examples, including global screening results, navigation path planning, and multi-scale reasoning outputs, are shown in Appendix Figure A12 to Figure A15. The prompts used for CPathAgent through these stages are provided in Appendix Figure A26 to Figure A28.

### 3.2 CPathAgent-Instruct Dataset Construction

In contrast to methods relying on inference-time prompting with closed-source models (e.g., OpenAI o3, Gemini-2.5-Pro), we focus on curating high-quality data to train open-source models for expert-level agent-based pathology analysis. To this end, we construct the CPathAgent-Instruct dataset, comprising corresponding subsets that target the three critical stages detailed in Section 3.1. While we use Gemini-2.5-Pro following the approaches in Section 3.1 to synthesize training data, prompting alone proves insufficient for expert-level performance. We therefore incorporate reference information such as WSI reports as guidance to ensure data quality. The construction process is as follows:

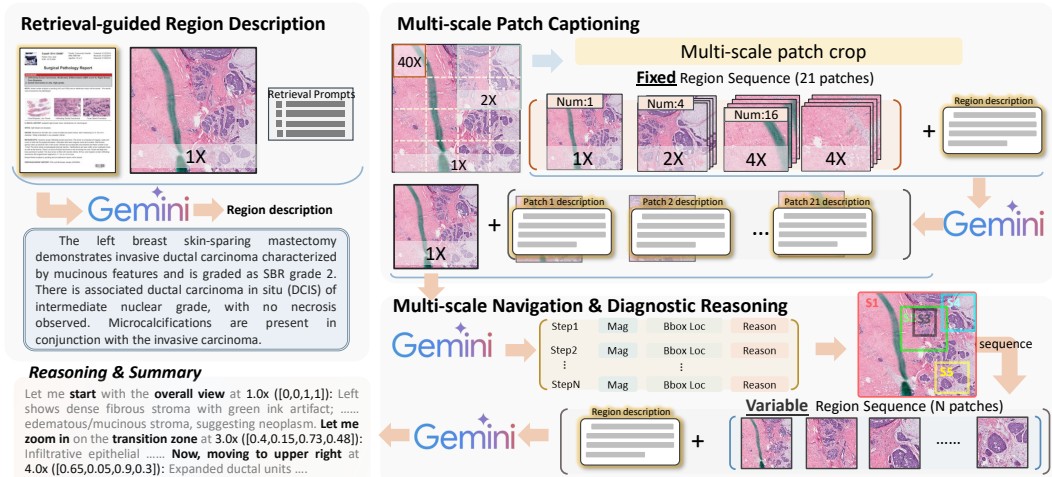

Figure 3: Overview of the generation process for CPathAgent's navigation planning subset and multi-scale multi-view sequence reasoning instruction-tuning data.

**Source Data:** We use WSI reports from HistGen [49] and corresponding WSIs from TCGA, splitting 80% (5,254 WSIs) for training and 20% for testing and PathMMU-HR² generation. We apply patient-level splitting to ensure no data leakage across sets. Details are provided in Appendix A.

**Global Screening Subset:** This dataset follows the same WSI processing as CPathAgent's global screening stage, with one key difference: we leverage Gemini-2.5-Pro, using WSI overview as primary inputs and corresponding WSI reports as additional guidance, to generate high-quality structured outputs (region groupings, priority scores, and examination flags) as training data.

**Navigation Planning Subset:** We implement a reference-guided three-step generation pipeline to synthesize high-quality navigation plans $\mathcal{P}_i = \{(x_1, y_1, m_1, o_1), \ldots, (x_T, y_T, m_T, o_T)\}$ that decide when and where to examine tissue at different magnifications, leveraging expert knowledge from WSI reports as shown in Figure 3: *1) Retrieval-guided Region Description*: For each important region identified during global screening, we extract region-specific descriptions from expert WSI reports by prompting Gemini-2.5-Pro with the WSI report and region image, extracting a region description $D_{\text{region}}$ that captures the features of this region. *2) Multi-scale Patch Captioning*: To provide comprehensive visual references for navigation path generation, we segment patches at three scales: $1\times$ (overview), $2\times$ (four views), and $4\times$ (sixteen views), forming a multi-scale patch set $\{\mathcal{I}^{1\times}, \mathcal{I}^{2\times}, \mathcal{I}^{4\times}\}$. We then prompt Gemini-2.5-Pro with these patches and $D_{\text{region}}$ to produce descriptions $\{d_1, d_2, \ldots, d_{21}\}$ for these patches that guide subsequent navigation path generation. *3) Navigation Path Generation*: We prompt Gemini-2.5-Pro with coordinate-annotated region images (marked with 0.1 relative position intervals), patch descriptions $\{d_1, \ldots, d_{21}\}$, and $D_{\text{region}}$ to generate navigation plans where each step $(x_t, y_t, m_t, o_t)$ can navigate to any position and magnification, enabling flexible navigation strategies that fully mimic pathologist rather than fixed sequential routes.

**Multi-scale Multi-view Sequence Reasoning Subset:** The synthesis of this subset follows the same process as CPathAgent's multi-scale reasoning stage (Section 3.1). In addition to the cropped image sequences extracted along the planned navigation paths, we provide the region description $D_{\text{region}}$ as reference guidance to Gemini-2.5-Pro for generating high-quality step-by-step reasoning chains.

**VQA-oriented Subset:** To enable CPathAgent to handle both general diagnosis and VQA scenarios, this is an additional subset that adapts the navigation planning process to question-driven contexts. We first prompt Gemini-2.5-Pro to generate pathology questions from multi-scale patch descriptions $\{d_1, d_2, \ldots, d_{21}\}$, then generate question-oriented navigation paths by prompting with both the question and region description $D_{\text{region}}$. Finally, we prompt Gemini-2.5-Pro with the cropped image sequences along with the question and region description $D_{\text{region}}$ to produce question-oriented reasoning chains. This trains CPathAgent to dynamically adjust its examination strategy based on question-oriented targeted areas and magnifications to efficiently answer diagnostic questions.

Overall, CPathAgent-Instruct comprises 278K instruction-tuning samples for training CPathAgent's pathologist-like agent capabilities. Detailed data statistics are provided in Appendix A.

### 3.3 PathMMU-HR² Dataset Construction

To evaluate CPathAgent's multi-scale reasoning capabilities on huge regions, we construct PathMMU-HR², a **H**uge **R**egion **H**uge **R**esolution benchmark specifically designed to assess the model's capability for huge region analysis. We sample huge regions from the held-out TCGA test set, ensuring broad coverage across tissue types (e.g., TCGA-BRCA, TCGA-LUAD). Following the construction pipeline of CPathAgent-Instruct VQA-oriented subset, we generate VQA pairs that require synthesizing observations across different scales, necessitating multi-scale reasoning for accurate diagnosis. Three board-certified pathologists independently review and filter generated VQA pairs based on clinical relevance and diagnostic accuracy, necessity of multi-scale integration, and alignment with standard pathology practice. The final PathMMU-HR² comprises 1,668 expert-validated VQA samples, providing a robust benchmark for evaluating huge region analysis capabilities.

### 3.4 CPathAgent Model Architecture and Training

CPathAgent builds upon the LLaVA-OneVision [50], integrating Qwen3-14B [51] as the LLM backbone and CPath-CLIP [10] as the vision encoder, connected via a two-layer MLP. We adopt a three-stage progressive training strategy that systematically develops CPathAgent's capabilities, advancing from foundational multimodal understanding to agent-based diagnostic reasoning.

To be specific, stages 1 and 2 follow the CPath-Omni [10] training protocol: stage 1 aligns vision and language components using CPath-PatchCaption with only the MLP trainable, while stage 2 fine-tunes all parameters on CPath-PathInstruct to develop comprehensive patch-level pathology understanding, including VQA, classification, and image description capabilities.

Notably, stage 3 is the critical phase that transforms CPathAgent into a pathologist-like agent. During this stage, we train on our specialized CPathAgent-Instruct dataset, supplemented with 20% of CPath-Instruct data, with all parameters unfrozen. This phase equips CPathAgent with advanced agent capabilities that emulate pathologists' diagnostic workflows while preserving the strong patch-level analysis skills from earlier stages. This multi-stage approach yields a versatile model that seamlessly integrates fine-grained patch-level analysis with pathologist-like agent-based WSI navigation.

## 4 Experiments

We evaluate CPathAgent across multiple tasks: patch understanding, huge region analysis, and WSI classification to assess its capabilities and benchmark performance against existing approaches.

### 4.1 Patch Understanding Evaluation

Since patch-level understanding serves as the foundation for reasoning over larger-region reasoning, we first evaluate CPathAgent's ability to interpret standard-resolution pathology patches on PathMMU [48], currently the largest and most diverse expert-verified pathology dataset. We compare CPathAgent with the most advanced general-purpose LMMs, including InstructBLIP-FLAN-T5-XXL[52], LLaVA-1.5-13B[53], LLaVA-OneVision[50] series, Qwen2.5-VL[54] series, GPT-4V[55], GPT-4.1[56] series and Gemini-2.5[57] series, along with domain-specific pathology LMMs such as LLaVA-Med[58], Quilt-LLaVA[8], PathGen-LLaVA[26] and CPath-Omni[10].

***Results: CPathAgent significantly outperforms both general-purpose and pathology-specific LMMs.*** As shown in Table 1, it achieves the highest performance across all subsets, with 80.5% on the tiny test set and 78.6% on the full test set, surpassing the strongest general-purpose model (Gemini-2.5-Pro, 68.7% Tiny / 67.5% All) and surpasses the SOTA domain-specific model (CPath-Omni, 72.4% Tiny / 72.2% All). While CPathAgent shares patch-level training data with CPath-Omni, its superior performance stems from the third-stage training on high-quality, multi-scale data, which strengthens its analytical capabilities and lays the foundation for agent-based diagnostic workflow.

***CPathAgent even outperforms the expert-annotated baselines in several subsets,*** suggesting that its general patch-level understanding is already quite promising. This may also reflect limitations in human annotations, as individual experts often specialize in narrow domains and may struggle with unfamiliar cases in a diverse dataset like PathMMU. While CPathAgent benefits from broad training and shows potential as a general-purpose pathology model, patch-level tasks remain relatively simple compared to more complex reasoning across larger spatial contexts.

Table 1: Overall results of models on the PathMMU **test set**. The best-performing LMM in each subset for general and pathology LMMs is **in-bold**, and the second-best performing LMM is underlined.

| | Test Overall | | PubMed | | SocialPath | | EduContent | | Atlas | | PathCLS | |
|---|---|---|---|---|---|---|---|---|---|---|---|---|
| | Tiny (1156) | ALL (9677) | Tiny (281) | ALL (3068) | Tiny (235) | All (1855) | Tiny (255) | All (1938) | Tiny (208) | ALL (1007) | Tiny (177) | ALL (1809) |
| Expert performance | 71.8 | - | 72.9 | - | 71.5 | - | 69.0 | - | 68.3 | - | 78.9 | - |
| **General Large Multimodal Models** | | | | | | | | | | | | |
| InstructBLIP-FLAN-T5-XXL | 34.3 | 33.9 | 39.1 | 37.2 | 33.6 | 34.3 | 34.5 | 36.0 | 38.5 | 39.3 | 22.6 | 22.7 |
| LLaVA-1.5-13B | 38.8 | 37.6 | 44.5 | 41.0 | 40.4 | 40.4 | 34.1 | 39.4 | 47.1 | 44.3 | 24.9 | 23.5 |
| LLaVA-OneVision-Qwen2-7B-OV | 36.9 | 34.4 | 37.7 | 36.4 | 35.7 | 38.4 | 47.1 | 38.3 | 38.9 | 38.4 | 20.3 | 20.4 |
| LLaVA-OneVision-Qwen2-72B-OV | 51.0 | 46.4 | 60.5 | 51.0 | 59.1 | 52.9 | 54.9 | 49.8 | 43.8 | 49.2 | 27.7 | 26.5 |
| Qwen2.5-VL-7B-Instruct | 39.7 | 37.5 | 40.2 | 40.2 | 41.7 | 39.0 | 47.1 | 42.0 | 40.4 | 38.9 | 24.9 | 25.6 |
| Qwen2.5-VL-72B-Instruct | 56.2 | 51.2 | 63.3 | 56.1 | 62.4 | 55.5 | 65.5 | 58.3 | 54.3 | 55.2 | 26.0 | 28.7 |
| GPT-4V-1106 | 53.9 | 49.8 | 59.4 | 53.5 | 58.7 | 53.9 | 60.4 | 53.6 | 48.1 | 52.8 | 36.2 | 33.8 |
| GPT-4.1-mini-2025-04-14 | 60.7 | 59.9 | 66.9 | 63.4 | 62.1 | 61.8 | 65.1 | 61.1 | 60.1 | 62.8 | 43.5 | 48.8 |
| GPT-4.1-2025-04-14 | 67.7 | 64.4 | 73.0 | 66.9 | 70.6 | 65.5 | 69.4 | 64.9 | 64.9 | 66.8 | 56.5 | 57.2 |
| Gemini-2.5-Flash-Preview-04-17 | 68.0 | 65.2 | 74.9 | 69.2 | 71.1 | 64.9 | 68.1 | 65.3 | 67.5 | 69.3 | 53.7 | 56.3 |
| Gemini-2.5-Pro-Preview-03-25 | 68.7 | 67.5 | 73.8 | 71.5 | 72.1 | 68.1 | 70.7 | 68.6 | 68.3 | 70.6 | 53.7 | 57.5 |
| **Pathology-specific Large Multimodal Models** | | | | | | | | | | | | |
| LLaVA-Med | 25.3 | 26.2 | 28.5 | 27.7 | 28.9 | 27.3 | 22.7 | 27.2 | 22.6 | 30.7 | 22.6 | 20.3 |
| Quilt-LLaVA | 45.6 | 41.5 | 47.3 | 42.6 | 46.4 | 46.6 | 51.8 | 45.3 | 46.2 | 42.7 | 32.2 | 29.2 |
| PathGen-LLaVA | 60.1 | 58.4 | 60.1 | 60.1 | 60.9 | 58.8 | 60.8 | 60.7 | 63.5 | 64.9 | 54.2 | 48.9 |
| CPath-Omni | 72.4 | 72.2 | 74.0 | 69.9 | 76.6 | 71.8 | 69.8 | 70.6 | 65.9 | 70.6 | 75.7 | 79.0 |
| CPathAgent (Ours) | **80.5** | **78.6** | **80.8** | **78.4** | **83.4** | **77.9** | **83.5** | **79.6** | **75.4** | **77.6** | **78.0** | **79.2** |

## 4.2 Huge Region Understanding Evaluation

We evaluate CPathAgent's ability to autonomously navigate and reason over huge, high-resolution regions using PathMMU-HR², which includes 1,668 expert-validated VQA pairs requiring multi-scale analysis. We benchmark the same models as in Section 4.1. Additionally, we apply CPathAgent's prompt (from global screening to multi-scale reasoning, detailed in Section 3.1) to stronger agent-capable models like Gemini-2.5-Pro and GPT-4.1-mini, enabling them to mimic pathologists' workflows as CPathAgent does and assess whether this approach yields performance gains.

***Results: CPathAgent demonstrates superior performance on large, high-resolution regions across diverse cancer types.*** As shown in Table 2, CPathAgent achieves 88.6% on the PathMMU-HR², substantially outperforming both general-purpose models (Gemini-2.5-Pro by 15.4%) and pathology-specific models (CPath-Omni by 16.9%). Despite operating at lower single-view resolution (1008×1008) compared to closed-source models like Gemini-2.5-Pro (3072×3072), CPathAgent compensates through emulating pathologists' diagnostic workflow through strategic navigation and multi-view reasoning over huge regions. Notably, while CPathAgent and CPathOmni share patch-level training data from stages 1-2 (Section 3.4), they differ significantly in their stage 3 training, where CPathAgent incorporates agent-based navigation and reasoning for WSI and large region analysis. The substantial performance gain validates the effectiveness of our CPathAgent-Instruct dataset and agent training approach. Representative examples are provided in Appendix E.

***Agent-based approaches consistently enhance model performance for pathology analysis.*** As shown in Table 2, incorporating agent-based approach improves performance across models. Specifically, with CPathAgent's prompting strategy, Gemini-2.5-Pro improves from 73.2% to 76.4% and GPT-4.1-mini from 60.1% to 62.3%. The enhancement is particularly pronounced in complex cancer types like KICH, where Gemini-2.5-Pro shows a dramatic improvement from 56.5% to 69.3% (+12.8%). This confirms that dynamically navigating and reasoning over huge regions that are similar to expert pathologists is crucial for accurate diagnosis, especially for cases requiring integration of findings across multiple scales and regions. These results suggest that empowering LMMs with expert-like navigation and reasoning strategies is a promising direction for advancing AI in pathology tasks.

## 4.3 WSI Classification

We evaluate CPathAgent's performance on WSI classification tasks across six diverse datasets spanning multiple cancer types and diagnostic tasks: TCGA-BRCA (breast cancer subtyping), TCGA-NSCLC (lung cancer subtyping), TCGA-RCC (renal cell carcinoma subtyping), TCGA-ESCA (esophageal cancer subtyping), TCGA-BLCA (bladder cancer subtyping), and TCGA-THCA (thyroid cancer subtyping). We benchmark CPathAgent against traditional MIL approaches, including ABMIL [4] and DSMIL [5], as well as both general-purpose and pathology-specific LMMs.

Table 2: Performance comparison of huge region classification across multiple cancer types using two approaches: General LMMs, and Agent-based Methods. The best-performing model for each cancer type is **in-bold**, and the second-best performing model is underlined.

| | BRCA (368) | LUAD (192) | LUSC (230) | KIRP (153) | KIRC (151) | KICH (62) | ESCA (70) | THCA (249) | BLCA (141) | TGCT (52) | Overall (1668) |
|---|---|---|---|---|---|---|---|---|---|---|---|
| **General LMM Approach** | | | | | | | | | | | |
| LLaVA-OneVision-Qwen2-7B-OV | 42.9 | 43.2 | 33.5 | 28.1 | 37.1 | 43.5 | 40.0 | 41.8 | 39.7 | 30.8 | 38.8 |
| LLaVA-OneVision-Qwen2-72B-OV | 40.8 | 52.1 | 40.0 | 39.2 | 36.4 | 33.9 | 35.7 | 45.8 | 57.4 | 36.5 | 43.0 |
| Qwen2.5-VL-7B-Instruct | 47.3 | 57.3 | 35.2 | 46.4 | 47.7 | 25.8 | 51.4 | 49.8 | 56.0 | 38.5 | 46.9 |
| Qwen2.5-VL-72B-Instruct | 48.4 | 61.5 | 44.3 | 52.9 | 56.3 | 27.4 | 52.9 | 48.6 | 62.4 | 51.9 | 51.2 |
| GPT-4.1-mini-2025-04-14 | 56.0 | 64.6 | 60.4 | 62.1 | 61.6 | 35.5 | 70.0 | 59.4 | 68.1 | 59.6 | 60.1 |
| GPT-4.1-2025-04-14 | 57.9 | 63.5 | 62.2 | 67.3 | 69.5 | 54.8 | 65.7 | 66.3 | 67.4 | 71.2 | 63.7 |
| Gemini-2.5-Flash-Preview-04-17 | 66.6 | 74.0 | 76.1 | 65.4 | 76.8 | 40.3 | 70.0 | 72.7 | 76.6 | 63.5 | 70.4 |
| Gemini-2.5-Pro-Preview-03-25 | 64.4 | 70.8 | 78.3 | 74.5 | 80.8 | 56.5 | 77.1 | 76.7 | 80.1 | 75.0 | 73.2 |
| Quilt-LLaVA | 29.3 | 45.8 | 44.8 | 38.6 | 38.4 | 30.6 | 35.7 | 39.8 | 47.5 | 40.4 | 38.8 |
| PathGen-LLaVA | 62.0 | 77.6 | 67.4 | 58.2 | 74.2 | 48.4 | 67.1 | **84.4** | 48.1 | 67.2 | 67.2 |
| CPath-Omni | 72.6 | 77.6 | 71.3 | 64.1 | 67.5 | 59.7 | 74.3 | 71.5 | 76.6 | 78.8 | 71.7 |
| **Agent-based Approach** | | | | | | | | | | | |
| GPT-4.1-mini-2025-04-14 | 56.0 | 69.3 | 63.9 | 62.7 | 64.9 | 41.9 | 71.4 | 59.8 | 68.1 | 73.1 | 62.3 |
| Gemini-2.5-Pro-Preview-03-25 | 68.8 | 68.8 | 80.0 | 77.8 | 83.4 | 69.3 | 82.8 | 78.7 | 85.1 | 76.9 | 76.4 |
| CPathAgent (Ours) | **87.0** | **88.5** | **87.8** | **87.9** | **92.9** | **78.9** | **90.7** | **89.0** | **90.7** | **93.0** | **88.6** |

Since CPathAgent generates detailed diagnostic descriptions rather than dataset-specific classification labels like MIL methods, we implement a two-stage evaluation process for fair comparison with WSI benchmarks. In the first stage, CPathAgent employs our agent-based approach to identify suspicious regions and generate comprehensive diagnostic summaries of significant findings within each WSI. While these detailed descriptions effectively capture complex pathological characteristics, they must be mapped to the predefined classification schemes used in standard benchmarks. Therefore, in the second stage, we provide these descriptions to Gemini-2.5-Pro for conversion into the specific labels required by each dataset. Consequently, more accurate and detailed diagnostic descriptions from CPathAgent lead to more precise WSI-level classifications. For other LMM-based approaches, following CPath-Omni [10], we directly prompt the models to generate descriptions for all WSI regions. These descriptions are then similarly provided to Gemini-2.5-Pro to map them to the corresponding classification labels, ensuring fair evaluation across all methods.

***Results: CPathAgent achieves competitive performance on WSI classification using a fundamentally different agent-based approach.*** As shown in Table 3, CPathAgent achieves an average accuracy of 82.8% across six different cancer classification tasks, outperforming traditional MIL approaches (ABMIL: 79.9%, DSMIL: 76.8%) and substantially surpassing both general-purpose LMMs like Gemini-2.5-Pro (72.1%) and previous pathology-specific SOTA CPath-Omni (77.4%). CPathAgent shows particularly strong performance on TCGA-BRCA (88.5%) and TCGA-ESCA (97.1%), where it matches or exceeds specialized MIL methods. Although the overall margin over MIL approaches is modest (+2.9% vs. ABMIL), this represents a significant paradigm shift: CPathAgent achieves these results through interpretable, expert-like diagnostic reasoning rather than opaque feature aggregation.

***Upper bound results validate synthetic data quality and reveal substantial room for improvement.*** To assess data quality, we establish a theoretical upper bound by directly using our generated agent training data for WSI classification (Table 3). The near-perfect upper bound accuracy across most datasets (TCGA-ESCA: 100%, TCGA-NSCLC: 97.9%, TCGA-RCC: 96.5%, TCGA-BRCA: 97.0%) confirms that our synthetic navigation trajectories and diagnostic reasoning are clinically accurate and diagnostically complete. While CPathAgent achieves significant advances with 82.8% overall accuracy, the 9% gap to the upper bound (91.7%) indicates considerable room for further improvement, especially on challenging datasets like TCGA-BLCA and TCGA-THCA. As an agent-based exploration for pathology, CPathAgent demonstrates substantial potential for advancement.

## 4.4 Out of Distribution Evaluation

Although our model achieves strong diagnostic performance across multiple TCGA cancer types, we further evaluate its generalization capability on out-of-distribution (OOD) datasets. Specifically, we assess CPathAgent on two tasks: (1) binary WSI classification for distinguishing LUAD and LSCC in the CPTAC-Lung dataset [59; 60], and (2) three-class classification (benign tumors, atypical tumors, malignant tumors) on huge image regions (up to 10K × 10K pixels) in the BRACS dataset [61]. We benchmark our method against representative WSI-level (PRISM [9], CPath-Omni [10], TITAN [28]) and region-level models (GPT-4.1, Gemini-2.5-Pro, CPath-Omni [10]).

Table 3: Overall results of WSI classification tasks. The best-performing model in each subset is **in-bold**, and the second-best performing model is underlined. CPath-Omni* indicates a retrained version using CPathAgent's train/test split to ensure fair comparison, as the original training data differed. Balanced accuracy (%) is reported.

| | TCGA-BRCA | TCGA-NSCLC | TCGA-RCC | TCGA-ESCA | TCGA-BLCA | TCGA-THCA | Avg. |
|---|---|---|---|---|---|---|---|
| **Multi-instance Learning Approach (Traditional)** | | | | | | | |
| ABMIL | 80.5 | **92.8** | **96.4** | 91.2 | 50.1 | **68.4** | 79.9 |
| DSMIL | 84.7 | 87.2 | 88.9 | 89.9 | 52.2 | 57.8 | 76.8 |
| **Large Multimodal Models** | | | | | | | |
| GPT-4.1-mini-2025-04-14 | 52.8 | 65.2 | 52.8 | 85.3 | 57.0 | 54.3 | 59.6 |
| GPT-4.1-2025-04-14 | 61.0 | 65.8 | 53.8 | 85.3 | 61.3 | 58.9 | 64.3 |
| Gemini-2.5-Flash-Preview-04-17 | 56.7 | 84.5 | 57.9 | 93.8 | 61.8 | 55.4 | 68.4 |
| Gemini-2.5-Pro-Preview-03-25 | 72.4 | 89.2 | 69.2 | **97.1** | 59.8 | 44.9 | 72.1 |
| Quilt-LLaVA | 55.3 | 54.0 | 53.3 | 58.6 | 61.9 | 45.8 | 54.8 |
| PathGen-LLaVA | 66.5 | 53.5 | 59.0 | 78.6 | 54.7 | 49.2 | 60.2 |
| CPath-Omni* | 78.6 | 86.7 | 91.3 | 89.9 | **65.5** | 52.6 | 77.4 |
| **Agent-based Approach** | | | | | | | |
| CPathAgent (Ours) | **88.5** | 90.8 | 94.6 | **97.1** | 62.7 | 63.2 | **82.8** |
| Upper Bound | 97.0 | 97.9 | 96.5 | 100.0 | 77.8 | 81.0 | 91.7 |

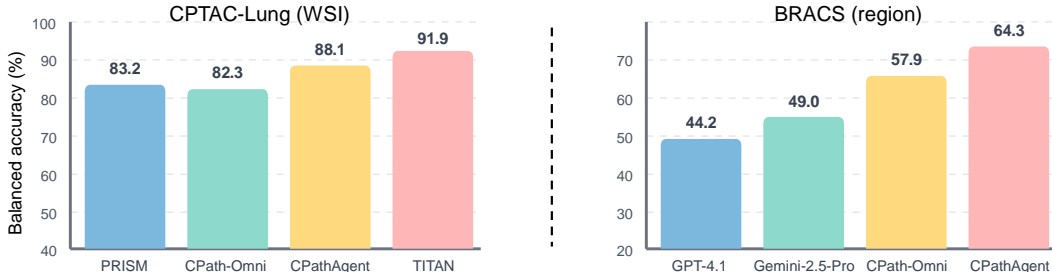

Figure 4: OOD results for WSI (left) and region-level (right) classification tasks. Representative models from WSI and region-level approaches are selected for comparison.

***Results: CPathAgent achieves competitive OOD performance with exceptional data efficiency.*** As shown in Figure 4, despite being trained on significantly fewer slides than baseline models, CPathAgent demonstrates reasonable generalization on both WSI-level and region-level OOD tasks. On CPTAC-Lung, CPathAgent achieves 88.1% balanced accuracy with only 5,254 training WSIs, outperforming CPath-Omni (82.3%, 11,728 WSIs) and PRISM (83.2%, 587,196 WSIs), while slightly below TITAN (91.9%, 335,645 WSIs). Notably, CPathAgent uses $64\times$ fewer WSIs than TITAN, $2.2\times$ fewer than CPath-Omni, and $112\times$ fewer than PRISM, yet achieves competitive or superior performance. For BRACS region classification, CPathAgent achieves 64.3% accuracy, substantially surpassing general LMMs GPT-4.1 (44.2%) and Gemini-2.5-Pro (49.0%), as well as the pathology-specialized CPath-Omni (57.9%), demonstrating its capability in huge regions analysis.

While current OOD performance may not yet meet clinical deployment standards, these results demonstrate meaningful generalization beyond TCGA with remarkable data efficiency, suggesting that scaling to larger datasets represents a promising direction for further improvement.

## 5 Conclusion

In this study, we present CPathAgent, an agent-based framework for computational pathology that emulates pathologists' diagnostic reasoning through strategic examination of high-resolution pathology images. Through a systematic three-stage process of global screening, navigation planning, and multi-scale sequence reasoning, CPathAgent faithfully replicates the critical diagnostic workflow by dynamically zooming in and out while systematically shifting focus across regions of interest, transparently documenting not only what features are observed but also why specific regions warrant closer examination and how evidence accumulates across viewing scales, thereby achieving both superior interpretability and state-of-the-art performance across diverse pathology tasks. Our comprehensive evaluation spanning three scales of image analysis (patch-level understanding, large regional assessment, and whole slide image analysis) demonstrates that this agent-based paradigm, which closely mirrors how experienced pathologists navigate and scrutinize tissue specimens, effectively bridges the crucial gap between human diagnostic expertise and AI-assisted analysis.

# 6 Acknowledgements

This study was partially supported by "Pioneer" and "Leading Goose" R&D Program of Zhejiang (Grant 2025SDXHDX0003), the National Natural Science Foundation of China (Grant No.62506306), foundation of Muyuan Laboratory (Program ID: 14106022401,14106022402), and the Westlake Education Foundation.

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

# CPathAgent: An Agent-based Foundation Model for Interpretable High-Resolution Pathology Image Analysis Mimicking Pathologists' Diagnostic Logic
## *Supplementary Material*

## Table of Contents

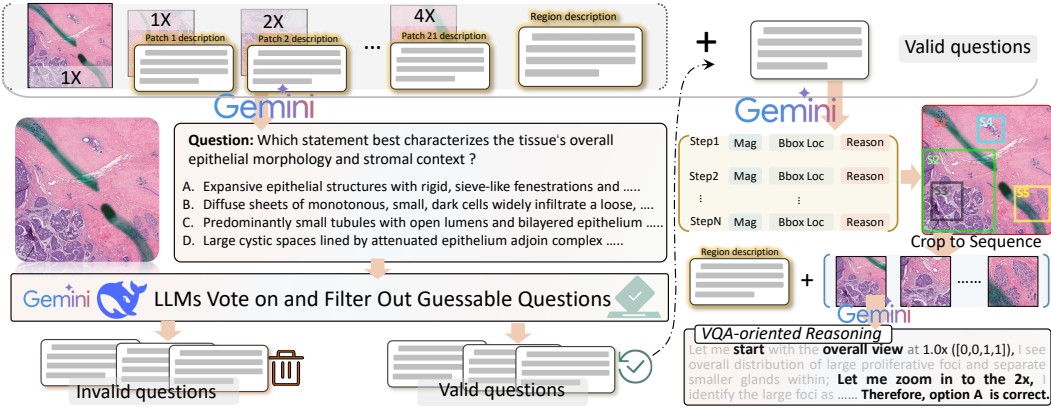

Figure A1: Overview of the VQA-oriented dataset generation process: from multi-scale description to question generation, question filtering, question-oriented navigation path generation, and VQA-related visual reasoning.

# A  Additional Details of Proposed Dataset

In this section, we present detailed information about our datasets: CPathAgent-Instruct and PathMMU-HR². We provide comprehensive statistical analysis of both datasets, along with further details regarding their construction process.

## A.1  Details of CPathAgent-Instruct

**Data Sources:** CPathAgent derives its WSI report data from HistGen [49], while the corresponding WSI data is sourced from The Cancer Genome Atlas (TCGA). TCGA provides a comprehensive collection of pathology WSIs contributed by various participating institutions, encompassing a diverse range of distinct tissue and cancer types. This extensive repository ensures diverse coverage of pathological conditions and features essential for dataset and model development.

**Additional Details of CPathAgent-Instruct:** Our dataset utilizes 5,254 WSIs, which is 80% of the total HisGen data for constructing CPathAgent-Instruct. The remaining WSIs are reserved for model testing and for constructing the PathMMU-HR² dataset.

We provide the following additional dataset construction details that could not be included in the main text due to space constraints:

*1) Global Screening Subset Enhancement:* For the region selection subset within CPathAgent-Instruct, we face a challenge as this capability was not developed during earlier training stages (stage 1 and stage 2 as detailed in Section 3.4), and 5,254 WSIs are insufficient for acquiring this skill. Considering that region selection is manageable for advanced large models like Gemini-2.5-Pro and doesn't heavily depend on WSI reports, we expand our data by utilizing all 24,429 TCGA overview images (excluding test data). The WSI overview images are 32× downsampled versions of the original WSIs as mentioned in the main paper, providing a overview the entire slide at lower resolution. To be specific:

- For WSIs with associated pathology reports, we prompt Gemini-2.5-Pro with both the WSI overview image and the corresponding report. The prompt specifically guides the model in identifying and selecting suspicious regions, particularly those exhibiting pathological features explicitly mentioned in the report.

- For WSIs without accompanying reports, we provide only the WSI overview image, prompting the Gemini-2.5-Pro to identify regions showing abnormal tissue characteristics or potential pathological features by leveraging Gemini-2.5-Pro's powerful visual recognition capabilities.

To increase the diversity of our data, we use a generation temperature of 0.8 and create two separate region selection results for each WSI. This strategy produces slightly different region selections from the same WSI overview, effectively doubling our dataset size. At the same time, it introduces natural variations in how regions are selected, which reflects the inherent subjectivity in how different

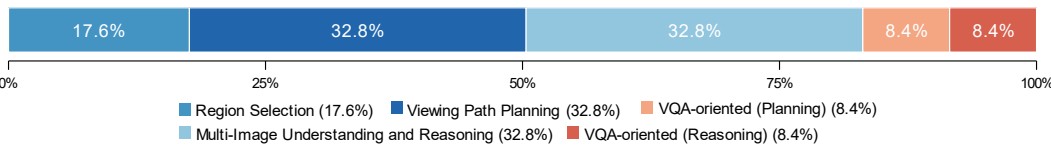

Figure A2: Illustration of the proportional distribution of each subset within the CPathAgent-Instruct dataset.

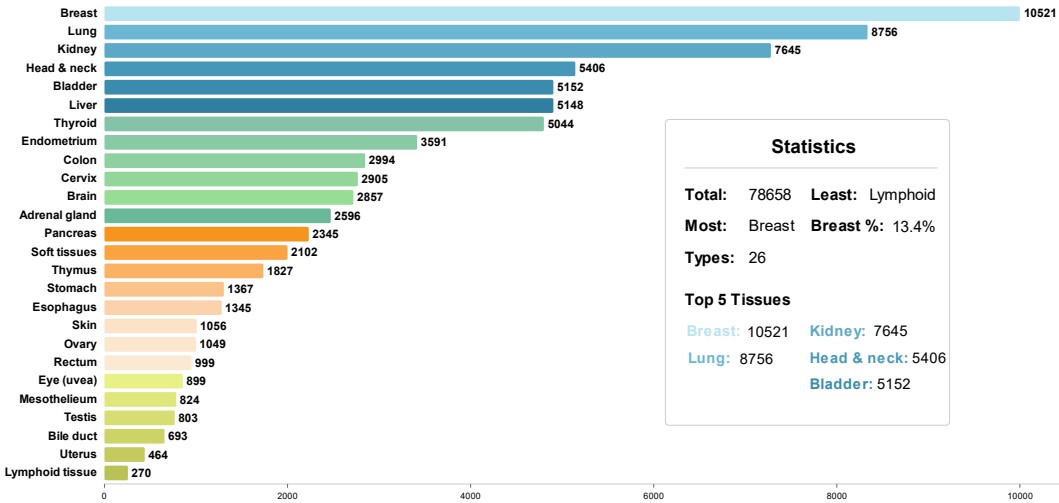

Figure A3: Illustration of the distribution of regions across tissue types.

pathologists might evaluate the same slide. The specific prompt templates we use can be found in Section E.5.

***2) Navigation Path Diversity Augmentation:*** Similarly, for the navigation path planning subset, we recognize that pathologistsmay follow multiple valid trajectories when examining a large region to reach accurate diagnoses. To capture this inherent diversity, we set the generation temperature to 0.8 and prompt Gemini-2.5-Pro randomly 1-3 times per WSI using the approach described in Section 3.2. This stochastic prompting process generates diverse navigation paths for identical regions. Correspondingly, the multi-scale multi-image reasoning also changes based on the different navigation paths, creating additional variations in our dataset. As a result, we substantially expand our dataset while improving the model's ability to generate diverse, valid trajectories.

***3) VQA-oriented Subset Construction and Quality Control:*** We systematically construct the VQA subset as shown in Figure A1. First, we use 21 multi-scale descriptions from sub-patches at different magnifications (1x, 2x, 4x), which are generated through WSI-report guided prompting (detailed in Section 3.2). We then input these descriptions along with their corresponding region images to Gemini-2.5-Pro to create the initial VQA pairs.

However, we recognize that models often exploit textual shortcuts when answering VQA questions rather than performing genuine image analysis. Inspired by PathMMU [48], we present text-only questions to both Gemini-2.5-Pro [57] and DeepSeekV3 [62], then eliminate any questions both models answer correctly without visual input. This ensures our dataset exclusively targets vision-dependent reasoning tasks that cannot be solved through textual cues alone.

For questions that pass validation, we proceed to generate navigation plans through a two-stage process. First, we provide Gemini-2.5-Pro with the validated question, multi-scale image descriptions, region descriptions, and full region images to generate a navigation path. Second, following the approach established in our region description reasoning generation, we combine the validated question with (1) cropped patches extracted along the planned navigation path and (2) corresponding region descriptions. This combined input prompts Gemini-2.5-Pro to generate comprehensive reasoning that systematically addresses the question—progressing logically from overall tissue architecture to specific diagnostic features, comparing and eliminating answer options through continuous reasoning to arrive at the correct conclusion.

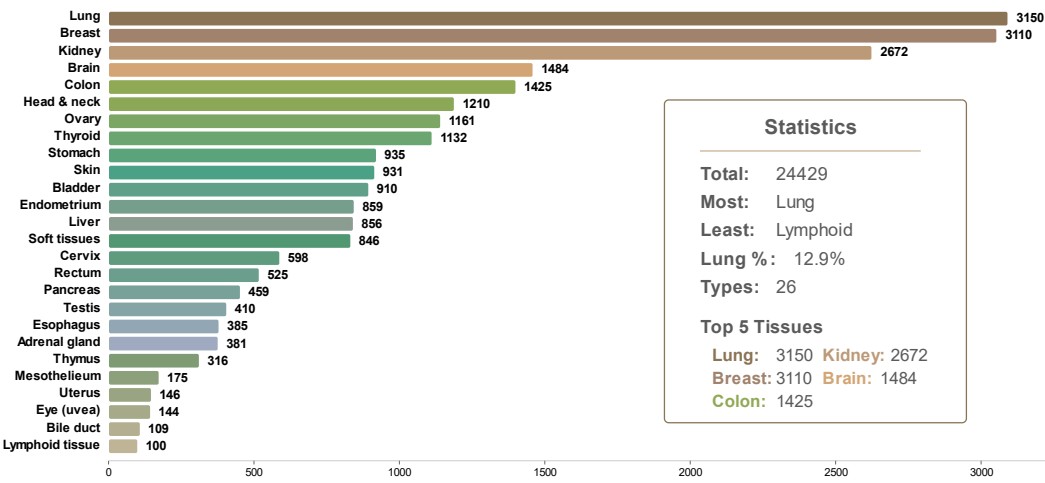

Figure A4: Illustration of the distribution of regions across tissue types.

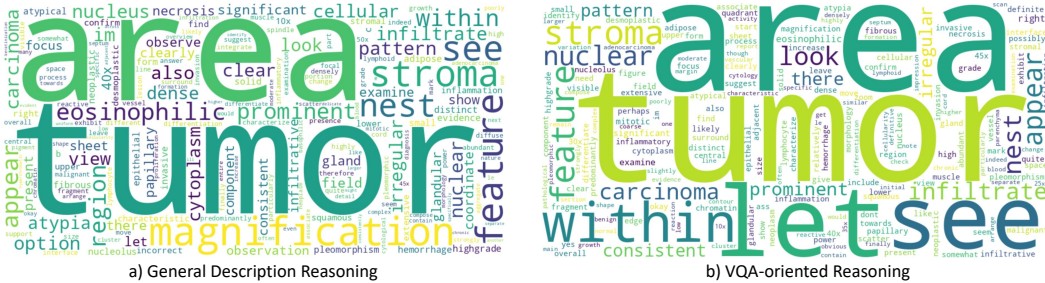

Figure A5: Visualization of a word cloud derived from general description-based and VQA-oriented reasoning.

This question-guided navigation and reasoning framework yields a dataset that enables CPathAgent to dynamically adapt to diverse diagnostic queries by focusing on task-relevant regions and features based on the specific question at hand.

**Proportion of CPathAgent-Instruct subsets** Figure A2 illustrates the distribution of different subsets within the CPathAgent-Instruct dataset. Among them, "Viewing Path Planning" and "Multi-Image Understanding and Reasoning" each constitute 32.8% of the total dataset. These equal proportions exist because each planning instance directly corresponds to a reasoning instance, creating a one-to-one relationship between these components. The VQA-oriented subset, though smaller, is sufficient for model adaptation as it shares structural similarities with the main components.

**Statistics of Source Regions:** As shown in Figure A3, CPathAgent dataset encompasses 26 distinct tissue types with a total of 78,658 individual regions. Breast tissue represents the highest proportion at 13.4% (10,521 samples), while lymphoid tissue appears with 270 samples.

**Statistics of Overview WSIs within region selection subset:** As shown in Figure A4, the CPathAgent dataset consists of 26 distinct tissue types with a total of 24,429 WSI overviews. Lung tissue represents the highest proportion at 12.9% (3,150 WSIs), followed by Breast with 3,110 WSIs and Kidney with 2,672 WSIs.

**Word Frequency in Multi-scale Multi-image Reasoning Subsets:** We analyze word frequency distributions in our General description-based (left, Figure A5) and VQA-oriented (right, Figure A5) reasoning approaches. The visualization reveals distinctive reasoning patterns in pathology diagnosis. General description reasoning emphasizes procedural terms like "zoom," "move," and "examine"—simulating a pathologist's physical navigation through tissue samples as they methodically adjust magnification, reposition focus areas, and explore specimens. Meanwhile, VQA-oriented reasoning prioritizes diagnostic decision terms like "look," "see," and "feature," reflecting hypothesis testing and evidence evaluation in clinical reasoning. Both approaches incorporate critical pathological terminology ("stroma," "nuclear," "infiltrate") essential for accurate diagnosis, demonstrating

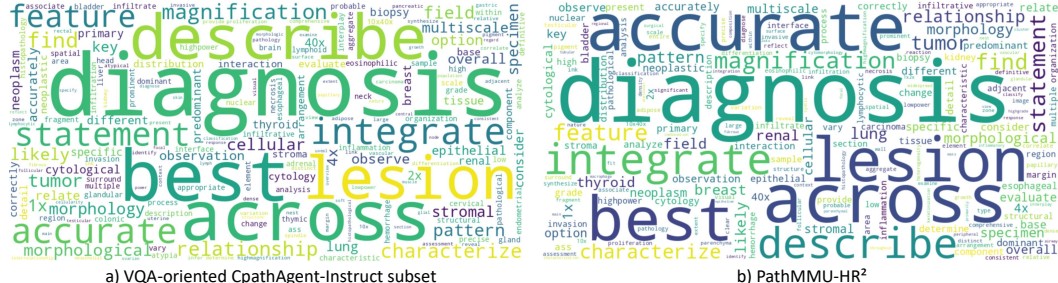

a) VQA-oriented CpathAgent-Instruct subset        b) PathMMU-HR²

Figure A6: Visualization of a word cloud derived from questions in VQA-oriented subsets in CPathAgent-instruct and PathMMU-HR².

that these reasoning strategies center on key diagnostic elements fundamental to clinical pathology assessment and disease classification.

**Word Frequency of Questions in VQA-oriented Subset:** We analyze the question terminology in our VQA-oriented subset, as shown in the left part of Figure A6. The visualization reveals "diagnosis" as the central focus, surrounded by practical terms like "describe," "integrate," and "feature." Magnification terms ("40x," "10x") indicate how pathologists navigate between different viewing scales, while specific organs ("thyroid," "breast," "lung") appear frequently in questions. Action words like "evaluate" and "observe," along with uncertainty terms like "likely" and "consider," demonstrate the methodical reasoning process. This word pattern demonstrates that our constructed diagnostic questions demand multiscale examination with systematic feature assessment.

## A.2 Details of PathMMU-HR²

The construction of PathMMU-HR² is similar to the construction of the VQA-oriented subset in CPathAgen-Instruct, as shown in Figure A1. However, they differ in that after filtering VQA pairs that can be solved through text-only shortcuts, we engage three pathologists with over 10 years of experience for human validation. The evaluation criteria include: clinical relevance of questions, accuracy of provided answers, necessity of multi-scale integration for answering, and alignment with standard pathology practice. Questions failing any criterion are deemed invalid. Under pathologists' evaluation, we identify 1,668 valid questions from the initially annotated 2,200 questions, with the annotation platform interface shown in Figure A7.

We also visualize the question part of PathMMU-HR² with a word cloud in Figure A6. As shown, the most prominent terms—such as "diagnose," "integrate," "accurately," and "across"—highlight that many queries require multi-scale analysis and complex reasoning. The presence of micro-level descriptors like "nuclear," "cellular," and "morphology," alongside macro-level terms such as "tumor," "lesion," and "pattern" underscores the necessity of integrating observations across different magnification levels (indicated by terms like "40x," "10x," and "2x"). This multi-scale integration is essential for correctly answering questions that demand the model to synthesize information from subcellular details to tissue architecture, evaluating both cytological features and architectural patterns simultaneously to arrive at accurate answers.

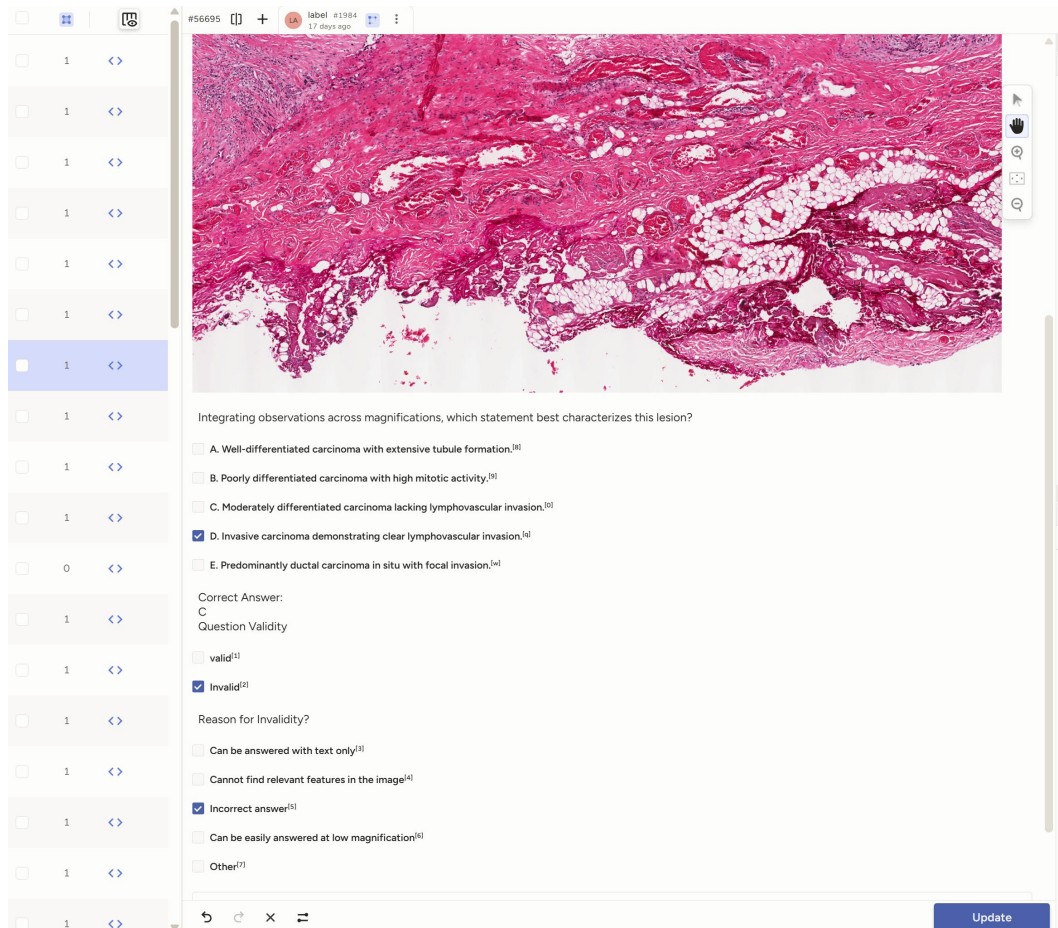

Figure A7: Example of the PathMMU-HR2 annotation interface.

# B  Additional Experiments and Discussion

## B.1  Comparison of Description Generation Performance Among Pathology-specific LMMs

We randomly sample 100 region samples from TCGA and compare them with two advanced pathology-specific trained LMMs, PathGen-LLaVA [26] and Quilt-LLaVA [47], to generate descriptions for these regions. To assess model performance, we utilize Gemini-2.5-Pro and GPT-4.1 as evaluators, with outputs categorized as wins, losses, or ties when performance is comparable. To ensure clinical relevance of the evaluation, we also engage a professional pathologist with over 10 years of clinical experience to provide expert assessment.

***Results: CPathAgent significantly outperforms both PathGen-LLaVA and Quilt-LLaVA across all evaluation metrics.*** As shown in Figure A8, most notably, CPathAgent achieves a perfect 100% win rate against Quilt-LLaVA across all three evaluators. Against PathGen-LLaVA, CPathAgent maintains consistently strong performance with over 96% win rates across all evaluators.

The evaluation consistency between human and LMM evaluators is remarkably high. However, we observe subtle differences in evaluation patterns: compared to the general-purpose LMMs, the pathologist evaluator awards PathGen-LLaVA slightly more ties and one additional win. We hypothesize this difference stems from the distinct evaluation priorities between expert pathologists and LMMs. General-purpose LMMs, may favor CPathAgent's comprehensive reasoning, while the pathologist evaluator places equal weight on both reasoning quality and diagnostic accuracy, resulting in a more reliable assessment that recognizes PathGen-LLaVA's clinical precision in some cases.

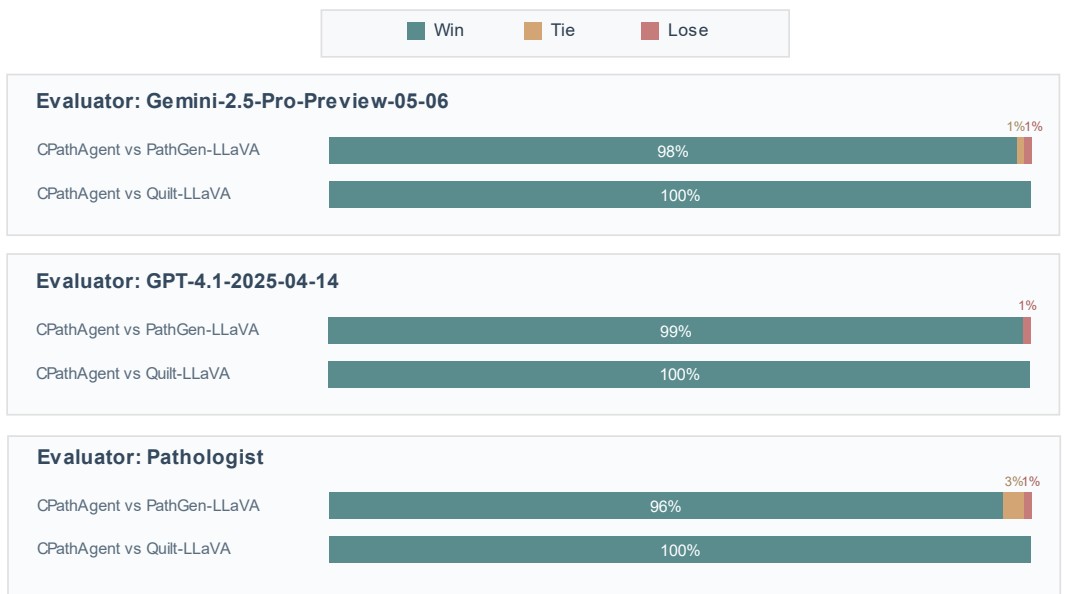

Figure A8: Comparison of CPathAgent with different pathology-specific LMMs for description/diagnosis generation on 100 randomly sampled cases, evaluated by Gemini-2.5-Pro, GPT-4.1, and expert pathologists.

## B.2  Pathologist evaluation results

We invited a pathology expert to evaluate both diagnostic description generation and VQA tasks across 40 regions, assessing whether the navigation path planning and multi-scale multi-view reasoning align with clinical logic. The model was run 8 times to calculate pass@k metrics.

***Results: Pathologist evaluation confirms that both navigation paths and diagnostic reasoning meet clinical expectations.*** As shown in Figure A9, the model demonstrates strong performance with both tasks. For navigation path evaluation, CPathAgent's initial satisfaction rates are acceptable but modest, starting at 70% for general description generation and 77.5% for VQA. However, the pass@k metric demonstrates promising improvement when increasing model runs, with both tasks ultimately exceeding 85% satisfaction. For reasoning evaluation, satisfaction rates start higher at 82.5% for

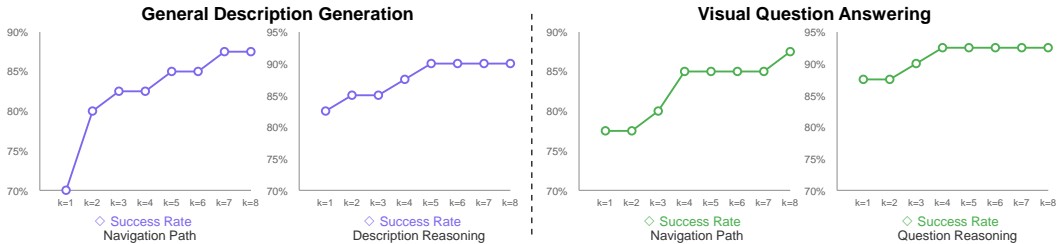

Figure A9: Success rates (pass@k) across 8 generations, evaluated by pathologists across 40 regions. Results show the percentage of pathologist satisfaction with navigation paths and reasoning for General Description Generation (left) and VQA (right) as the number of generations increases.

Table A4: Overall results of models on the PathMMU-Pro. The best-performing model in each subset is **in-bold**.

| | Test Overall | | PubMed | | SocialPath | | EduContent | |
| --- | --- | --- | --- | --- | --- | --- | --- | --- |
| | Tiny (430) | ALL (4755) | Tiny (147) | ALL (1865) | Tiny (136) | All (1081) | Tiny (147) | All (1809) |
| Expert performance | **69.4** | - | **71.2** | - | **70.1** | - | 66.9 | - |
| CPath-Omni | 62.1 | 61.8 | 60.5 | 60.8 | 58.8 | 62.1 | 66.7 | 63.0 |
| CPathAgent (Ours) | 66.3 | 65.3 | 64.6 | 65.8 | 65.4 | 63.6 | **68.7** | 66.2 |

general description generation and 87.5% for VQA, and show continued improvement, reaching 90% for general description generation and 92.5% for VQA at higher k values.

Across both tasks, the consistent upward trend in pass@k metrics suggests that the model can reliably generate clinically sound navigation strategies and reasoning when given multiple attempts. Additionally, VQA-based approaches consistently outperform general description tasks across both navigation planning and reasoning. We attribute this to VQA's focused attention on target features that enables better identification of key elements, whereas general descriptions require additional judgment about content prioritization.

## B.3  Additional Results on PathMMU-Pro

While CPathAgent demonstrates strong performance on PathMMU, achieving results that exceed the reported expert baselines, we acknowledge that the observed gap between our model and expert performance can be attributed to two primary factors inherent to the benchmark design. First, the expert annotations in PathMMU were conducted by pathologists with specialized expertise in one or two specific domains, whereas the dataset encompasses a broad spectrum of tissue types and pathological conditions across multiple subspecialties. Consequently, no individual expert possessed comprehensive knowledge across all domains represented in the evaluation set. Second, as discovered by the PathMMU, large multimodal models may exploit text-based reasoning shortcuts, leveraging strong language priors and contextual cues from question formulations that are not readily accessible to human experts performing traditional diagnostic tasks.

To address the concern regarding text-based shortcuts and provide a more rigorous assessment of our model's genuine pathological reasoning capabilities, we conducted additional evaluation on PathMMU-Pro [63], a refined version of the benchmark specifically designed with more carefully constructed questions that minimize textual biases.

***Results: On PathMMU-Pro, CPathAgent no longer surpasses expert performance but maintains a substantial lead over the previous SOTA CPath-Omni***. As shown in Table A4, expert performance now leads with an average accuracy of 69.4%, surpassing CPathAgent's results. This shift substantiates that the more rigorous question design in PathMMU-Pro effectively mitigates the text-based guessing capabilities that LMMs could previously exploit. Nevertheless, CPathAgent demonstrates consistent advantages over CPath-Omni, achieving 66.3% versus 62.1% on the Test-Tiny subset and 65.3% versus 61.8% on the Test-All subset, demonstrating that CPathAgent possesses stronger multimodal understanding capabilities for pathology.

Table A5: Results of ablation study on agent components for WSI classification. Balanced accuracy (%) is reported.

| Model | TCGA-BRCA | TCGA-NSCLC | TCGA-RCC | TCGA-ESCA | TCGA-BLCA | TCGA-THCA | Avg. |
|---|---|---|---|---|---|---|---|
| Full CPathAgent | **88.5** | **90.8** | **94.6** | **97.1** | **62.7** | **63.2** | **82.8** |
| w/o Gloabal Screening | 84.7 | 88.7 | 94.6 | 94.1 | 60.8 | 58.9 | 80.3 |
| w/o Navigation & Reasoning | 80.0 | 83.6 | 88.7 | 94.1 | 52.2 | 51.3 | 75.0 |

Table A6: Results of ablation study on applying Gemini-2.5-Pro as an agent for WSI tasks. Balanced accuracy (%) is reported

| | TCGA-BRCA | TCGA-NSCLC | TCGA-RCC | TCGA-ESCA | TCGA-BLCA | TCGA-THCA | Avg. |
|---|---|---|---|---|---|---|---|
| **General Large Multimodal Models** | | | | | | | |
| Gemini-2.5-Pro-Preview-03-25 | 72.4 | 89.2 | 69.2 | **97.1** | 59.8 | 44.9 | 72.1 |
| **Agent-based Approach** | | | | | | | |
| Gemini-2.5-Pro-Preview-03-25 | 52.0 | 73.4 | 58.8 | **97.1** | **64.7** | 46.3 | 65.4 |
| CPathAgent (Ours) | **88.5** | **90.8** | **94.6** | **97.1** | 62.7 | **63.2** | **82.8** |

## B.4 Ablation Study on Agent Components

To assess the contribution of each agent component, we conduct systematic ablation studies by isolating individual module impacts. We evaluate two critical components. For *global screening*, we replace the learned region selection module with random selection of 70% of WSI regions. For *navigation planning and multi-scale reasoning*, we ablate the dynamic navigation and multi-view analysis by directly generating descriptions from entire regions at once, followed by WSI classification based on these single-view descriptions.

***Results: Both agent components contribute meaningfully to performance, with navigation and multi-scale reasoning being particularly critical for diagnostic accuracy.*** As shown in Table A5, removing strategic global screening leads to a 2.5% performance drop (from 82.8% to 80.3%), demonstrating that learned region selection effectively identifies diagnostically informative areas and mimics pathologists' selective attention during screening. The impact of removing navigation and multi-scale reasoning is substantially more pronounced, with performance degrading by 7.8% (from 82.8% to 75.0%). This significant drop reveals two key limitations of direct WSI region processing: (1) directly processing high-resolution regions (up to 16000×16000 pixels) necessitates aggressive downsampling to fit model input constraints, causing critical diagnostic details to become indiscernible; (2) the absence of pathologist-like reasoning that strategically examines multiple fields of view at varying magnifications fundamentally limits the model's diagnostic capability. These results confirm that CPathAgent's effectiveness stems from the integration of all components, with navigation planning and multi-scale reasoning being particularly essential for achieving high diagnostic accuracy on WSI tasks.

## B.5 Ablation Study on Applying Gemini-2.5-Pro as an Agent for WSI Tasks

We also apply CPathAgent's inference prompts to directly prompt Gemini-2.5-Pro, enabling it to perform agent-based reasoning and diagnosis similar to CPathAgent.

***Results: Interestingly, while this approach yields modest improvements on TCGA-BLCA and TCGA-THCA datasets, the overall performance decreases by 6.7% (from 72.1% to 65.4%)***, as shown in the "Agent-based Approach" part of Table A6. This contrasts with the findings in Section 4.2 of the main paper, where applying agent-based approaches to advanced LMMs on large regions VQA demonstrated performance gains.

We attribute this discrepancy to the fundamental differences between task types. Since our WSI classification task involves first generating descriptions for all regions and then performing classification based on these descriptions, unlike VQA tasks that provide clear directional guidance for navigation planning, general description tasks lack explicit objectives to guide the agent's exploration strategy. Consequently, when directly applying the agent approach to Gemini-2.5-Pro, the model struggles with effective navigation path planning within large regions, as it lacks clear guidance on which critical areas to examine. Since subsequent reasoning heavily depends on the planned navigation path, performance suffers significantly, particularly on datasets like TCGA-BRCA and TCGA-NSCLC.

For example, we observe that on TCGA-NSCLC (Non-Small Cell Lung Cancer) tasks, where all WSIs belong to Non-Small Cell Lung Cancer and the task is to classify subtypes into lung adenocarcinoma and lung squamous cell carcinoma, after applying the agent-based approach, Gemini-2.5-Pro sometimes even misclassifies samples as Small Cell Lung Cancer, demonstrating fundamental diagnostic errors or hallucinations.

This finding validates our progressive training data generation framework, which leverages comprehensive WSI reports to guide region-specific descriptions. These targeted descriptions enable us to prompt Gemini-2.5-Pro to capture multi-scale morphological features across different viewing perspectives. By understanding which locations contain specific pathological features, we can further prompt Gemini-2.5-Pro to generate higher-quality navigation paths with enhanced spatial awareness, which in turn enables more precise and accurate multi-scale, multi-view reasoning generation. This creates a hierarchical sequence where each stage builds upon the quality of previous stages, resulting in cascading improvements throughout the entire data generation pipeline.

## C  Experimental Details

### C.1  Details for MIL-based WSI classification

#### C.1.1  Details for Training and Testing Dataset

To ensure a fair comparison, MIL-based methods are evaluated using identical training and testing datasets such as CPathAgent and other comparative LMMs. All datasets were derived from WSI samples corresponding to the HistGen [49] WSI reports. We select representative subsets suitable for whole slide image classification tasks, comprising the following datasets:

**TCGA-RCC** (Renal Cell Carcinoma, train: 422, test: 238): A three-class classification task including kidney chromophobe, kidney renal clear cell carcinoma, and kidney renal papillary cell carcinoma.

**TCGA-NSCLC** (Non-Small Cell Lung Cancer) (train: 642, test: 195): A binary classification task distinguishing between lung adenocarcinoma and lung squamous cell carcinoma.

**TCGA-ESCA** (Esophageal Carcinoma, train: 83, test: 31): A binary classification task differentiating adenomas and adenocarcinomas from squamous cell neoplasms.

**TCGA-BRCA** (Breast Invasive Carcinoma, train: 637, test: 226): A binary classification task between invasive ductal carcinoma and invasive lobular carcinoma.

**TCGA-THCA** (Thyroid Carcinoma, train: 323, test: 119): A three-class classification task encompassing papillary adenocarcinoma, papillary carcinoma columnar cell, and papillary carcinoma follicular variant.

**TCGA-BLCA** (Bladder Urothelial Carcinoma, train: 227, test: 79): A binary classification task distinguishing papillary transitional cell carcinoma from transitional cell carcinoma.

Additionally, we extract 10% of WSIs from the complete TCGA dataset that are not included in the HistGen dataset, as training the MIL approach requires a separate validation set. Models with the best performance on the validation set are selected for evaluation on the test set.

#### C.1.2  Details for MIL preprocessing

Our WSI preprocessing pipeline follows the approach established by CPath-Omni [10] to implement a multi-scale MIL-based methodology. The preprocessing workflow consists of several key steps: First, we employ CLAM [64] to automatically identify and segment tissue regions by applying appropriate segmentation threshold. From these identified regions, we extract non-overlapping patches of 2048 × 2048 pixels at 40× magnification. To ensure tissue quality, patches are only retained if they contain more than 10% valid tissue area. To capture multi-scale information, each 2048 × 2048 patch undergoes hierarchical subdivision: we maintain the original 2048 × 2048 patch while simultaneously extracting four 1024 × 1024 patches and sixteen 512 × 512 patches from the same region. Features are extracted from all these multi-scale patches using CPath-CLIP [10] and subsequently averaged to generate a unified representation for each 2048-resolution region.

#### C.1.3  Details for WSI Task-Specific Fine-Tuning

All models are trained for 20 epochs using a fixed learning rate of $1 \times 10^{-5}$ without any learning rate scheduling. We employ the Adam optimizer without weight decay, maintaining a batch size of 1 throughout training. To ensure reliable evaluation, we conduct experiments across 5 different random seeds and report the averaged results.

**Model Architecture.** The MIL framework commonly used for WSI classification includes three learnable components: (1) a fully-connected layer to reduce the dimensionality of features to 256, (2) an attention network to aggregate and transform the instance features, and (3) a final fully-connected layer for making predictions. We experiment with ABMIL [4] and DSMIL [5], both of which use the same fully-connected layers for reducing feature dimensionality and making predictions. For the attention network, ABMIL uses a gated attention mechanism, while DSMIL introduces a dual-stream architecture.

## C.2 Details for Patch Level Benchmark

**PathMMU** [48]: An expert-validated pathology benchmark comprising 33,428 multimodal multiple-choice questions and 24,067 images from diverse sources including PubMed, pathology atlases, expert social media posts, and educational content. Constructed using GPT-4V with rigorous validation by seven pathologists, it covers multiple organ systems and pathology subjects for expert-level evaluation of LMMs' pathology understanding and reasoning capabilities.

**PathMMU-Pro** [63]: Building upon PathMMU, PathMMU-Pro introduces more strict assessment to address a critical limitation in multimodal evaluation: the potential for models to solve questions using text-only reasoning without truly leveraging visual information. PathMMU-Pro employs a systematic two-step refinement process: (1) training specialized text-only "question-guessing" models that receive only textual inputs to identify and filter out questions answerable without examining pathology images, and (2) enhancing remaining questions by generating more confusing options that are textually similar but require careful visual examination to distinguish, thereby preventing models from guessing answers through superficial text patterns. This enhanced benchmark provides a more strict assessment of models' intrinsic visual reasoning capabilities in pathology.

## C.3 Hardware and Training Cost of CPathAgent

We employ 8 NVIDIA H800-80G GPUs for CPathAgent training, with computational requirements of 9 hours for Stage 1 (multimodal alignment), 25 hours for Stage 2 (pathology task training), and 39 hours for Stage 3 (agent capability development).

## C.4 Training Hyperparameters

For Stages 1 and 2 of CPathAgent training, we follow the hyperparameter configuration of CPath-Omni. For Stage 3 training, we employ 8 NVIDIA H800-80G GPUs and train the model for a single epoch using a per-device batch size of 1 with gradient accumulation of 8 steps. We implement a cosine learning rate scheduler with a base learning rate of 1e-5 and a warmup ratio of 0.03, while fine-tuning the vision tower with a reduced learning rate of 2e-6.

# D  Limitations

While CPathAgent successfully demonstrates the feasibility of mimicking pathologists' diagnostic reasoning pathways, several limitations merit future improvement:

1. **Synthetic diagnostic pathways.** Currently, the diagnostic viewing paths employed by CPathAgent are synthetically generated rather than derived from real pathologist workflows. Although our synthetic paths are designed based on established diagnostic principles and demonstrate effectiveness in our experiments, they may not fully capture the nuanced decision-making processes and adaptive strategies employed by experienced pathologists in clinical practice. Future work should incorporate authentic diagnostic paths collected from pathologists through eye-tracking studies or explicit annotation of their reasoning sequences.

2. **Limited task scope.** CPathAgent currently focuses exclusively on diagnostic classification tasks. While this represents a critical clinical application, computational pathology encompasses a broader spectrum of tasks including prognostic prediction and biomarker detection. Extending CPathAgent to these domains is theoretically feasible and would significantly expand its clinical utility.

3. **Training data scale.** Our model was trained on 5,254 WSIs, which, while sufficient to demonstrate proof-of-concept, remains considerably smaller than datasets used by state-of-the-art foundation models such as PRISM (587,196 WSIs) and TITAN (335,645 WSIs). This disparity in scale may limit the model's generalization capabilities and its ability to capture rare morphological patterns. Scaling up the training data represents a clear path toward improved performance and robustness.

# E Examples and Prompts

## E.1 Examples of Region Descriptions Extracted from WSI Report Pairs

As shown in Figure A10, we extract relevant region descriptions from WSI reports by identifying specific diagnostic findings in targeted areas. This process systematically parses pathological reports to highlight key diagnostic information related to the given region, addressing the limitation in TCGA datasets where WSI reports exist but region-specific descriptions are missing.

## E.2 Examples of PathMMU-HR² Samples

Figure A11 shows the examples of PathMMU-HR² dataset. These VQA samples require multi-scale, multi-view reasoning to arrive at accurate diagnoses. As illustrated in the examples, the correct identification of conditions such as fibromatosis (desmoid tumor) and well-demarcated clear cell tumors necessitates careful examination of architectural patterns and cellular features across different magnification levels. The diagnostic process requires the model to analyze tissue structure at lower magnifications while progressively zooming in to evaluate the cellular morphology and stromal characteristics at higher magnifications. This multi-scale analytical approach reflects real-world pathological practice, where definitive diagnoses depend on the integration of macro-architectural features with microscopic cellular details.

## E.3 Examples of Region Selection Process Based on WSI Overview

Figure A12 and FigureA13 demonstrate the generated region selection results. As illustrated, the model segments the WSI overview into distinct regions and identifies which areas require high-magnification examination. For example, in Figure A12, the model correctly prioritizes core tumor areas for detailed analysis while also identifying critical tumor interface/periphery regions and margin assessment areas to evaluate potential residual tumor presence. In addition, the model appropriately excludes the benign/background breast tissue areas from high-magnification requirements, as these regions primarily consist of adipose tissue and fibrous stroma that appear benign and uninvolved at low magnification, thus optimizing the diagnostic workflow by focusing computational resources on clinically significant regions.

## E.4 Examples of Navigation Paths and Multi-Scale Reasoning Generated by CPathAgent

### E.4.1 General Navigation Path Planning and Reasoning

Figures A14 and A15 show how CPathAgent navigates and analyzes pathology images like an expert pathologist. For example, as illustrated in Figure A14, CPathAgent begins with a comprehensive overview at low magnification (1.0x), scanning the entire tissue to identify distinct regions and architectural patterns. In this case, it recognizes adipose tissue on the left and dense cellular areas on the right, establishing the overall tissue landscape.

Next, CPathAgent focuses on the main area of concern, examining the primary tumor mass at moderate magnification (2.5x). Here it identifies irregular cell clusters and abnormal tissue architecture that warrant closer investigation. The agent then increases magnification (4.0x) to evaluate cellular details crucial for grading, such as nuclear pleomorphism, mitotic activity, and tubule formation patterns. CPathAgent systematically examines the tumor's relationship with surrounding tissues, checking the tumor-stroma interface (3.0x) for invasive patterns and examining infiltration into adjacent adipose tissue. It also assesses the immune response by evaluating lymphocytic infiltrates near tumor edges (3.5x) and investigates suspicious ductal structures in other regions (3.0x).

To ensure diagnostic accuracy, CPathAgent performs validation checks by examining additional areas at various magnifications (2.5x, 4.0x), confirming that findings are consistent across different regions of the tissue. This systematic, multi-scale approach allows CPathAgent to integrate architectural observations from low magnification with detailed cellular features from high magnification, ultimately reaching a comprehensive diagnosis of invasive ductal carcinoma with appropriate histological grading—mirroring the thorough diagnostic process used by expert pathologists in clinical practice.

### E.4.2 VQA-oriented Navigation Path Planning and Reasoning

Figure A16 demonstrates CPathAgent's question-guided reasoning approach for VQA task. When presented with a specific question about "morphological patterns describing the relationship between neoplastic cell populations and stromal elements at their interface," CPathAgent tailors its entire analysis strategy to address this inquiry.

CPathAgent begins with a systematic overview at low magnification (1.0x), scanning the tissue to locate regions relevant to the question. It identifies striking heterogeneity across the tissue and recognizes that the lower right quadrant contains the key neoplastic-stromal interface mentioned in the question. With the target area identified, CPathAgent zooms in progressively (2.5x, 4.0x) to characterize the neoplastic component, confirming the presence of nested clear cells with high-grade nuclei. At higher magnification (5.0x), it performs detailed cytological assessment, noting the striking nuclear features that support high-grade classification. CPathAgent then specifically examines the critical interface zone at multiple magnifications (3.5x, 4.0x), systematically documenting the relationship between neoplastic cells and surrounding stromal elements. It identifies key morphological features including the transitional pattern between clear cell neoplasm and eosinophilic tissue, along with dense lymphocytic infiltrate at the junction. Finally, CPathAgent moves to eosinophilic areas (3.0x) to contrast these regions with the neoplastic component, ensuring a comprehensive understanding of the interface characteristics.

This question-driven navigation allows CPathAgent to methodically evaluate multiple-choice options, ultimately selecting the answer that best describes the observed "nested clear cells focally transitioning to solid eosinophilic cells with high-grade nuclei and dense lymphocytes." This demonstrates how CPathAgent's reasoning process is specifically optimized to answer targeted pathological questions through systematic, multi-scale analysis.

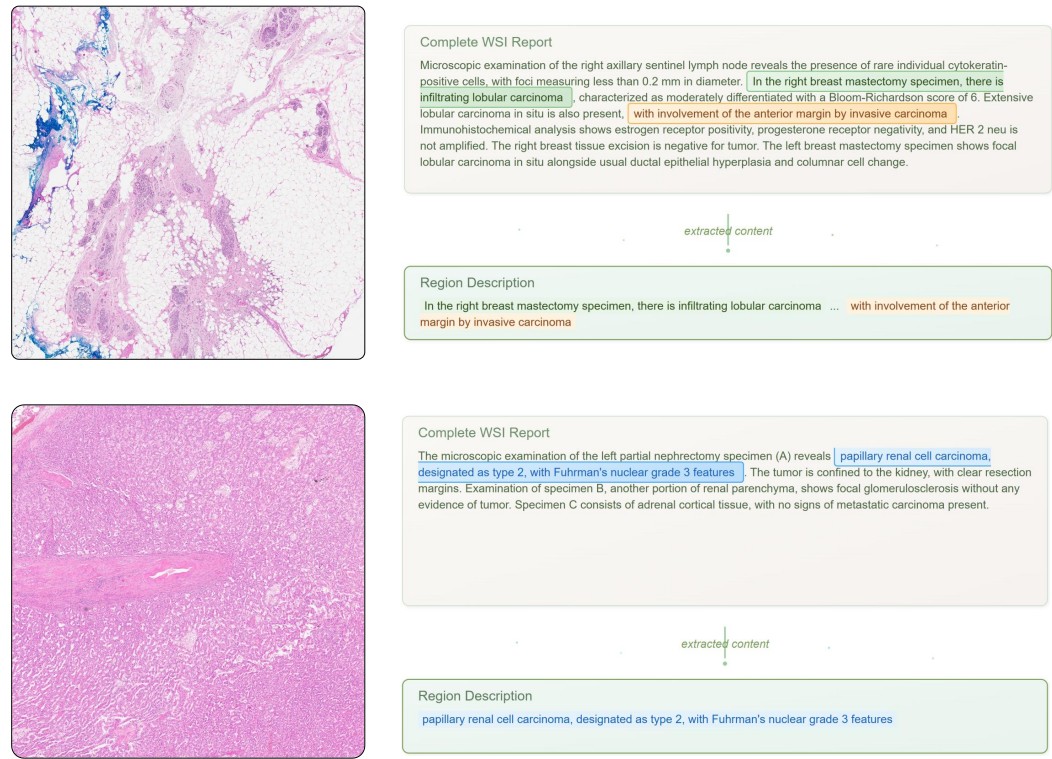

**Complete WSI Report**

Microscopic examination of the right axillary sentinel lymph node reveals the presence of rare individual cytokeratin-positive cells, with foci measuring less than 0.2 mm in diameter. In the right breast mastectomy specimen, there is infiltrating lobular carcinoma , characterized as moderately differentiated with a Bloom-Richardson score of 6. Extensive lobular carcinoma in situ is also present, with involvement of the anterior margin by invasive carcinoma . Immunohistochemical analysis shows estrogen receptor positivity, progesterone receptor negativity, and HER 2 neu is not amplified. The right breast tissue excision is negative for tumor. The left breast mastectomy specimen shows focal lobular carcinoma in situ alongside usual ductal epithelial hyperplasia and columnar cell change.

*extracted content*

**Region Description**

In the right breast mastectomy specimen, there is infiltrating lobular carcinoma   ...   with involvement of the anterior margin by invasive carcinoma

**Complete WSI Report**

The microscopic examination of the left partial nephrectomy specimen (A) reveals papillary renal cell carcinoma, designated as type 2, with Fuhrman's nuclear grade 3 features . The tumor is confined to the kidney, with clear resection margins. Examination of specimen B, another portion of renal parenchyma, shows focal glomerulosclerosis without any evidence of tumor. Specimen C consists of adrenal cortical tissue, with no signs of metastatic carcinoma present.

*extracted content*

**Region Description**

papillary renal cell carcinoma, designated as type 2, with Fuhrman's nuclear grade 3 features

Figure A10: Examples of region description extraction from WSI reports.

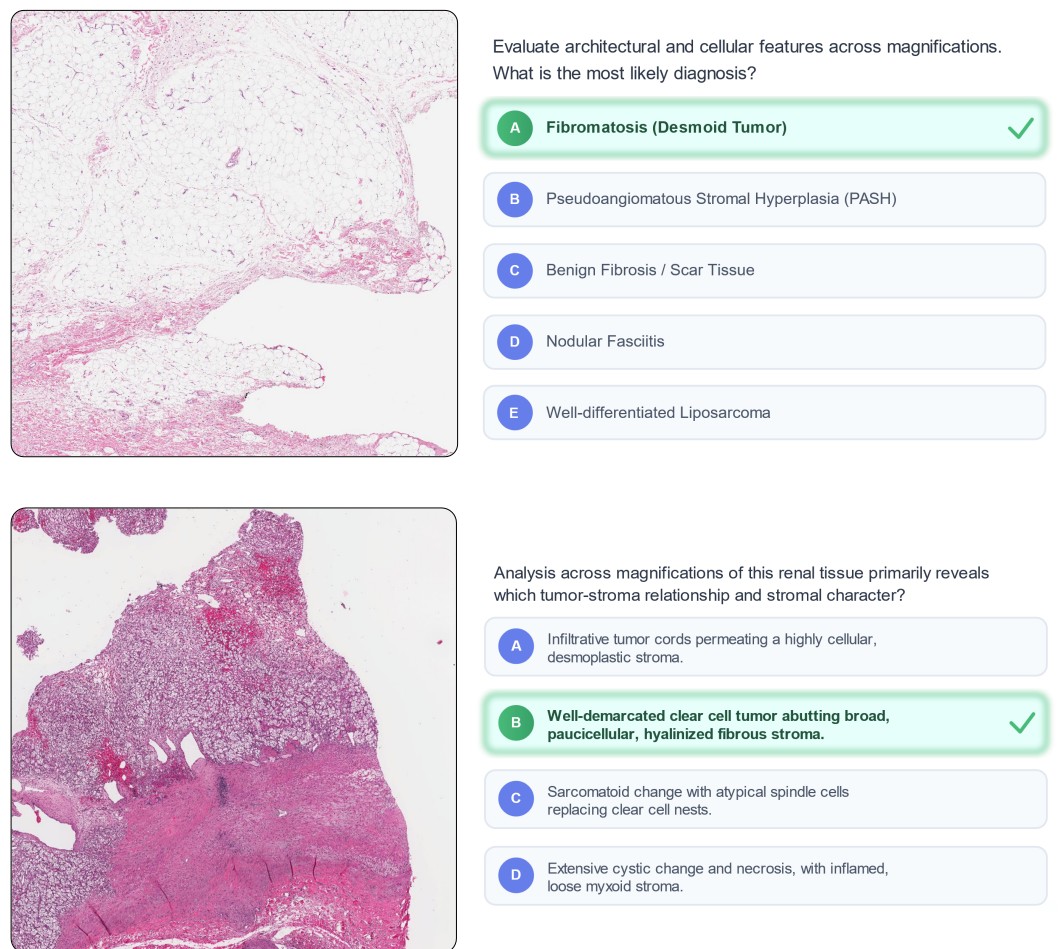

Evaluate architectural and cellular features across magnifications. What is the most likely diagnosis?

A **Fibromatosis (Desmoid Tumor)** ✓

B Pseudoangiomatous Stromal Hyperplasia (PASH)

C Benign Fibrosis / Scar Tissue

D Nodular Fasciitis

E Well-differentiated Liposarcoma

Analysis across magnifications of this renal tissue primarily reveals which tumor-stroma relationship and stromal character?

A Infiltrative tumor cords permeating a highly cellular, desmoplastic stroma.

B **Well-demarcated clear cell tumor abutting broad, paucicellular, hyalinized fibrous stroma.** ✓

C Sarcomatoid change with atypical spindle cells replacing clear cell nests.

D Extensive cystic change and necrosis, with inflamed, loose myxoid stroma.

Figure A11: Examples of VQAs from the PathMMU-HR² dataset.

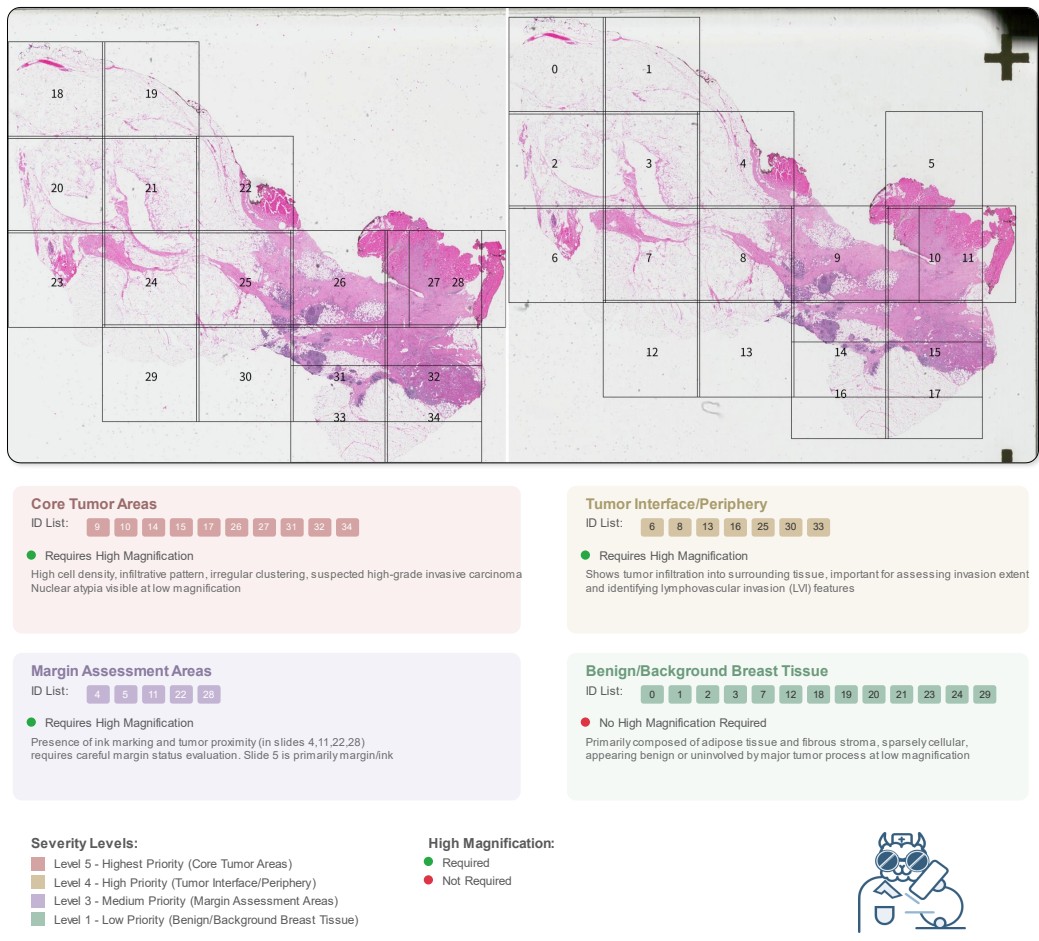

Figure A12: An example of region selection based on a WSI overview.

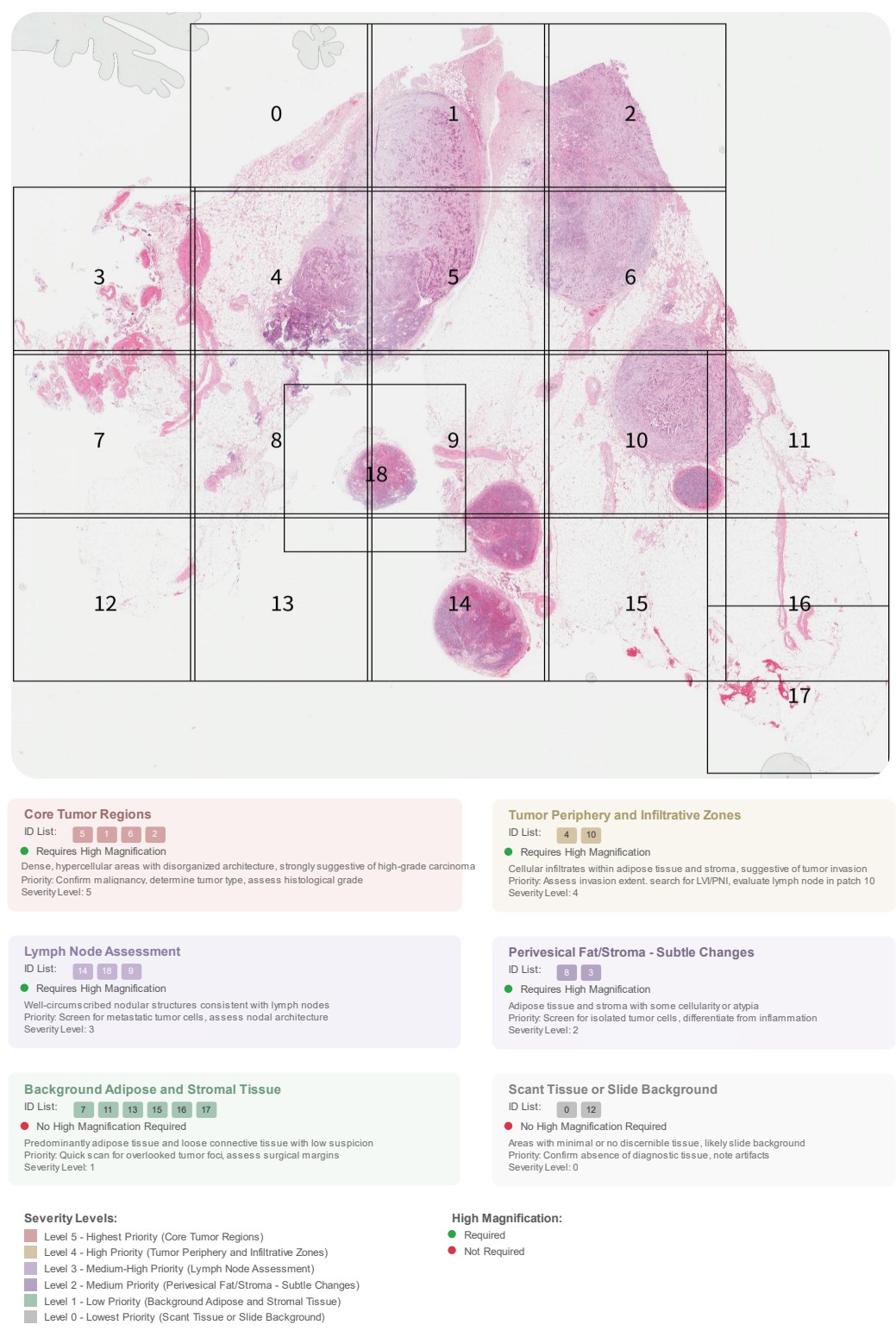

**Core Tumor Regions**
ID List: 5 1 6 2
● Requires High Magnification
Dense, hypercellular areas with disorganized architecture, strongly suggestive of high-grade carcinoma
Priority: Confirm malignancy, determine tumor type, assess histological grade
Severity Level: 5

**Tumor Periphery and Infiltrative Zones**
ID List: 4 10
● Requires High Magnification
Cellular infiltrates within adipose tissue and stroma, suggestive of tumor invasion
Priority: Assess invasion extent, search for LVI/PNI, evaluate lymph node in patch 10
Severity Level: 4

**Lymph Node Assessment**
ID List: 14 18 9
● Requires High Magnification
Well-circumscribed nodular structures consistent with lymph nodes
Priority: Screen for metastatic tumor cells, assess nodal architecture
Severity Level: 3

**Perivesical Fat/Stroma - Subtle Changes**
ID List: 8 3
● Requires High Magnification
Adipose tissue and stroma with some cellularity or atypia
Priority: Screen for isolated tumor cells, differentiate from inflammation
Severity Level: 2

**Background Adipose and Stromal Tissue**
ID List: 7 11 13 15 16 17
● No High Magnification Required
Predominantly adipose tissue and loose connective tissue with low suspicion
Priority: Quick scan for overlooked tumor foci, assess surgical margins
Severity Level: 1

**Scant Tissue or Slide Background**
ID List: 0 12
● No High Magnification Required
Areas with minimal or no discernible tissue, likely slide background
Priority: Confirm absence of diagnostic tissue, note artifacts
Severity Level: 0

**Severity Levels:**
Level 5 - Highest Priority (Core Tumor Regions)
Level 4 - High Priority (Tumor Periphery and Infiltrative Zones)
Level 3 - Medium-High Priority (Lymph Node Assessment)
Level 2 - Medium Priority (Perivesical Fat/Stroma - Subtle Changes)
Level 1 - Low Priority (Background Adipose and Stromal Tissue)
Level 0 - Lowest Priority (Scant Tissue or Slide Background)

**High Magnification:**
● Required
● Not Required

Figure A13: An example of region selection based on a WSI overview.

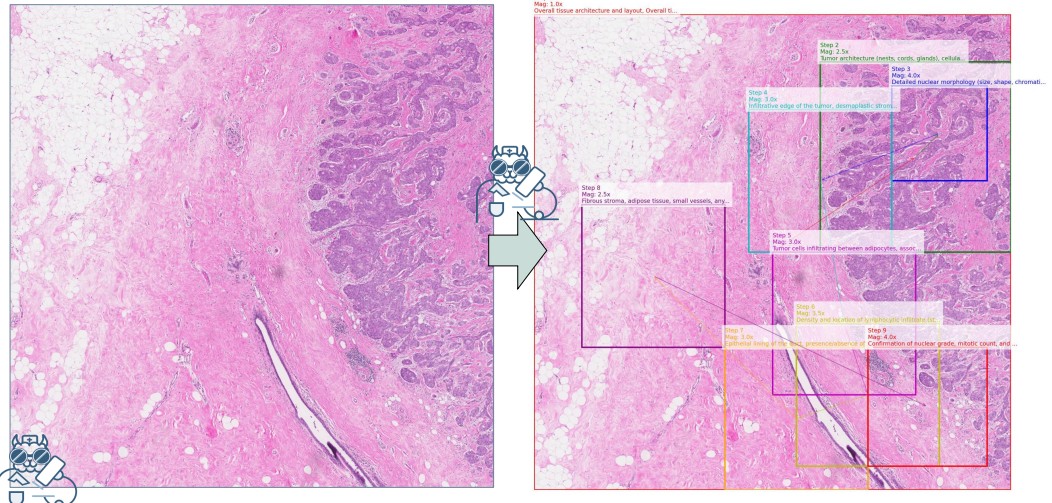

Okay, let's start the examination. At `low power (1.0x)` , `[0, 0, 1, 1]` , the overall architecture shows **a distinct demarcation between adipose tissue on the left and a densely cellular, infiltrative process occupying the right side** , set within a fibrous stroma. There's also **a prominent, somewhat dilated duct-like structure towards the lower central area** . The cellular area appears quite extensive. My **initial impression is an invasive carcinoma** .

Let's **zoom into the main tumor mass** in the **upper right quadrant** `(2.5x)` , `[0.6, 0.1, 1.0, 0.5]` . Here, I can appreciate the tumor architecture better. It's composed of **irregular nests and cords of epithelial cells infiltrating a desmoplastic stroma** . There's **some attempt at gland formation, but it seems limited** . The cellularity is high.

Now, I need **a closer look at the cytology for grading** `(4.0x)` , `[0.75, 0.15, 0.95, 0.35]` . **The nuclei show moderate pleomorphism – variation in size and shape is apparent, maybe 2-3 fold variation** . **Chromatin appears somewhat coarse, and nucleoli are visible but generally not prominent or large** . I'm **scanning for mitotic figures** ... I see a few, but they don't seem particularly numerous in this field. **Tubule formation is definitely present but constitutes less than 75% of the tumor area, probably somewhere between 10-75%** . This points towards **an intermediate grade, likely grade 2** .

Let's **examine the interface between the tumor and the stroma** more centrally `(3.0x)` , `[0.45, 0.2, 0.75, 0.5]` . **The infiltrative pattern is clear, with nests and cords pushing into the surrounding fibrous tissue, which shows a desmoplastic reaction** . I'm **looking carefully for lymphovascular invasion (LVI)** within this stromal interface... I don't see definite LVI in this specific field, but I'll keep an eye out.

**Moving down towards the adipose tissue** `(3.0x)` , `[0.5, 0.5, 0.8, 0.8]` , I can **confirm invasion into fat** . **Tumor nests are clearly infiltrating between adipocytes** , accompanied by the reactive stroma.

Now, let's **assess that lymphocytic infiltrate noted near the tumor edge** `(3.5x)` , `[0.55, 0.65, 0.85, 0.95]` . There's **a moderately dense lymphocytic infiltrate, predominantly stromal, clustering around some tumor nests** . It doesn't appear to be significantly intra-tumoral. This represents a host immune response.

Let's **investigate that large ductal structure** in the **lower central area** `(3.0x)` , `[0.4, 0.7, 0.7, 1.0]` . **The epithelial lining looks relatively bland, perhaps some usual ductal hyperplasia or apocrine metaplasia** , but it doesn't show the cytological atypia of the invasive component or features of high-grade DCIS. It seems distinct from the invasive carcinoma, possibly **a benign entrapped duct or showing low-grade DCIS** , though the latter is less likely given the bland appearance.

Now, I'll **scan the stromal and adipose tissue on the left side** , away from the main tumor mass `(2.5x)` , `[0.1, 0.4, 0.4, 0.7]` , to **check for satellite foci or LVI** . This area consists mainly of **fibrous stroma and mature adipose tissue with some small vessels** . I don't see any definite isolated tumor nests or clear LVI here.

Finally, let's do **a high-power check** in a different area of the tumor, the **lower right** `(4.0x)` , `[0.7, 0.7, 0.95, 0.95]` , to ensure consistency. Yes, the features are similar here: **moderate nuclear pleomorphism, visible nucleoli, limited tubule formation, and occasional mitotic figures** . **The overall impression remains consistent with an invasive ductal carcinoma, Nottingham grade 2** .

**The differential diagnosis would include other types of invasive carcinoma, but the morphology strongly favors ductal type, NOS (No Special Type)** . There's no evidence of lobular features (lack of single-filing, presence of cohesion), metaplastic elements, or other specific subtypes. **The presence of some tubule formation and moderate pleomorphism fits well with grade 2** .

**Pathological Report:**

**The examined breast tissue shows an invasive carcinoma characterized by infiltrating nests and cords of epithelial cells within a desmoplastic stroma, extending into adjacent adipose tissue** . **The tumor cells exhibit moderate nuclear pleomorphism, visible nucleoli, and limited tubule formation (estimated 10-75%)** . **Mitotic figures are present but not numerous** . **A moderate stromal lymphocytic infiltrate is noted** . **No definite lymphovascular invasion is identified in the sections examined** . **A large ductal structure appears benign or possibly shows low-grade DCIS, distinct from the invasive component** .

**Overall findings are consistent with invasive ductal carcinoma, No Special Type (NOS), Nottingham histological grade 2 (Tubules: 2, Nuclei: 2, Mitoses: 1; Total Score: 5)** .

Figure A14: An example of CPathAgent's navigation path planning and multi-scale, multi-view reasoning process.

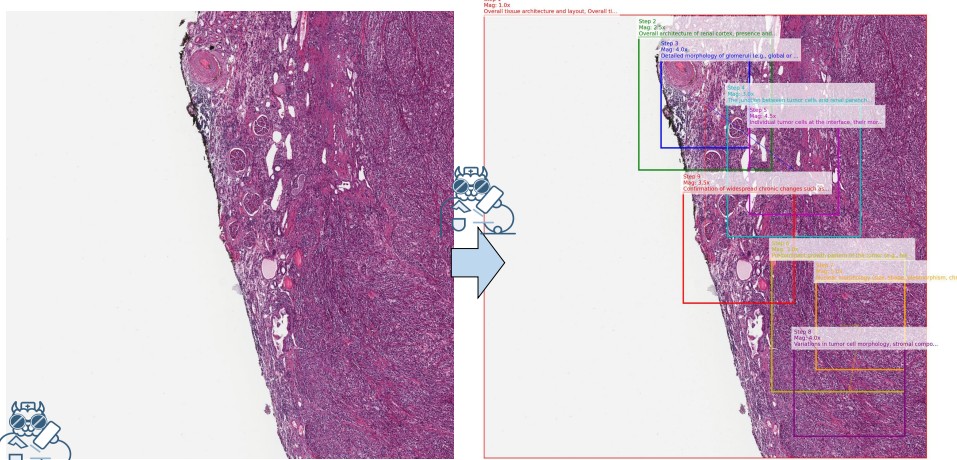

Starting with **an overall view of the entire slide** at `1.0x magnification` ([0, 0, 1, 1]) , I can immediately see **two distinct regions** . On the left, there's **a more organized, somewhat lobulated tissue that appears to be renal parenchyma, characterized by the presence of glomeruli and tubules** . On the right, there's **a much larger, densely cellular, and more homogenous area that looks like a neoplasm** . The interface between these two regions is somewhat **irregular** . My **initial impression is a renal tumor adjacent to non-neoplastic kidney tissue** .

Now, I'm **moving to the upper left quadrant** at `2.5x magnification` ([0.35, 0.05, 0.65, 0.35]) to **examine the non-neoplastic renal parenchyma more closely** . I can identify **several glomeruli, some of which appear globally sclerotic, meaning they are solidified and acellular** . **The tubules are variably sized; some are atrophic with thickened basement membranes, while others are dilated** . **The interstitium is clearly expanded by fibrosis and contains a significant inflammatory infiltrate, predominantly lymphocytic** . There's also **a prominent, thick-walled artery in the upper left of this field, showing features of arteriosclerosis** . These findings are **indicative of chronic kidney disease** .

Let's **increase the magnification to** `4.0x` within this non-neoplastic kidney area ([0.4, 0.1, 0.6, 0.3]) to **get a better look at the cellular details** . **The glomeruli show varying degrees of sclerosis, some with segmental and others global** . **The arterioles exhibit hyaline arteriolosclerosis, with homogenous eosinophilic thickening of their walls** . **The tubules show clear signs of atrophy, with simplified epithelium and thickened basement membranes, alongside some dilated tubules** . **The interstitial fibrosis is extensive, and the chronic inflammatory infiltrate is composed mainly of lymphocytes and some plasma cells** . These features **confirm significant chronic nephropathy** .

Now, I'm **shifting my attention to the interface between the non-neoplastic kidney and the neoplasm** at `3.0x magnification` ([0.55, 0.2, 0.85, 0.5]) . **The tumor, on the right, is composed of densely packed cells with a different morphology from the renal parenchyma on the left** . **The border appears somewhat irregular, suggesting an infiltrative growth pattern rather than a well-defined pushing margin or a thick capsule** . There's **a desmoplastic stromal reaction at the interface, with increased collagen deposition and some inflammatory cells** . I can see **some entrapped, atrophic tubules within the tumor at the periphery** .

**Zooming in further on this tumor-kidney interface** at `4.5x` ([0.6, 0.25, 0.8, 0.45]) , I can **better appreciate the interaction** . **The tumor cells are clearly infiltrating between the residual renal tubules** . These **tumor cells are predominantly spindle-shaped, with elongated, hyperchromatic nuclei and eosinophilic cytoplasm** . They are **arranged in fascicles** . **The adjacent renal tubules show reactive changes, including atrophy and some epithelial simplification** . **The infiltrative nature is quite evident here** .

Next, I'm **moving into the main body of the neoplasm** at `3.0x magnification` ([0.65, 0.55, 0.95, 0.85]) to **assess its overall architecture** . **The tumor is highly cellular and predominantly composed of spindle cells arranged in intersecting fascicles, creating a storiform or herringbone pattern in some areas** . **The stroma is relatively scant in some parts, more collagenous in others** . **Vascularity is present, with small caliber vessels coursing through the tumor** . The overall impression is that of **a sarcomatoid or spindle cell neoplasm** .

Let's go to **a higher magnification** of `5.0x` **within a representative area of the neoplasm** ([0.75, 0.6, 0.95, 0.8]) to **evaluate the cytological features** . **The spindle cells exhibit significant nuclear pleomorphism, with variation in size and shape** . **The nuclei are hyperchromatic, some are vesicular with visible, though not strikingly prominent, nucleoli** . **The cytoplasm is eosinophilic and fibrillar** . **Cell borders are indistinct** . I am **actively looking for mitotic figures** , and I can **identify a few, suggesting proliferative activity** . These features are consistent with **a high-grade malignancy** .

I'll **examine another area within the neoplasm** at `4.0x` ([0.7, 0.75, 0.95, 0.95]) to **assess for any heterogeneity** . The morphology here is **largely consistent with what I've seen before – densely packed spindle cells in fascicles** . However, in this field, I can see **some entrapped, atrophic renal tubules within the tumor, further supporting its infiltrative nature** . **The stroma appears somewhat more collagenous in this particular view** . **The high-grade cytological features persist** .

Finally, I'm **revisiting the non-neoplastic kidney parenchyma in a different location** , **more central/left** , at `3.5x magnification` ([0.45, 0.4, 0.7, 0.65]) to **confirm the consistency of the chronic changes** . Indeed, the findings are **similar to what I observed earlier: widespread glomerulosclerosis, tubular atrophy, interstitial fibrosis, and a chronic inflammatory infiltrate** . **Some glomeruli are globally sclerosed, others show segmental sclerosis** . **The tubules are variably atrophic and dilated** . **The interstitium is fibrotic and inflamed** . These **chronic changes appear to be diffuse throughout the non-neoplastic kidney tissue presented in these sections** .

Based on these observations, the **non-neoplastic kidney shows significant chronic nephropathy, including glomerulosclerosis, tubular atrophy, interstitial fibrosis, and chronic inflammation, along with vascular changes** . The **neoplasm is a high-grade spindle cell sarcoma, characterized by marked nuclear pleomorphism, hyperchromasia, mitotic activity, and an infiltrative growth pattern into the adjacent renal parenchyma** . **The tumor cells are arranged in fascicles** . Given the kidney location, **a sarcomatoid renal cell carcinoma would be a primary consideration, although other spindle cell sarcomas could be in the differential** . The reference to "sarcomatoid features" aligns with my findings of **a high-grade spindle cell malignancy** .

**Pathological Report:**

The kidney tissue examined shows **two distinct components** . **The non-neoplastic renal parenchyma exhibits features of severe chronic nephropathy, characterized by widespread glomerulosclerosis (both global and segmental), tubular atrophy, interstitial fibrosis, and a significant chronic inflammatory infiltrate** . **Vascular changes, including arteriosclerosis and hyaline arteriolosclerosis, are also present** . Adjacent to this, there is **a high-grade malignant neoplasm composed predominantly of spindle cells** . These **neoplastic cells are arranged in intersecting fascicles and display marked nuclear pleomorphism, hyperchromasia, and mitotic activity** . **The tumor demonstrates an infiltrative growth pattern, invading the adjacent renal parenchyma and entrapping atrophic tubules** . The overall features are consistent with **a high-grade spindle cell sarcoma, such as a sarcomatoid renal cell carcinoma, arising in a background of severe chronic kidney disease** .

Figure A15: An example of CPathAgent's navigation path planning and multi-scale, multi-view reasoning process.

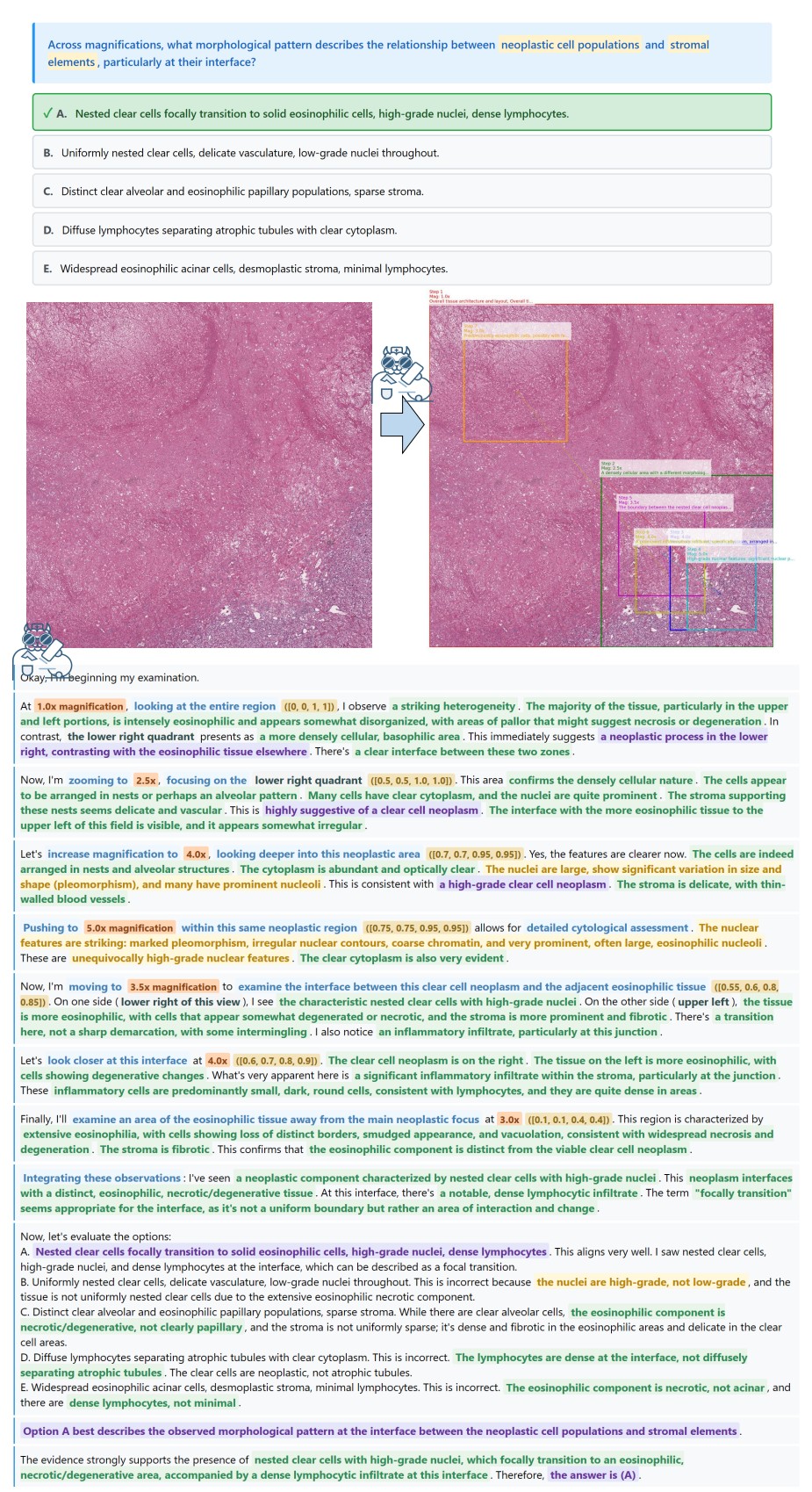

Figure A16: An example of CPathAgent's VQA-oriented navigation path planning and multi-scale, multi-view reasoning process.

### E.5 Prompts

This section presents the prompts used in our dataset construction and experiments.

**Gemini-2.5-Pro Prompts for Dataset Construction:** We design a series of prompts to guide Gemini-2.5-Pro in constructing our pathologist-like planning and reasoning dataset generation pipeline. Figure A17 presents the prompt used to extract region-relevant descriptions from WSI reports, while Figure A18 shows the prompt for region selection using WSI overview images and paired reports. For multi-scale image description generation, Figure A19 displays the prompt that generates detailed descriptions for image patches at different magnifications. Figure A20 contains the prompt for creating navigation plans using coordinate-annotated region overviews and multi-scale descriptions. Finally, Figure A21 shows the prompt that generates pathologist-like diagnostic reasoning based on extracted patches from navigation paths and paired region descriptions.

For VQA dataset construction, we use several specialized prompts: Figure A22 shows the prompt that instructs the model to generate VQA samples, Figure A23 presents the prompt for predicting question answers from text-only inputs (applied to both Gemini-2.5-Pro and DeepSeek-V3), Figure A24 displays the prompt that guides the model to create question-guided navigation plans, and Figure A25 contains the prompt for generating reasoning-based answers.

**CPathAgent Training and Inference Prompts:** The prompts for CPathAgent follow a similar design, while removing the additional information utilized by Gemini-2.5-Pro (such as pathology reports), focusing on vision-only pathological analysis that replicates clinical diagnostic processes. Figures A26, A27, and A28 present the core prompts used for CPathAgent's training and inference. These prompts guide CPathAgent to perform navigation planning and reasoning using only visual input. The VQA-specific prompts shown in Figures A29 and A30 display the prompts that used for CPathAgent's question-guided navigation and reasoning.

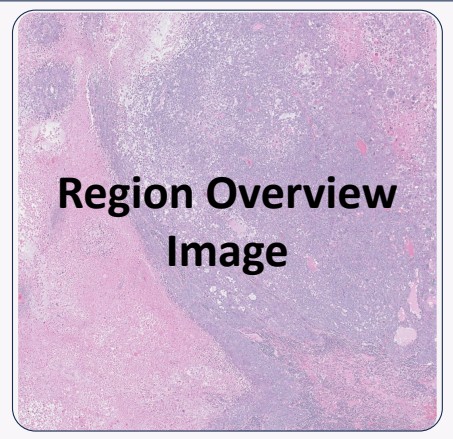
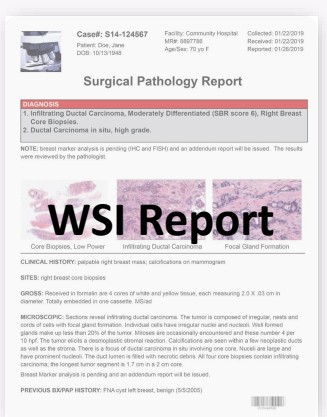

Prompt: This is the report for {**Tissue Type**} whole slide image. What you're seeing now is one specific region from this whole slide image. Please determine if there are any descriptions in the report that match this region.

1. First, carefully examine the visual features present in this specific region of the whole slide image.
2. Next, go through each description in the report one by one.
3. For each description, assess whether it could match the visual features you observe in this region.
4. Consider both positive findings (features that are present) and negative findings (features that are explicitly absent).
5. If a description seems to match, evaluate how complete the match is - does it fully match or only partially?
6. For partial matches, consider whether the description might modification.
7. Identify the most detailed and granular matching descriptions from the report.
8. Evaluate whether this specific region is representative of the overall whole slide image diagnosis and, if so, modify or attach the final diagnosis to better represent the features observed in this region.
9. If no descriptions match at all, prepare to output 'None'.

Please output your final matched description (start with **matched description:**).

Report: {**WSI Report**}

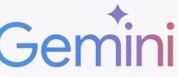

Figure A17: Prompt for Gemini-2.5-Pro to extract description related to a given region from the corresponding WSI paired pathology report.

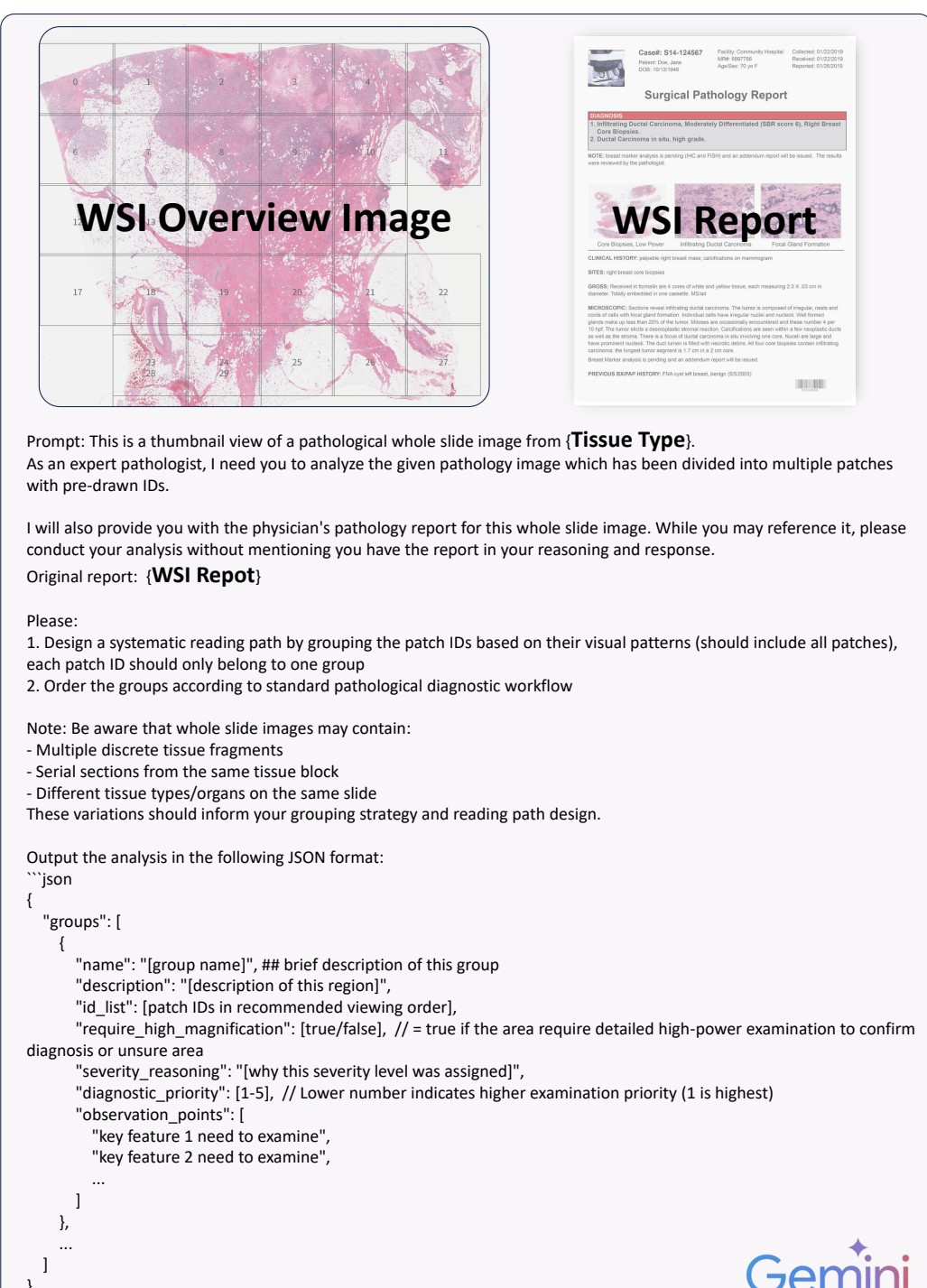

Figure A18: Prompt for Gemini-2.5-Pro to generate region selection results given a WSI overview and its paired pathology report.

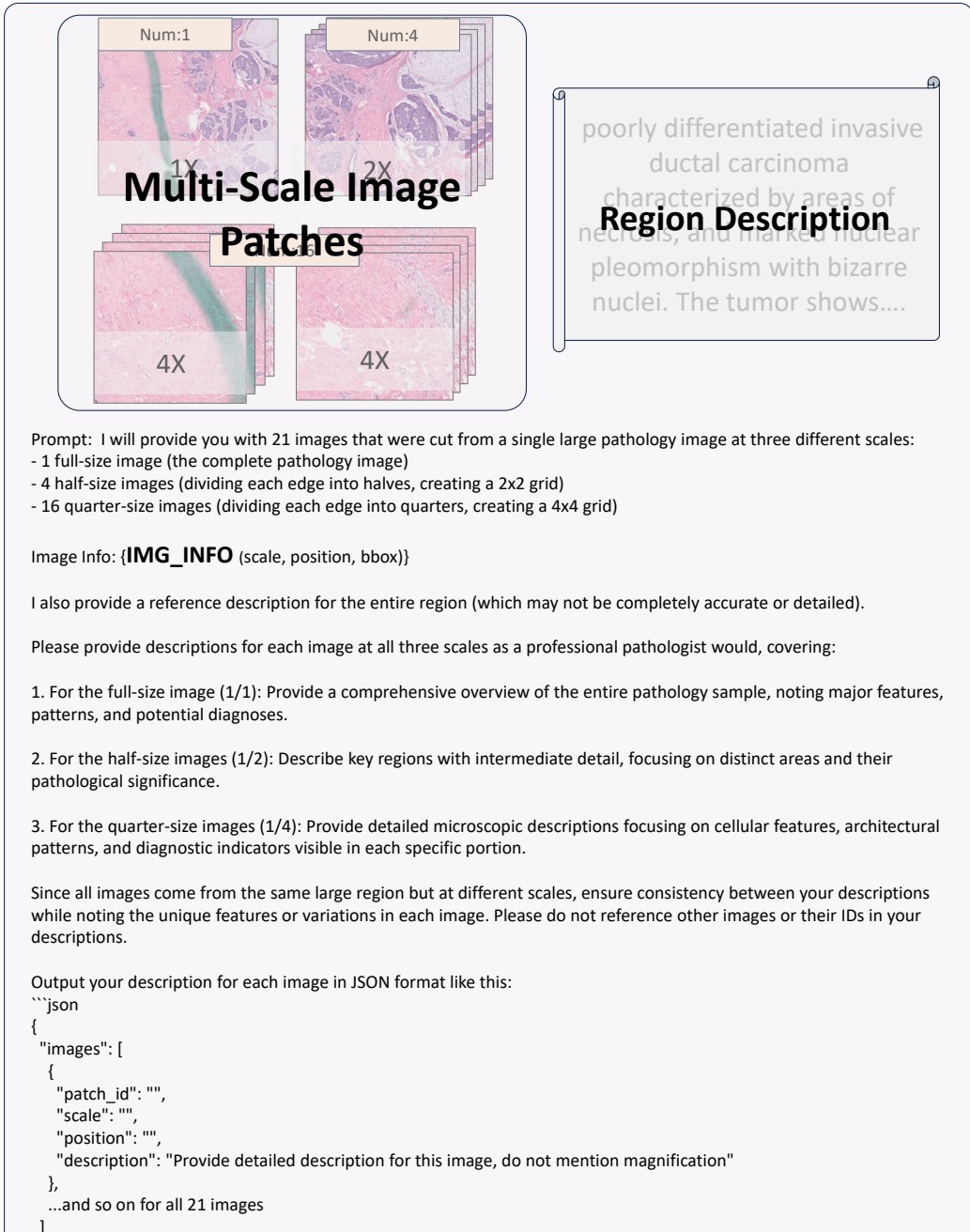

Prompt: I will provide you with 21 images that were cut from a single large pathology image at three different scales:
- 1 full-size image (the complete pathology image)
- 4 half-size images (dividing each edge into halves, creating a 2x2 grid)
- 16 quarter-size images (dividing each edge into quarters, creating a 4x4 grid)

Image Info: {**IMG_INFO** (scale, position, bbox)}

I also provide a reference description for the entire region (which may not be completely accurate or detailed).

Please provide descriptions for each image at all three scales as a professional pathologist would, covering:

1. For the full-size image (1/1): Provide a comprehensive overview of the entire pathology sample, noting major features, patterns, and potential diagnoses.

2. For the half-size images (1/2): Describe key regions with intermediate detail, focusing on distinct areas and their pathological significance.

3. For the quarter-size images (1/4): Provide detailed microscopic descriptions focusing on cellular features, architectural patterns, and diagnostic indicators visible in each specific portion.

Since all images come from the same large region but at different scales, ensure consistency between your descriptions while noting the unique features or variations in each image. Please do not reference other images or their IDs in your descriptions.

Output your description for each image in JSON format like this:
```json
{
 "images": [
  {
    "patch_id": "",
    "scale": "",
    "position": "",
    "description": "Provide detailed description for this image, do not mention magnification"
  },
  ...and so on for all 21 images
 ]
}
```
Reference description: {Region Description}

Figure A19: Prompt for Gemini-2.5-Pro to generate descriptions for 1 image at 1X scale, 4 images at 2X scale, and 16 images at 4X scale, given these images and their paired region descriptions.

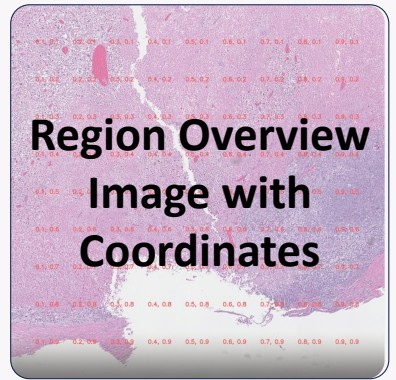

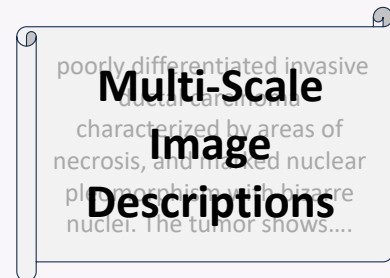

Prompt: Please carefully examine this pathology image and simulate a pathologist's viewing path. For areas showing potential pathological changes that warrant careful observation, please analyze using these guidelines:

You can zoom in and out to observe the image. In the default state (1.0x), you see the complete image at 1/5 of its original resolution.
For any zoom level Z:  - Field of view: $1/Z^2$ of the total image area (1/Z of each side length)   - Downscale ratio: 5/Z
Examples:
- 2.0x: Shows 1/4 area (downscaled 2.5x)          - 2.5x: Shows 1/6.25 area (downscaled 2x)
- 3.0x: Shows 1/9 area (downscaled 1.67x)          - 3.5x: Shows 1/12.25 area (downscaled 1.43x)
- 4.5x: Shows 1/20.25 area (downscaled 1.11x)      - 5.0x: Shows 1/25 area (downscaled 1x)
You can use any zoom level up to 5.0x, including decimal values (such as 2.3x or 4.5x). Image coordinates are represented in relative values ranging from 0 to 1.

**Reference Image Captions**
- I will provide reference images captions at different magnifications:
  - 1.0x (full image)    - 2.0x (4 sections)    - 4.0x (16 sections)
  - These are for reference only - your viewing path should:
  - Not exactly match these reference images' coordinate
  - Consider other magnification levels
  - Create regions that may span across my provided regions

Your viewing path steps should:
- Begin with low-power scanning to identify key histopathological patterns and tissue architecture
- Check and avoid empty/background/unimportant areas that lack diagnostic value
- Focus on diagnostic hotspots (areas showing pathological changes, abnormal cell populations, or architectural distortion)
- Pay special attention to transition zones between normal and abnormal tissue
- Selectively analyze areas with representative features using a diagnostic hierarchy (primary lesions before secondary changes)
- Apply systematic differential diagnostic reasoning when examining cellular and tissue features
- Ensure minimal overlap between sampled patches to reduce redundancy, except when:
  * Examining the same region at different magnifications is clinically necessary
  * Comparing similar features across multiple sites is relevant for diagnosis
- Demonstrate logical continuity between each step that reflects pathological reasoning
- Movements between steps should follow a systematic pattern
- Prioritize fields that demonstrate key diagnostic criteria rather than simply sliding the window
- Consider the blue coordinate grid for position reference
- Only Zoom in/Zoom out when necessary

Please provide the viewing path as a JSON list:
```json
[
  {
    "step": [step number],
    "magnification": [zoom level from 2.0x-5.0x],
    "region_coordinate": [x1, y1, x2, y2], #  Coordinates represent [top_left_x, top_left_y, bottom_right_x, bottom_right_y]
"reasoning": [reasoning for moving to this position],
    "need_to_see": [description of what should be observed at this position],
  },
  ...
]
```

Reference image captions: {**Multi-Scale Image Descriptions**}'''

Gemini

Figure A20: Prompt for Gemini-2.5-Pro to generate navigation path planning given the region overview with coordinates and 21 multi-scale image descriptions generated in the previous step.

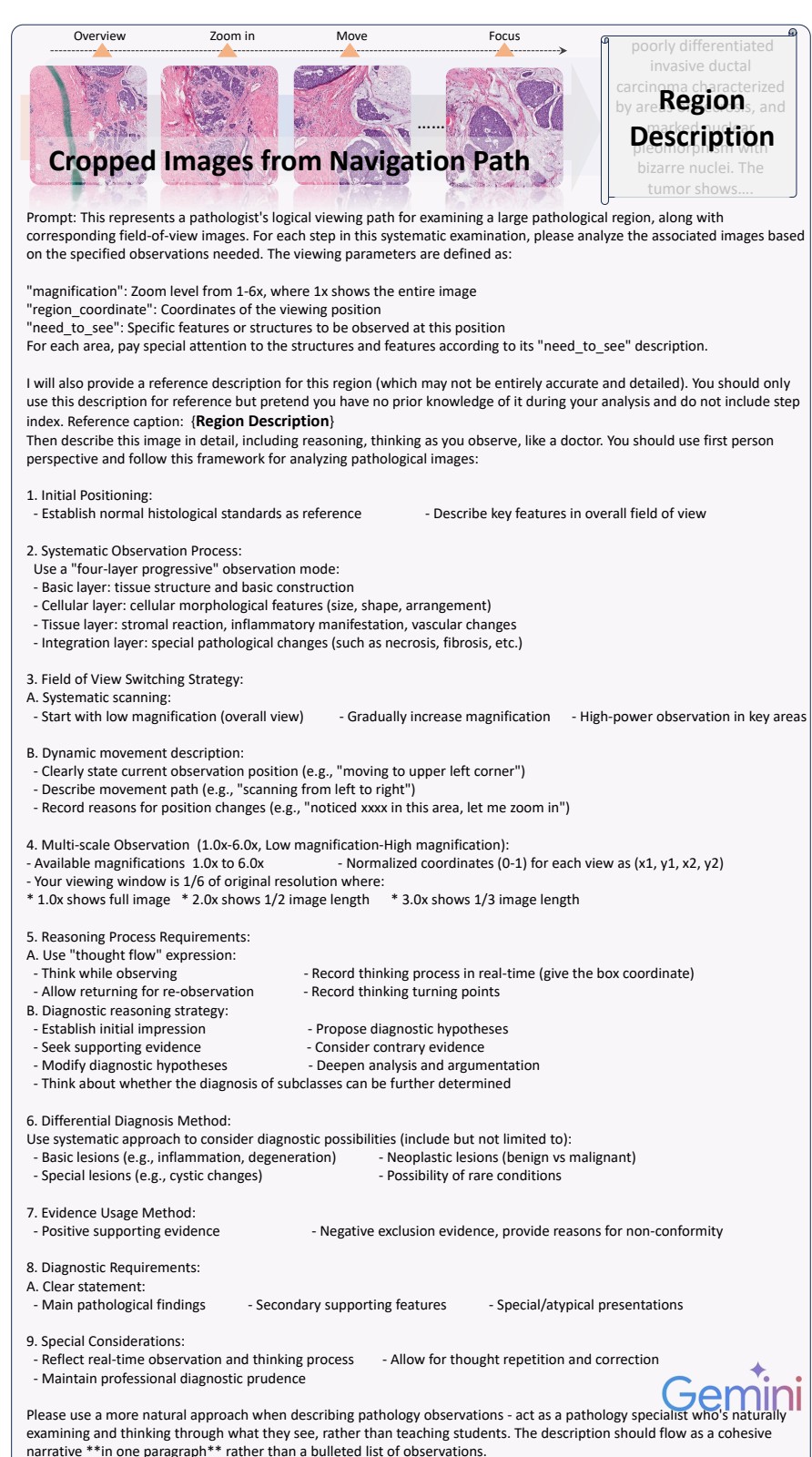

Figure A21: Prompt for Gemini-2.5-Pro to generate multi-scale multi-image reasoning for description and diagnosis based on the planned navigation path, given the cropped images along the navigation path and the overall region description extracted from the previous step.

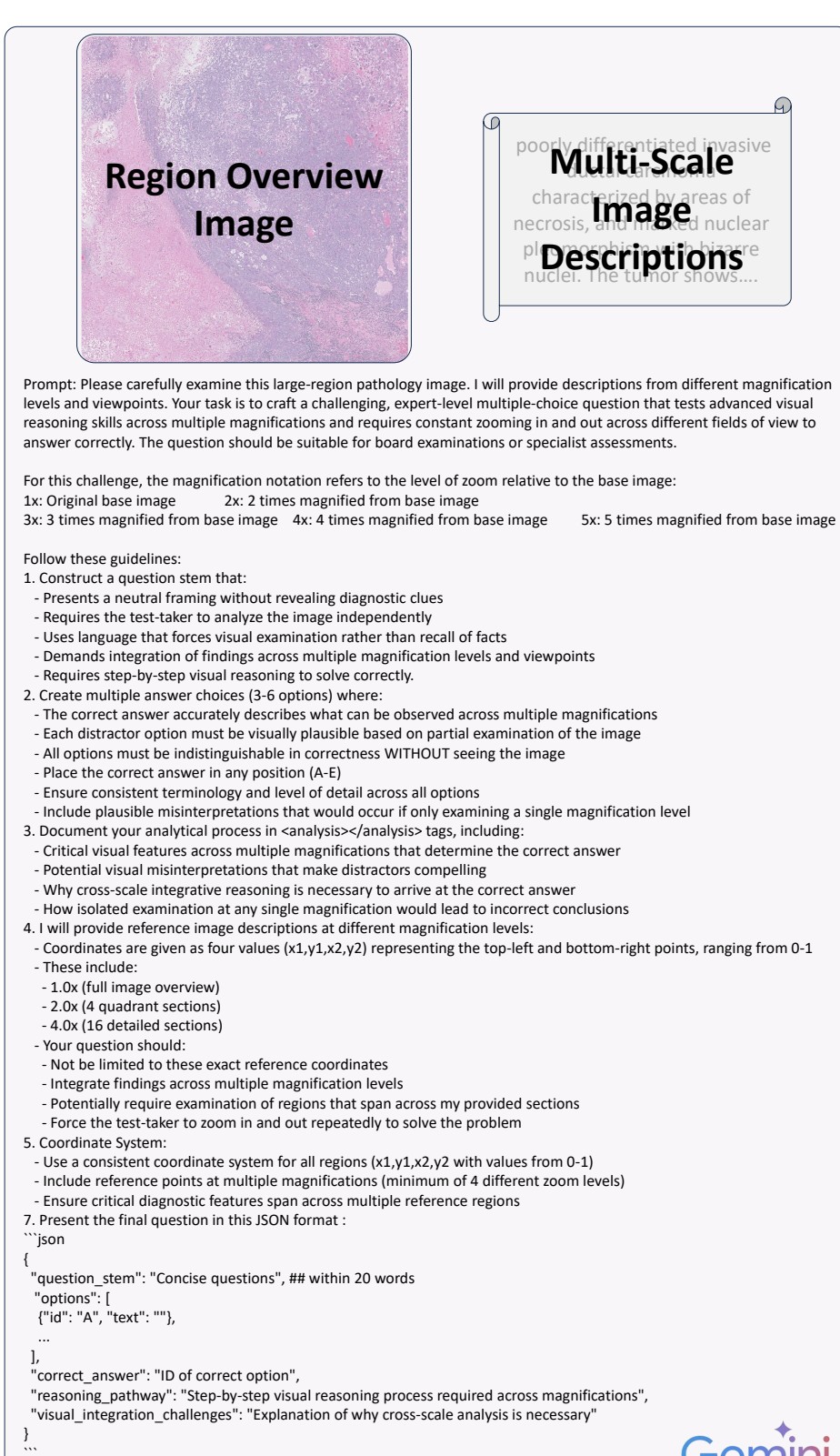

Figure A22: Prompt for Gemini-2.5-Pro to generate VQA samples given the region overview and the 21 multi-scale image descriptions generated from the previous step.

This question was originally meant to be paired with an image. No image is provided, so please answer based on the text alone.

{**Question**}

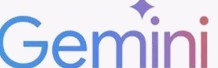

Think step by step.
You must **ensure** that your answer ends with "The answer is X", where X is your final answer index.

Figure A23: Prompt for Gemini-2.5-Pro to conduct educated guesses given only the text portion of the generated VQA samples.

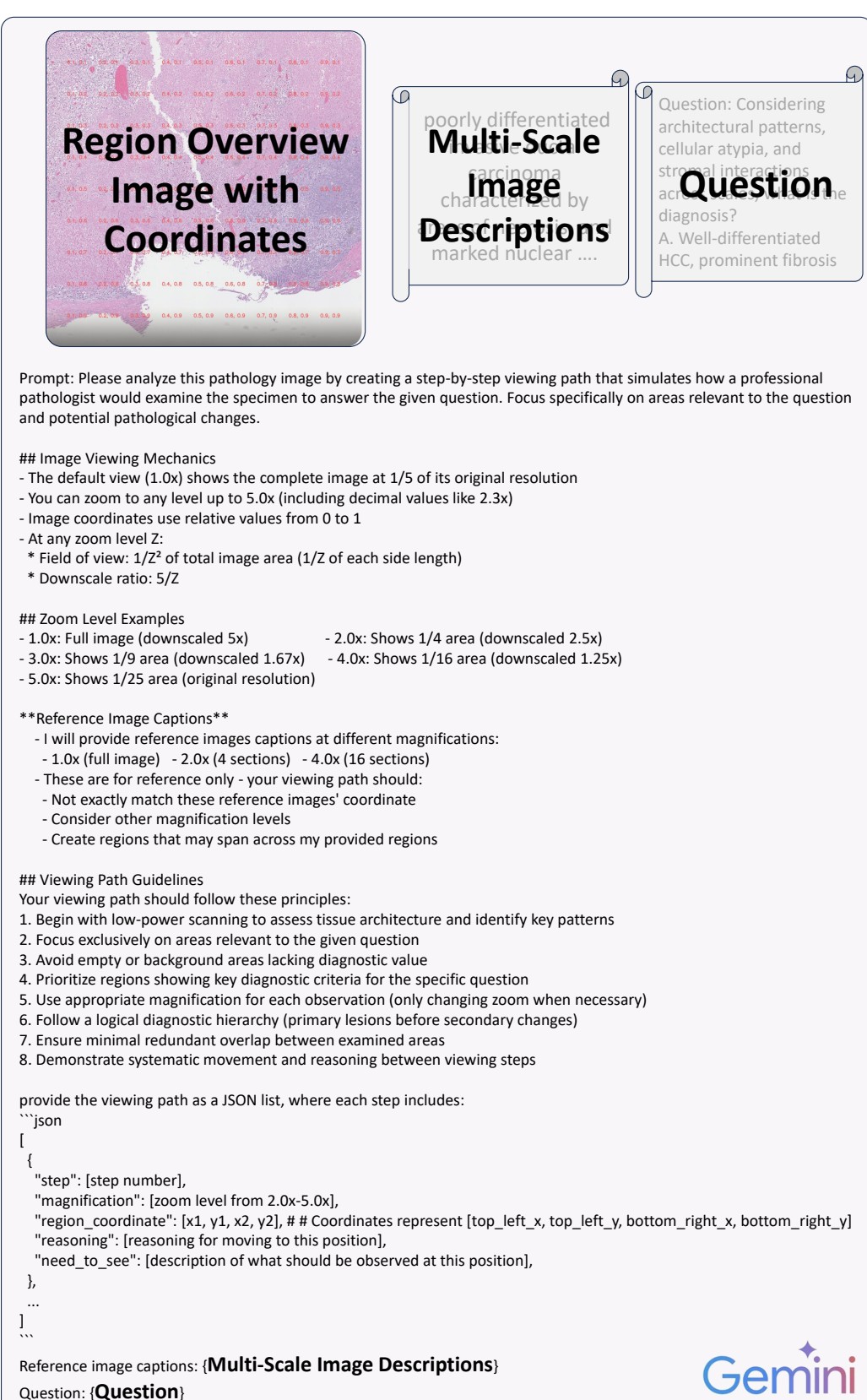

Figure A24: Prompt for Gemini-2.5-Pro to generate navigation path planning given the region overview with coordinates, 21 multi-scale image descriptions and the question generated in the previous step.

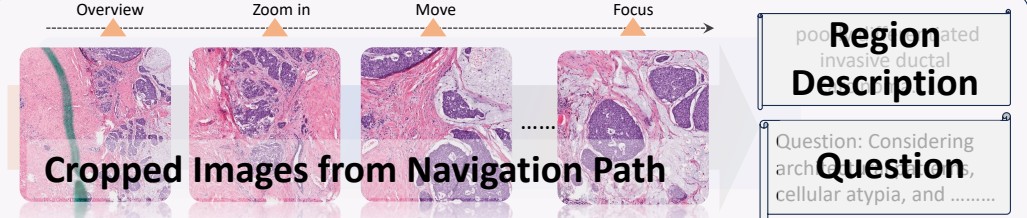

**Cropped Images from Navigation Path**

**Region Description**

**Question**

Prompt: You are a pathology specialist , you are examining images from a pathologist's viewing path through a pathological region. Based on these images, you need to provide detailed reasoning to answer a specific question or select the most appropriate option from multiple choices.

For each image in the viewing path, you will receive:
- Magnification: Zoom level from 1.0x to 5.0x (where 1.0x shows the entire image)
- Region coordinates: Position you're viewing within the image (normalized from 0-1)
- Need to see: Specific features or structures to observe at this position

I will provide reference descriptions for overall region caption. Use these descriptions only as reference - pretend you don't have prior knowledge of them when conducting your analysis.

Region captions: {**Region description**}

When analyzing the images, follow this structured approach:
1. Observational Evidence
  - Document key findings visible in each image relevant to the question
  - Be specific about which image/region shows each finding
  - Describe cellular morphology, tissue architecture, and special features

2. Evidence Integration
  - Connect and compare findings across different images in the viewing path

3. Option Analysis (for multiple-choice questions)
  - Evaluate how the observed evidence supports or contradicts each option
  - Explain why incorrect options don't align with the visual evidence
  - Justify why the best option is supported by your findings

4. Question-Specific Reasoning
  - Apply pathological principles directly to the question
  - Explain how specific visual clues lead to your conclusion
  - Address any ambiguities or limitations in the visible evidence

5. Multi-scale Observation  (1.0x-5.0x, Low magnification-High magnification):
- Consider how features appear across different magnifications (1.0x-5.0x)
- Note how your viewing window represents portions of the original image at different zoom levels
- Your viewing window is 1/5 of original resolution where:
* 1.0x shows full image
* 2.0x shows 1/2 image length
* 3.0x shows 1/3 image length

6. Reasoning Process Requirements:
 Use "thought flow" expression:
  - Think while observing
  - Record thinking process in real-time (give the box coordinate)
  - Allow returning for re-observation
  - Record thinking turning points

Present your analysis as a natural, flowing narrative from a first-person perspective. Think and reason as you observe, like an experienced pathologist examining the images in real-time. Record your thought process, including any moments when you need to revisit certain images or when your thinking changes based on new observations. The description should flow as a cohesive narrative **several paragraphs** rather than a bulleted list of observations.

Conclude with a brief one-sentence summary of your answer, followed by "Therefore, the answer is (X).

Question: {**Question**}

Figure A25: Prompt for Gemini-2.5-Pro to generate VQA-oriented reasoning based on the planned navigation path, given the cropped images along the navigation path, the overall region description extracted from the previous step, and the target question.

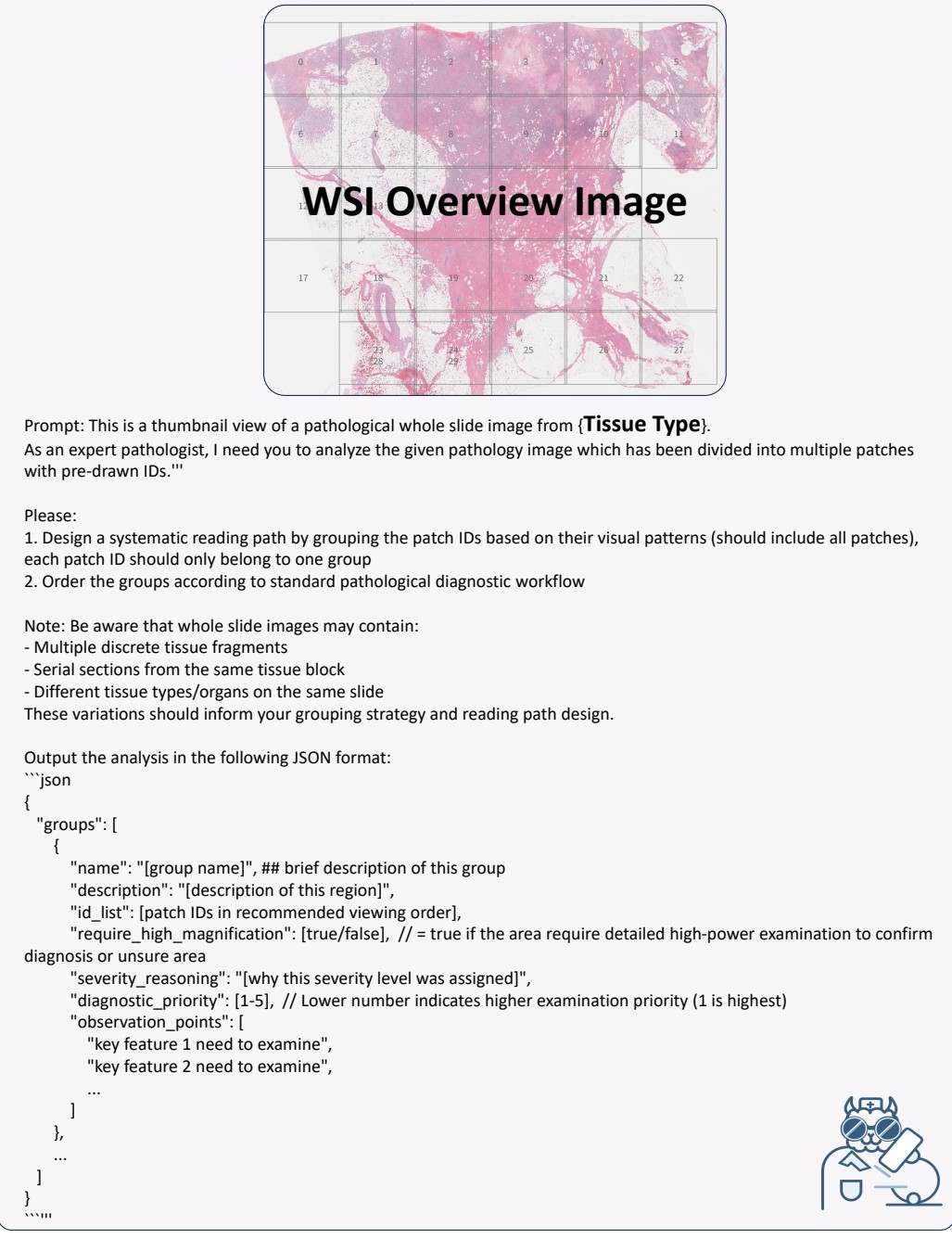

Prompt: This is a thumbnail view of a pathological whole slide image from {**Tissue Type**}.
As an expert pathologist, I need you to analyze the given pathology image which has been divided into multiple patches with pre-drawn IDs.'''

Please:
1. Design a systematic reading path by grouping the patch IDs based on their visual patterns (should include all patches), each patch ID should only belong to one group
2. Order the groups according to standard pathological diagnostic workflow

Note: Be aware that whole slide images may contain:
- Multiple discrete tissue fragments
- Serial sections from the same tissue block
- Different tissue types/organs on the same slide
These variations should inform your grouping strategy and reading path design.

Output the analysis in the following JSON format:
```json
{
  "groups": [
    {
      "name": "[group name]", ## brief description of this group
      "description": "[description of this region]",
      "id_list": [patch IDs in recommended viewing order],
      "require_high_magnification": [true/false],  // = true if the area require detailed high-power examination to confirm diagnosis or unsure area
      "severity_reasoning": "[why this severity level was assigned]",
      "diagnostic_priority": [1-5],  // Lower number indicates higher examination priority (1 is highest)
      "observation_points": [
        "key feature 1 need to examine",
        "key feature 2 need to examine",
        ...
      ]
    },
    ...
  ]
}
```"""

Figure A26: Prompt for CPathAgent to generate region selection results given a WSI overview.

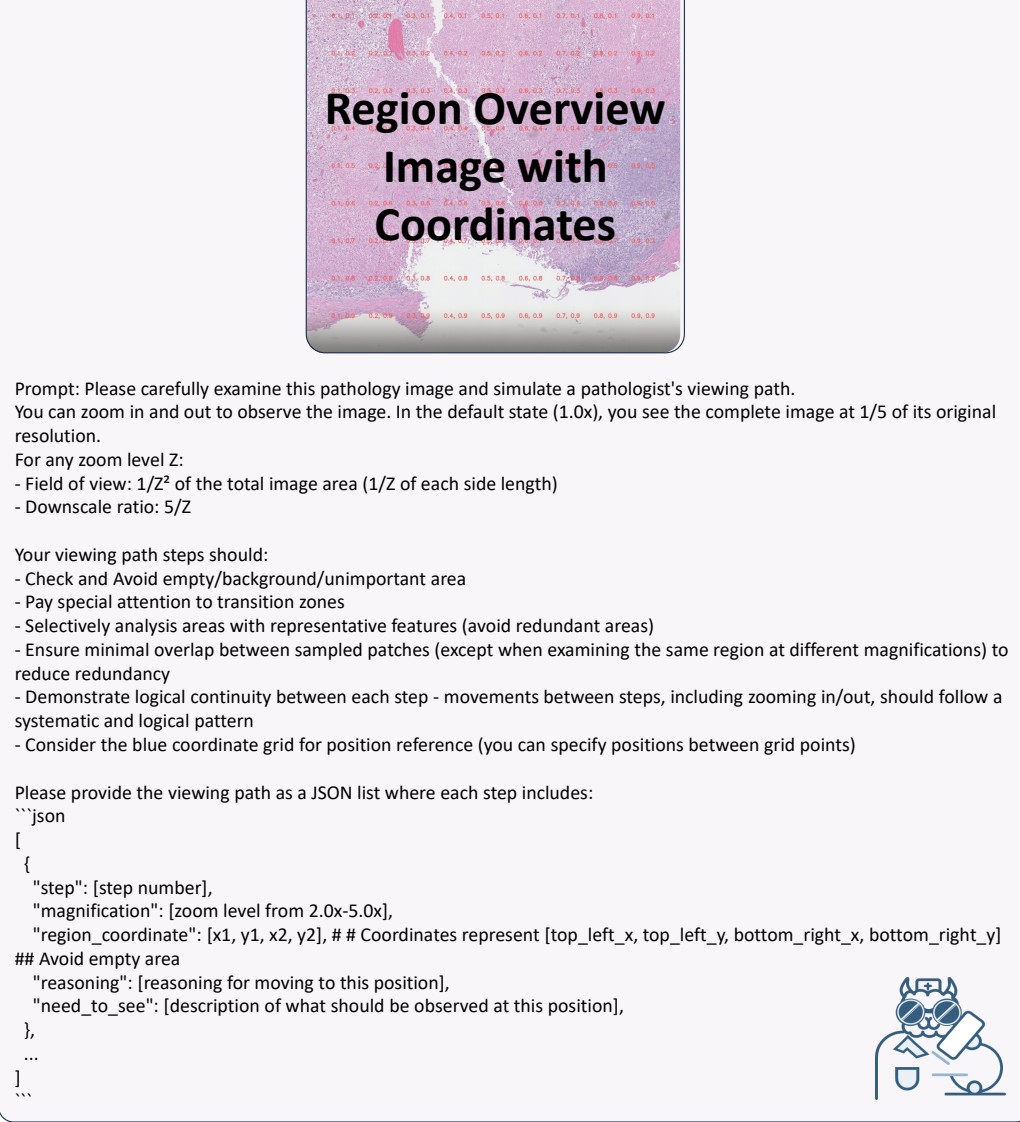

**Region Overview Image with Coordinates**

Prompt: Please carefully examine this pathology image and simulate a pathologist's viewing path.
You can zoom in and out to observe the image. In the default state (1.0x), you see the complete image at 1/5 of its original resolution.
For any zoom level Z:
- Field of view: $1/Z^2$ of the total image area (1/Z of each side length)
- Downscale ratio: 5/Z

Your viewing path steps should:
- Check and Avoid empty/background/unimportant area
- Pay special attention to transition zones
- Selectively analysis areas with representative features (avoid redundant areas)
- Ensure minimal overlap between sampled patches (except when examining the same region at different magnifications) to reduce redundancy
- Demonstrate logical continuity between each step - movements between steps, including zooming in/out, should follow a systematic and logical pattern
- Consider the blue coordinate grid for position reference (you can specify positions between grid points)

Please provide the viewing path as a JSON list where each step includes:
```json
[
 {
   "step": [step number],
   "magnification": [zoom level from 2.0x-5.0x],
   "region_coordinate": [x1, y1, x2, y2], # # Coordinates represent [top_left_x, top_left_y, bottom_right_x, bottom_right_y]
## Avoid empty area
   "reasoning": [reasoning for moving to this position],
   "need_to_see": [description of what should be observed at this position],
 },
 ...
]
```

Figure A27: Prompt for CPathAgent to generate navigation path planning given the region overview with coordinates.

Overview        Zoom in        Move        Focus

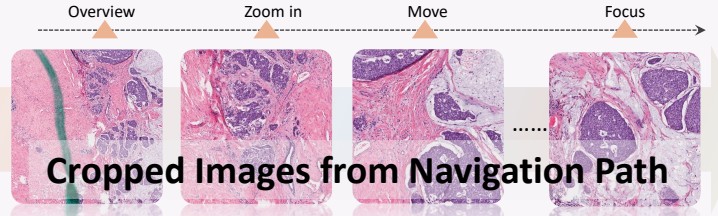

**Cropped Images from Navigation Path**

Prompt: This represents a pathologist's logical viewing path for examining a large pathological region, along with corresponding field-of-view images. For each step in this systematic examination, please analyze the associated images based on the specified observations needed. The viewing parameters are defined as:

"magnification": Zoom level from 1-5x, where 1x shows the entire image
"region_coordinate": Coordinates of the viewing position
"need_to_see": Specific features or structures to be observed at this position

For each area, pay special attention to the structures and features according to its "need_to_see" description.

Then describe this image in detail, including reasoning, thinking as you observe, like a doctor. You should use first person perspective and follow this framework for analyzing pathological images:

1. Multi-scale Observation  (1.0x-6.0x, Low magnification-High magnification):
- Available magnifications  1.0x to 6.0x
- Normalized coordinates (0-1) for each view as (x1, y1, x2, y2)
- Your viewing window is 1/6 of original resolution where:
* 1.0x shows full image
* 2.0x shows 1/2 image length
* 3.0x shows 1/3 image length

2. Reasoning Process Requirements:
A. Use "thought flow" expression:
  - Think while observing
  - Record thinking process in real-time (give the box coordinate)
  - Allow returning for re-observation
  - Record thinking turning points

B. Diagnostic reasoning strategy:
  - Establish initial impression
  - Propose diagnostic hypotheses
  - Seek supporting evidence
  - Consider contrary evidence
  - Modify diagnostic hypotheses
  - Deepen analysis and argumentation
  - Think about whether the diagnosis of subclasses can be further determined

3. Evidence Usage Method:
  - Positive supporting evidence
  - Negative exclusion evidence, provide reasons for non-conformity

4. Special Considerations:
  - Reflect real-time observation and thinking process
  - Allow for thought repetition and correction
  - Maintain professional diagnostic prudence

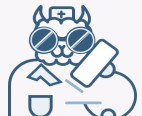

Finally, please output an additional breif summary of this region start with **Pathological Report:** in one paragraph.

Figure A28: Prompt for CPathAgent to generate multi-scale multi-image reasoning for description and diagnosis based on the planned navigation path, given the cropped images along the navigation path.

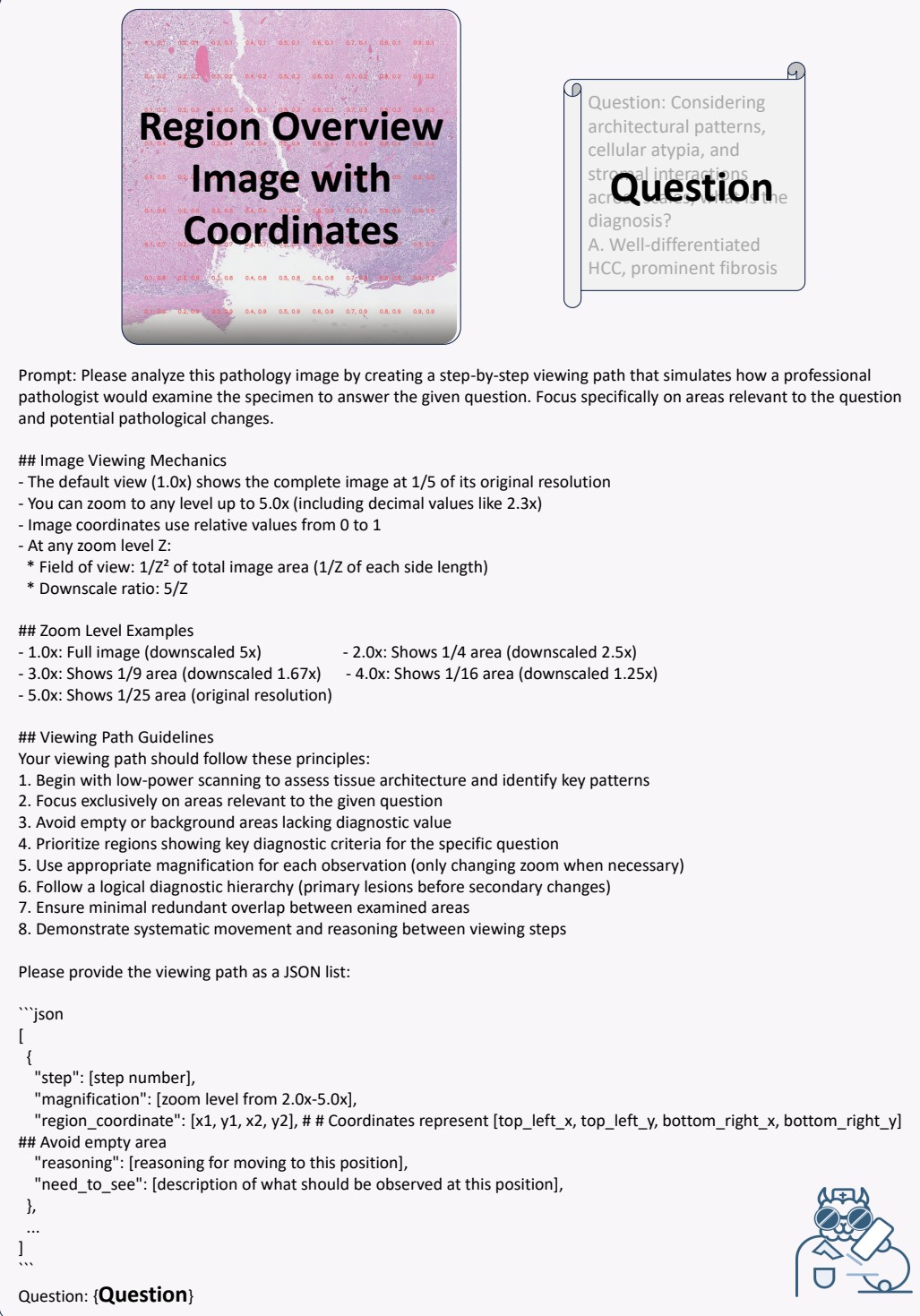

Figure A29: Prompt for CPathAgent to generate VQA-oriented navigation path planning given the region overview with coordinates and the question.

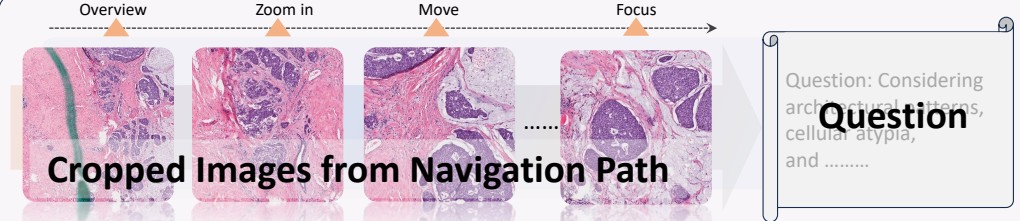

Prompt: You are a pathology specialist , you are examining images from a pathologist's viewing path through a pathological region. Based on these images, you need to provide detailed reasoning to answer a specific question or select the most appropriate option from multiple choices.

For each image in the viewing path, you will receive:
- Magnification: Zoom level from 1.0x to 5.0x (where 1.0x shows the entire image)
- Region coordinates: Position you're viewing within the image (normalized from 0-1)
-   Need to see: Specific features or structures to observe at this position

When analyzing the images, follow this structured approach:
1. Observational Evidence
   - Document key findings visible in each image relevant to the question
   - Be specific about which image/region shows each finding
   - Describe cellular morphology, tissue architecture, and special features

2. Evidence Integration
   - Connect and compare findings across different images in the viewing path

3. Option Analysis (for multiple-choice questions)
   - Evaluate how the observed evidence supports or contradicts each option
   - Explain why incorrect options don't align with the visual evidence
   - Justify why the best option is supported by your findings

4. Question-Specific Reasoning
   - Apply pathological principles directly to the question
   - Explain how specific visual clues lead to your conclusion
   - Address any ambiguities or limitations in the visible evidence

5. Multi-scale Observation  (1.0x-5.0x, Low magnification-High magnification):
- Consider how features appear across different magnifications (1.0x-5.0x)
- Note how your viewing window represents portions of the original image at different zoom levels
- Your viewing window is 1/5 of original resolution where:
* 1.0x shows full image
* 2.0x shows 1/2 image length
* 3.0x shows 1/3 image length

6. Reasoning Process Requirements:
 Use "thought flow" expression:
  - Think while observing
  - Record thinking process in real-time (give the box coordinate)
  - Allow returning for re-observation
  - Record thinking turning points

Present your analysis as a natural, flowing narrative from a first-person perspective. Think and reason as you observe, like an experienced pathologist examining the images in real-time. Record your thought process, including any moments when you need to revisit certain images or when your thinking changes based on new observations. The description should flow as a cohesive narrative **several paragraphs** rather than a bulleted list of observations.

Conclude with a brief one-sentence summary of your answer, followed by "Therefore, the answer is (X).

Question: {**Question**}

Figure A30: Prompt for CPathAgent to generate VQA-oriented reasoning based on the planned navigation path, given the cropped images along the navigation path and the target question.

