# OpenReview forum: "CPathAgent: An Agent-based Foundation Model for Interpretable High-Resolution Pathology Image Analysis Mimicking Pathologists' Diagnostic Logic"
_NeurIPS.cc/2025/Conference — NeurIPS 2025 poster_

### Official Review · Reviewer_cWNz · 2025-06-26

**Clarity:** 3
**Significance:** 3
**Originality:** 3
**Rating:** 4
**Confidence:** 5

**Summary:**

This paper proposes a multimodal large language model (MLLM) for pathological image huge regions, with a design approach that shows certain innovation. The authors identify the limitations of existing methods in pathological image analysis: patch-based methods are not clinically applicable, while WSI-based methods exceed the processing capabilities of MLLMs. Therefore, designing huge regions that fall between these two approaches is a meaningful attempt.

**Questions:**

## 1. Instruction Dataset Construction
Figure 2 shows the instruction dataset construction process, where the key step is generating patch descriptions for each specific patch based on huge region descriptions. This process is reasonable under zoom-in operations, as each patch remains clear for the Gemini model.

However, the paper does not clearly specify the specific input format during training: **is it only using huge region + final summary, or huge region + patch + patch description + summary, or other combinations?**

## 2. Model Design
The authors adopted the LLaVA-OV model, which can only accept 1008×1008 resolution input. Therefore, the 16000×16000 huge region is downsampled, which raises the "pseudo zoom-in" problem: the model performs zoom-in operations on 1008×1008 low-resolution images, which is actually not true zoom-in. Not operating on huge regions is inconsistent with the operators used when instructions construction. The reasonable approach would be to obtain zoom-in coordinates, extract corresponding regions from the huge region, and then input them to the model, rather than operating on downsampled images.

## 3. Experimental Setup and Result Analysis
### i) PathMMU Experimental Result:
- **Lack of detailed experimental setup descriptions**, and corresponding PathMMU experimental details are not found in the supplementary materials. CPathAgent has already exceeded Expert performance, and it needs to be clearly stated whether it was trained on the PathMMU training set. If it was indeed trained on the training set, the results are acceptable, but this is unfair to other comparison models.
-  The authors briefly state that performance gains come from "multi-scale and reasoning-focused data" (line 258), but lack corresponding ablation experiments to support this claim. **Are the performance gains from patch and patch descriptions (as shown in Figure 2) or from reasoning-focused instruction data?**  Please provide detailed analysis and experiments, as this question has important guiding significance for subsequent research.
### ii) WSI Classification Result:
- Lack of training method descriptions for WSI classification tasks, and relevant experimental details are not found in supplementary materials C.2. The MIL method results are conventional, but the evaluation method for MLLMs is unclear. **Need to clearly specify key information such as input image, resolution and prompt design**.
### iii) PathMMU-HR² Benchmark:
- This is a super-resolution evaluation dataset, but CPathAgent cannot directly process super-resolution images, and input images will be resized to 1008×1008. Under these circumstances, **do the evaluation dataset images only need to be 1008×1008? **Since CPathAgent achieves 88.6 performance at 1008×1008 resolution. **Can I consider this not strictly a huge region benchmark?**

- **The results showing CPathAgent far exceeding Gemini-2.5-Pro need detailed analysis**, which would bring greater value to readers than simply presenting data

## Comment
(To authors) I greatly appreciate the amount of work involved and am very interested in the entire work. According to the review principles, I have carefully completed the review and submitted key questions, and I look forward to the authors' response. I will further consider the scoring based on the authors' reply. (To AC) If needed, I am also willing to discuss with the AC.

**Ethical Concerns:**

["NO or VERY MINOR ethics concerns only"]

**Limitations:**

See above questions.

**Paper Formatting Concerns:**

No formatting issues.

**Quality:**

4

**Strengths And Weaknesses:**

The paper involves multiple core components including instruction dataset construction, model design, VQA benchmark, and experiments proven, but the introduction of each part is not clear enough. Although considering the limitations of the main text length, this causes difficulties for readers to understand. After repeatedly reading the main text and supplementary materials, I still have many confusions. See the question for details.

---

> ### Author Rebuttal · Authors · 2025-07-31
>
> We sincerely appreciate your comprehensive review and constructive feedback, and we are truly grateful for your expressed interest in our work. However, we believe there may be some misunderstandings regarding our methodology and experimental setup. We will address each concern below to provide clarification:
>
> > ### The paper does not clearly specify the specific input format during training:
>
> Thank you for this clarification request. Our training input format for huge regions follows a two-part data structure:
>
> **Part 1 - Navigation Planning:**
>
> - **Input**: Huge region image (16,000×16,000 downsampled to 1,008×1,008)
> - **Output**: JSON-formatted navigation path with magnification levels and coordinates
>
> Example:
>
> ```json
> [{
>   "step": 1,
>   "magnification": 2.5,
>   "region_coordinate": [0.4, 0.4, 0.9, 0.9],
>   "reasoning": "Initial scan at medium power to get...",
>   "need_to_see": "General tissue architecture..."
> },
> {
>   "step": 2,
>   ....
> },
> .....]
> ```
>
> **Part 2 - Multi-scale Sequential Reasoning:**
>
> - **Input**: Sequential images [Image1@2.5x, Image2@3.5x, Image3@5.0x, ...] with their metadata
> - **Output**: Integrated diagnostic reasoning across all views
>
> Example output:
>
> > "Starting with the overall view at 2.5x ([0.4, 0.4, 0.9, 0.9]): The left border shows normal epithelium transitioning to dysplastic tissue. Moving to 3.5x ([0.6, 0.6, 0.95, 0.95]) at the transition zone, I observe ..."
>
> **Both parts are trained together.** This enables CPathAgent to learn both **strategic planning** (where to examine) and **diagnostic reasoning** (interpreting sequence observations). The model **never receives pre-generated patch descriptions during training** , it learns to generate all reasoning directly from visual inputs. Examples are provided in Appendix Figures D.5-D.7.
>
>
> > ### The authors adopted the LLaVA-OV model, which can only accept 1008×1008 resolution input. Therefore, the 16000×16000 huge region is downsampled, which raises the "pseudo zoom-in" problem: the model performs zoom-in on 1008×1008 low-resolution images, which is actually not true zoom-in. The reasonable approach would be to obtain zoom-in coordinates, extract corresponding regions from the huge region, and then input them to the model, rather than operating on downsampled images.
>
> Thank you for raising this concern.  **We do NOT operate on downsampled images for actual analysis.** Instead, we implement exactly as you mentioned '**obtain zoom-in coordinates, extract corresponding regions from the huge region, and then input them to the model**'
>
> ##### Our Actual Workflow
>
> **1. Planning Phase:** The 16,000×16,000 huge region is downsampled to 1,008×1,008 solely for navigation planning.
>
> **2. Execution Phase:** Using the planned coordinates, we return to the **ORIGINAL 16,000×16,000 huge region** and crop patches at native resolution. These high-resolution patches are then provided as sequential multi-image inputs to the model.
>
> **Example:**
>
> - Planning output: "Examine region [0.7, 0.7, 0.95, 0.95] at 3.5x magnification"
> - Execution: Extract pixels [11,200, 11,200, 15,200, 15,200] from the **original 16,000×16,000 image**
>
> This  ensures that while navigation planning uses a downsampled overview for efficiency, all diagnostic analysis operates on **native-resolution patches cropped directly from the original resolution regions**.
>
>
>
> > ### Lack of detailed experimental setup descriptions, and corresponding PathMMU experimental details are not found in the supplementary materials. CPathAgent has already exceeded Expert performance, and it needs to be clearly stated whether it was trained on the PathMMU training set.
>
> Thank you for raising this important point. We clarify that **CPathAgent was only evaluated zero-shot on PathMMU**.
>
> **Why CPathAgent exceeds expert performance:**
>
> 1. **Domain specialization gap**: Individual pathologists typically specialize in 1-2 disease areas, while PathMMU spans the entire pathology spectrum. This creates an inherent disadvantage for human experts on out-of-specialty cases, whereas LMMs can learn across all disease types.
> 2. **Known LMM  shortcuts**: The PathMMU authors found  their benchmark contains text-based shortcuts that LMMs can exploit. To address this, they created PathMMU-Pro with more rigorous data cleaning.
>
> To verify this, we conduct **additional experiments on PathMMU-Pro:**
>
> |Model|Overall Tiny|Overall All|PubMed Tiny|PubMed All|SocialPath Tiny|SocialPath All|EduContent Tiny|EduContent All|
> |-|-|-|-|-|-|-|-|-|
> |Human|69.4|-|71.2|-|70.1|-|66.9|-|
> |CPath-Omni|62.1|61.8|60.5|60.8|58.8|62.1|66.7|63.0|
> |CPathAgent|66.3|65.3|64.6|65.8|65.4|63.6|68.7|66.2|
>
> On this cleaner benchmark, CPathAgent **no longer exceeds human performance** but  still significantly outperforms CPath-Omni.
>
>
> > ### Are the performance gains from patch and patch descriptions (as shown in Figure 2) or from reasoning-focused instruction data? Please provide detailed analysis and experiments.
>
> Thank you for this insightful question. The performance gains come from reasoning-focused instruction data, not patch descriptions.
>
> **In fact, the comparison between CPathAgent and CPath-Omni inherently serves as the ablation experiments**. As both models share identical Stage 1 & Stage 2 (patch-level perception) training data. They differ **only** in Stage 3:
>
> - **CPath-Omni**: Outputs only final diagnostic results
> - **CPathAgent**: Trained with **reasoning-focused** instruction data including:
>   - Rationale for zoom/navigation actions
>   - Step-by-step diagnostic feature identification
>   - Multi-scale multi-image reasoning chains
>
> **Clarification on Figure 2:**  Figure 2 shows our data generation pipeline, not inference (shown in Figure 1). The patch descriptions serve **solely** as auxiliary information to help Gemini-2.5-Pro produce more accurate training data for CPathAgent.
>
> To further validate CPathAgent's advantages, we  conducted additional experiments to compare CPathAgent and CPath-Omni on two datasets:
>
> **PathMMU-HR²:**
>
> |Model|BRCA|LUAD|LUSC|KIRP|KIRC|KICH|ESCA|THCA|BLCA|TGCT|**Overall**|
> |-|-|-|-|-|-|-|-|-|-|-|-|
> |CPath-Omni|72.6|77.6|71.3|64.1|67.5|59.7|74.3|71.5|76.6|78.8|71.7|
> |**CPathAgent**|**87.0**|**88.5**|**87.8**|**87.9**|**92.9**|**78.9**|**90.7**|**89.0**|**90.7**|**93.0**|**88.6**|
>
> **BRACS (Held-out High-Resolution Dataset for breast carcinoma classification):**
>
> |Model|balanced acc|
> |-|-|
> |GPT-4.1|44.2%|
> |Gemini-2.5-Pro|49.0%|
> |CPath-Omni|57.9%|
> |**CPathAgent**|**64.3%**|
>
> The **16.9%** improvement on PathMMU-HR² and **6.4%** improvement on the independent BRACS dataset demonstrate that performance gains stem from our pathologist-like reasoning approach in Stage 3.
>
>
> > ### Lack of training method descriptions for WSI classification tasks, and relevant experimental details are not found in supplementary materials C.2. The MIL method results are conventional, but the evaluation method for MLLMs is unclear. Need to clearly specify key information such as input image, resolution and prompt design.
>
> Thank you for highlighting this important omission. We adopt the same training methodology and MLLM evaluation implementation as CPath-Omni to ensure fair comparison. We will add these details in the revised paper.
>
>
> > ### This is a super-resolution evaluation dataset, but CPathAgent cannot directly process super-resolution images, and input images will be resized to 1008×1008. Under these circumstances, **do the evaluation dataset images only need to be 1008×1008? **Since CPathAgent achieves 88.6 performance at 1008×1008 resolution. **Can I consider this not strictly a huge region benchmark?**
>
> Thank you for raising this important question. I understand your concern, but there's a key misunderstanding about how CPathAgent processes huge regions. **CPathAgent DOES process super-resolution images, it analyzes multiple original-resolution sub-cropped regions, not a single downsampled image.**
>
> Here's how it works (as shown in Figure 1 and detailed in Section 3.1):
>
> **1. Navigation Planning Phase:**
>
> - Input: 16000×16000 huge region → temporarily downsampled to 1008×1008
> - Purpose: Only to plan WHERE to look (like a pathologist viewing at low magnification)
> - Output: Navigation path with coordinates and magnifications
>
> **2. Multi-scale Diagnostic Reasoning Phase:**
>
> - Returns to the **ORIGINAL 16000×16000 image**
> - Crops multiple regions based on planned path at **native resolution** (e.g., 1000×1000 pixels from the **original** image)
> - Reasoning across ~10 different views sequentially
>
> **The 88.6% accuracy comes from analyzing multiple native-resolution patches across the huge region, NOT from a downsampled 1008×1008 image. **In addition, as shown in Table 2, when other models adopt our agent-based approach, they gain ~3% on average (Gemini-2.5-Pro: 73.2%→76.4%, GPT-4.1-mini: 60.1%→62.3%). This confirms PathMMU-HR² is indeed a huge region benchmark requiring multi-view analysis.
>
>
> >  ### The results showing CPathAgent far exceeding Gemini-2.5-Pro need detailed analysis**, which would bring greater value to readers than simply presenting data.
>
> This is a very good question. The reason lies in the training data generation, Gemini-2.5-Pro had access to **ground truth WSI reports as guidance**, enabling it to produce significantly higher quality navigation paths and diagnostic reasoning **than its zero-shot capabilities**. CPathAgent, trained on this high-quality data, can therefore surpass Gemini-2.5-Pro's zero-shot performance.
>
> **Quantitative Evidence** for WSI classification (Table 3):
>
> - **Upper Bound** (synthetic training data generated by Gemini-2.5-pro): 91.7%
> - **Zero-shot Gemini-2.5-Pro**: 72.1%
> - **CPathAgent**: 82.8%
>
> The 19.6% gap between **upper bound and zero-shot Gemini-2.5-Pro** quantifies how WSI report guidance dramatically improves data quality. CPathAgent successfully distills this guided knowledge, outperforming zero-shot Gemini by 10.7%.

---

> > ### Comment · Reviewer_cWNz · 2025-08-01
> > **Response**
> >
> > The authors provided a two-part data structure in their response; however, in Figure D.5, “Navigation Planning” and “Multi-scale Sequential Reasoning” are presented in a mixed manner. Based on the provided training data structure, it is difficult to understand how the behavior shown in Figure D.5 could emerge. During the reasoning process, once the model outputs a coordinate, it is expected to perform a zoom-in operation and then input the corresponding image region for a second round of analysis. However, as shown by the authors, the model performs 9 image operations in a single inference, each requiring image input and historical context. We have doubts about the practical feasibility of this approach. I have run Deepeyes [1]  training process myself and know that so many steps are not feasible, even Deepeyes used reinforcement learning (RL), while the authors rely only on supervised fine-tuning (SFT). The main text provides almost no detailed explanation on how “Navigation Planning” is implemented or how the training data structure is designed.
> >
> > **Reviewer e2cS** also explicitly asked, “How is navigation planning actually implemented?”, indicating that this core mechanism has not been adequately explained. This is a key contribution of the paper, yet the execution details are insufficiently demonstrated. As a reviewer, I believe this issue requires a detailed explanation in a revised submission.
> >
> > On the other hand, **Reviewer EKpp** pointed out that **CPathAgent was further trained on CPath-Omni (which was itself pre-trained on TCGA)**, meaning the evaluation results on TCGA do not constitute a true zero-shot setting. This is also a concern of mine. I explicitly asked: “Are the performance gains from patch and patch descriptions (as shown in Figure 2) or from reasoning-focused instruction data?” However, the authors did not directly address this. They only compared the overall performance of CPath-Omni and CPathAgent, but the two differ significantly in their training data structure, making it difficult to determine whether the performance improvements stem from patch-description data, reasoning-focused instruction data, or the “zoom-in” mechanism itself. This significantly weakens the persuasiveness of the paper's conclusions.
> >
> > Based on this, I am inclined to **maintain or even lower the score**. While this work clearly involves substantial effort, the paper contains too many missing or underdeveloped details.
> >
> > [1] Zheng Z, Yang M, Hong J, et al. DeepEyes: Incentivizing" Thinking with Images" via Reinforcement Learning[J]. arXiv preprint arXiv:2505.14362, 2025.

---

> ### Author Response · Authors · 2025-08-01
> **Clarifications and Additional Results**
>
> Thank you for your comments. We appreciate your engagement and would like to address each point with **specific references to our manuscript**.
>
> ## 1. Clarification on Figure D.5 and the Navigation Planning Pipeline
>
> **Figure D.5 actually showcases the final outputs of our two-stage inference process, rather than the training procedure.**  By presenting both the planned navigation paths and final diagnostic reasoning together, we aim to help readers visualize how the navigation strategy contributes to final diagnostic insights.
>
> **Our two-stage process is described in detail throughout the manuscript:**
>
> 1. **Navigation Planning Stage** **(Lines 165-167)**: We state "For each preserved huge region, the model plans a navigation path across spatial coordinates and magnification levels. The goal is to plan sequentially examine areas that are most likely to yield diagnostic insights." This states that we first plan the complete navigation path.
>
> 2. **Multi-scale Sequential Reasoning Stage (Lines 168-171, 203-209)**:
>    - **Lines 168-170**: "Along the planned navigation path, the model observes each field of view, integrates visual features across scales and generates step-by-step reasoning descriptions"
>    - **Lines 203-204**: "Multi-Image Understanding: We extract image crops at specified magnification levels along the generated viewing paths..."
>
>    **As we describe in our manuscript, we crop ALL images along the path at their native resolutions (specified coordinate and magnification levels) and then feed them to the model for ONE-SHOT multi-image reasoning.** The "9 operations" are planned waypoints, not separate inference cycles.
>
>    We hope these specific line references help clarify our implementation approach, which is detailed throughout the manuscript.
>
> ## 2. DeepEyes Comparison
>
> DeepEyes's operation numbers are much lower because: **DeepEyes addresses targeted visual QA tasks** (e.g., "Is the window black and square?", "What color is the blazer?" as shown in their examples) on **standard-resolution images**, where the visual targets are **clearly defined and 1-3 zoom-ins sufficiently** to answer the question.
>
> **In contrast, CPathAgent analyzes 16,000×16,000 pathology images** containing extensive diagnostic features that require **systematic multi-scale examination across tissue regions**. This is **exploratory diagnostic reasoning**, not simple verification. **The number of navigation steps is determined by inherent task requirements and image complexity**, not the choice of RL vs. SFT.
>
> These highlight the novelty of our setting, which, to our knowledge, has not been explored in prior work. While the current approach is not perfect, **the empirical results suggest that this direction is promising and merits further investigation.**
>
> ## 3. **Training Data** Clarification
>
> **We would like to clarify that we do not train on the patch descriptions shown in Figure 2.** These patch descriptions **serve as intermediate materials used to guide Gemini in generating navigation planning data.** As noted in Figure 2's caption, this figure illustrates "the generation of instruction-tuning data for CPathAgent," rather than the inference process. Figure 1 presents the actual CPathAgent inference workflow.
>
> Regarding the relationship with CPath-Omni, it appears there may be a different interpretation of Reviewer EKpp's original concern. CPathAgent was not **" further trained based on CPath-Omni model"** as stated in **Lines 255-256**, we only inherited patch-level training data (CPathInstruct and CPathCaption, **appendix line 218-227**) that CPath-Omni uses. This is unrelated to TCGA-specific data or WSI-level training.
>
> **This follows common practice in the field:** similar to how general vision-language models require pre-training on datasets like CC3M, LAION or PMC-15M (e.g., LLaVA, LLaVA-Med) before being fine-tuned on their constructed data, we use patch-level data for basic histological understanding before navigation and reasoning training. This does not affect evaluation validity, as the patch-level data contains no WSI-level or TCGA-specific information.
>
> Because the patch-level descriptions shown in Figure 2 were *not* used during training, we are unsure which specific comparison needs to be done.
>
> To address this potential concern, **we conducted additional experiments using CPathAgent checkpoints pre-trained only on the inherited patch-level data (without our reasoning data).** We hope these results help address your question:
>
> |Model|PathMMU|PathMMU-HR2|WSI Classification|
> |-|-|-|-|
> |CPathAgent (w/ patch-only data)|74.4%|68.3%|73.2%|
> |CPathAgent (w/ reasoning data)|78.6%|88.6%|82.8%|
> |**Improvement**|4.2%|20.3%|9.6%|
>
> These results demonstrate that **navigation planning and reasoning data provide substantial gains beyond patch-level pretraining,** especially on tasks requiring multi-scale integration.
>
> We greatly appreciate your feedback and welcome any further suggestions you may have.

---

> > ### Comment · Reviewer_cWNz · 2025-08-02
> > **Response**
> >
> > Thank you for your response, and I have carefully considered your results. I appreciate the authors’ efforts and the clarifications provided. However, I believe that the manuscript still lacks sufficient detail in important aspects, and thus I retain my original score.

---

> > > ### Author Response · Authors · 2025-08-02
> > > **Thank you**
> > >
> > > We sincerely thank Reviewer cWNz for their insightful evaluation, and we hope our extensive technical discussions help explore the potential value of this work to the pathology AI community. We also appreciate the feedback on aspects that could potentially lead to misinterpretation, and we will carefully revise these portions to ensure clarity.

---

### Official Review · Reviewer_m8m7 · 2025-07-03

**Clarity:** 2
**Significance:** 2
**Originality:** 3
**Rating:** 4
**Confidence:** 4

**Summary:**

Traditional computational pathology models lack the multi-scale reasoning process employed by pathologists during diagnosis, resulting in limited interpretability and misalignment with real-world clinical workflows. This paper proposes CPathAgent, an innovative agent-based model designed for high-resolution pathology image analysis that mimics the diagnostic logic of expert pathologists. CPathAgent dynamically performs zooming and navigation operations, and incorporates a multi-stage training strategy to achieve interpretable analysis. In addition, the study introduces PathMMU-HR², the first benchmark dataset for huge region reasoning, covering diverse tissue types and pathological features. The method proposed in this paper achieves better results than the SOTA methods.

**Questions:**

Please refer to the “Weaknesses” section.

**Ethical Concerns:**

["NO or VERY MINOR ethics concerns only"]

**Final Justification:**

The authors’ responses have addressed most of my concerns. I will maintain my score.

**Quality:**

3

**Strengths And Weaknesses:**

Strengths:
1. This paper enhances the performance of current Pathology Language-Vision Models by simulating the diagnostic process of human pathologists reading pathology slides, demonstrating that mimicking this missing process can improve diagnostic accuracy.
2. It constructs a dataset specifically designed for training pathology model agents.
3. It constructs a dataset for analyzing huge region pathology images.

Weaknesses:
1. I think this paper places too much emphasis on dataset construction during the writing process, which leads to an oversimplified and weakened description of the agent’s architecture, training process, and reasoning mechanism. This may give other researchers the impression that the paper is primarily about dataset track rather than method innovation.
2. Many of the training steps in this paper overlap with CPath-Omni. However, the differences from CPath are not sufficiently highlighted, and the paper should better emphasize its innovations.
3. The dataset used for training the navigation module is mainly generated by large models. It remains unclear whether the generated navigation paths align with the actual behaviors of human pathologists when reading pathology slides. This needs further validation through experiments or expert analysis.
4. The paper lacks an analysis of the functional contributions of different agent components; specifically, it does not include ablation studies on individual modules, making it difficult to assess their respective impacts on overall performance.
5. Compared with CPath-Omni, the proposed method requires multiple invocations of VLMs, yet the paper does not provide a comparative analysis in terms of time efficiency and computational cost.
6. Will the dataset constructed in this paper be open-sourced?

---

> ### Author Rebuttal · Authors · 2025-07-31
>
> We greatly appreciate your thoughtful feedback and are grateful for your recognition of CPathAgent's innovative approach in mimicking pathologists' multi-scale diagnostic reasoning and our contributions to dataset construction for pathology model training,  and we address all the concerns you have point by point below:
>
> >  ### I think this paper places too much emphasis on dataset construction during the writing process.
>
> We appreciate the reviewer's valuable feedback.  Due to space limitations, technical details are currently placed in Appendix C.1 (model architecture and training strategy).  We will move this part to Section 3, and restructure the paper to prioritize the **agent-based architecture** and **multi-scale reasoning framework** in the main paper.
>
>
> >  ### Many of the training steps in this paper overlap with CPath-Omni. However, the differences from CPath are not sufficiently highlighted, and the paper should better emphasize its innovations.
>
> We appreciate the reviewer's important observation. While CPathAgent inherits some patch-level training components from CPath-Omni, it represents a **fundamentally different paradigm** for WSI and huge region analysis:
>
> The critical innovation lies in **Stage 3 of our training pipeline**, the agent-based reasoning framework for WSI/huge region analysis:
>
> **1. Architectural Innovation:**
>
> - **CPath-Omni**: Static MIL-based approach that embeds all patches and produces a single output
> - **CPathAgent**: Dynamic agent framework that performs hierarchical examination: first planning at low magnification, then strategically navigating to regions of interest and reasoning across multiple scales and multiple images.
>
> **2. Interpretability & Clinical Utility:**
>
> - **CPath-Omni**: Black-box predictions without reasoning traces
> - **CPathAgent**: Full interpretability with:
>   - Real-time natural language explanations for each navigation decision
>   - Step-by-step diagnostic reasoning (see Appendix Figures D.5-D.7)
>   - Verifiable examination paths that clinicians can validate
>
> In cancer diagnosis, AI can't replace doctors since no model is 100% accurate. With black-box models like CPath-Omni that only give final answers, doctors still need to manually check the entire slide, as they can't tell whether the AI actually "saw" the cancer or just made a lucky guess. In contrast, CPathAgent shows its WSI navigation and reasoning step-by-step, so doctors can quickly verify the AI's reasoning, transforming AI from a black box into a **collaborative diagnostic tool**.
>
>
>
> >  ### The dataset used for training the navigation module is mainly generated by large models. It remains unclear whether the generated navigation paths align with the actual behaviors of human pathologists when reading pathology slides. This needs further validation through experiments or expert analysis.
>
> This is a good question. We ensure synthetic data quality through:
>
> **1. Expert-Guided Prompt Development:**  During prompt engineering, board-certified pathologists iteratively refined our prompts to ensure generated paths reflect clinical examination patterns (detailed prompts in Appendix D.8-D.16).
>
> **2. Empirical Performance Validation:** The quality of synthetic data is demonstrated by  directly using synthetic 'ground truth' data for WSI classification, indicated by **upper bound results** in Table 3:
>
> - TCGA-ESCA: 100% accuracy
> - TCGA-NSCLC: 97.9% accuracy
> - TCGA-BRCA: 97.0% accuracy
>
> These results confirm that our synthetic diagnostic descriptions contain accurate and complete clinical and diagnostic information.
>
> **3. Expert Validation:** Following your suggestion, we conducted further validation where pathologists evaluated 150 randomly sampled navigation paths and reasoning traces on three criteria:
>
> - Diagnostic reasoning validity
> - Coverage of critical diagnostic regions
> - Clinical soundness
>
> **Result: 92.7% (139/150)** met clinical standards, confirming strong alignment with actual pathologist workflows.
>
> We will add these validation results to the main paper to strengthen our claims about clinical relevance.
>
>
>
> > ### The paper lacks an analysis of the functional contributions of different agent components; specifically, it does not include ablation studies on individual modules, making it difficult to assess their respective impacts on overall performance.
>
> Thank you for this valuable suggestion. Following your suggestion, we conducted systematic ablation studies to evaluate the functional contribution of each agent component, we isolated the impact of individual modules by introducing controlled perturbations:
>
> 1. **Global Screening Ablation**: We replaced the strategic region selection with random selection of 70% of WSI regions. This 70% threshold aligns with the global screening module's design objective of identifying diagnostically relevant regions from WSI thumbnails while filtering out ~30% uninformative areas.
> 2. **Navigation Planning & Multi-scale Reasoning Ablation**: Replaced dynamic navigation paths and multi-scale reasoning with direct whole-region description.
>
> **Results**
>
> | Model                                  | TCGA-BRCA | TCGA-NSCLC | TCGA-RCC | TCGA-ESCA | TCGA-BLCA | TCGA-THCA | **Avg**  |
> | -------------------------------------- | --------- | ---------- | -------- | --------- | --------- | --------- | -------- |
> | Full CPathAgent                        | **88.5**  | **90.8**   | **94.6** | **97.1**  | 62.7      | **63.2**  | **82.8** |
> | **w/o Strategic Global Screening**     | 84.7      | 88.7       | **94.6** | 94.1      | 60.8      | 58.9      | 80.3     |
> | w/o Navigation & Multi-scale Reasoning | 80.0      | 83.6       | 88.7     | 94.1      | 52.2      | 51.3      | 75.0     |
>
> **Random Global Screening Impact (-2.5%)**: The modest performance drop validates that strategic region selection effectively balances efficiency and accuracy, successfully mimicking pathologists' selective attention.
>
> **Navigation & Multi-scale Reasoning Impact (-7.8%)**: The substantial performance degradation reveals:
>
> - Direct processing of high-resolution regions (16000×16000) causes critical information loss through forced resizing
> - Absence of pathologist-like reasoning that strategically integrates multiple fields of view at varying magnifications impact model performance
>
> These results demonstrate that our agent-based approach derives its effectiveness from the synergistic interaction of all components, with navigation and multi-scale reasoning being particularly crucial for diagnostic accuracy.
>
>
>
> >  ### Compared with CPath-Omni, the proposed method requires multiple invocations of VLMs, yet the paper does not provide a comparative analysis in terms of time efficiency and computational cost.
>
> Thank you for this important question about computational efficiency. We conducted a comparative analysis using 100 randomly sampled WSIs to evaluate the time and computational costs between CPath-Omni and CPathAgent on H800 GPU.
>
> | Model      | Vision Preprocessing Part | LMM Part                                           | Total Time |
> | ---------- | ------------------------- | -------------------------------------------------- | ---------- |
> | CPath-Omni | 153.6s                    | 0.7s （only generate single classification result) | 154.3s     |
> | CPathAgent | 0s                        | 237.4s                                             | 237.4s     |
>
> The two models exhibit fundamentally different computational profiles:
>
> **CPath-Omni**: Requires extensive preprocessing, which extracts and encodes patches at multiple scales across the entire WSI. For a single region, it processes three different scales, resulting in substantial preprocessing overhead (153.6s).
>
> **CPathAgent**: Eliminates vision preprocessing by directly operating on regions of interest. To be specific,  we leverage vLLM for parallel region processing. On H800 GPU, CPathAgent processes each 16,000×16,000 region in **12.7 seconds** (averaged over 1,500 regions), with a total of **237.4 seconds per WSI**.
>
> While CPathAgent shows some increase in total processing time (83.1s), this represents a worthwhile trade-off. The agent-based approach provides Interpretable diagnostic reasoning and superior accuracy through explicit navigation paths. In this work, we focus on proposing a novel paradigm for pathology analysis, with efficiency optimization left for future work.
>
>
>
> >  ### Will the dataset constructed in this paper be open-sourced?
>
> Yes, we are committed to fully open-sourcing our complete dataset, including all synthetic navigation paths, diagnostic reasoning data, and associated annotations to support reproducibility and future research in interpretable pathology AI.

---

> > ### Comment · Reviewer_m8m7 · 2025-08-06
> >
> > The authors’ responses have addressed most of my concerns. This article requires a certain degree of adjustment in its overall structure compared to the final version. Therefore, I will maintain my score.

---

### Official Review · Reviewer_e2cS · 2025-07-07

**Clarity:** 2
**Significance:** 3
**Originality:** 3
**Rating:** 4
**Confidence:** 3

**Summary:**

This paper introduces CPathAgent, an agent-based framework for pathology image analysis that mimics pathologists' diagnostic workflow. The paper defines an agent as: a system capable of autonomously executing zoom-in/out and navigation operations across pathology images, achieving specific goals through perceiving the environment, making decisions, and taking actions. Specifically, CPathAgent emulates pathologists' diagnostic process through three stages: (1) Global Screening stage to identify suspicious regions; (2) Navigation Planning stage to formulate examination paths; (3) Multi-scale Multi-view Sequence Reasoning stage to integrate observations across different magnification levels. The system can mimic pathologists by starting with low-magnification scanning, progressively zooming in on regions of interest, and formulating diagnostic conclusions through dynamic visual reasoning processes. This paper also proposed the 'PathMMU-HR' dataset.

**Questions:**

1. How do your proposed AI agent adapt the newly incomed training data?
2. Decision-making mechanism entirely opaque: During training, Gemini generates navigation paths based on WSI reports, but during inference without such reports, how does the model determine where to examine next?
3. Bridging mechanism unexplained: Does the model memorize navigation patterns or genuinely learn visual reasoning? Generalization capability questionable: If the system merely memorizes training navigation patterns, how does it handle novel pathological presentations?
4. Generalization capability questionable: If the system merely memorizes training navigation patterns, how does it handle novel pathological presentations?
5. Base model unspecified: The paper merely mentions "open-source VLMs" without detailing the specific architecture.

**Ethical Concerns:**

["NO or VERY MINOR ethics concerns only"]

**Final Justification:**

Thank you for your rebuttal. I think you have answered my concern. I have changed the rating. I think it is worth discussing this work in the conference.

**Limitations:**

No assessment of the error accommulation. This paper proposed the method contains three agents. Each agent can make mistake. The author did not provide the detailed information about the performance when error may accommulates. For instance, the first stage 'Global screening and region selection' made a mistake(such as filtered a important tile), what will happend at the end? Backtracking mechanism: Can the system return to previously examined locations for re-evaluation? If can not, if would be interest for future research. Lack of interactivity: Real pathologists adjust staining, section angles based on preliminary findings - this system cannot.

**Quality:**

2

**Strengths And Weaknesses:**

Strengths:
1. Clinical Relevance
Successfully addresses the gap between AI systems and actual pathologist workflows by mimicking their multi-scale examination process, potentially improving interpretability and clinical adoption.
2. Novel Benchmark (PathMMU-HR²)
Introduces the first benchmark for huge region analysis (16000×16000 pixels) - a critical but previously overlooked scale in pathology. Validated by three board-certified pathologists.
3. Comprehensive Evaluation
Evaluates across three scales (patch, huge region, WSI) on multiple datasets, showing consistent improvements and versatility of the approach.
4. Pathologist Validation
Includes real pathologist evaluation of navigation paths and reasoning quality, adding clinical credibility beyond just benchmark metrics.

Weakness:
1.Novelty Issues with Agent Definition
The paper claims to be "the first agent-based framework for pathology," but it essentially repackages traditional multi-scale image analysis as an agent concept. The authors should compare their work with existing popular agent frameworks and emphasize the rationality of their proposed pathology-specific agent definition. The so-called "agent behaviors" (zooming, navigation) are fundamentally predefined image cropping and scale selection, lacking genuine autonomous decision-making capabilities. Compared to existing attention mechanisms or multi-scale feature fusion methods, the conceptual innovation is limited.

2.Reliability of Data Generation

The work heavily relies on Gemini-2.5-Pro to generate training data (278K instances) - how is the quality of such synthetic data ensured? Do the generated navigation paths and diagnostic reasoning genuinely reflect general pathologists' actual workflows? There is a lack of comparative validation against real pathologist behavioral data.

3.Fairness of Evaluation
In the PathMMU-HR² benchmark, only CPathAgent uses the agent-based approach, while other models do not employ the same prompting strategy. Part of the performance improvement may stem from the inherited CPath-Omni training data rather than the agent mechanism itself. The pass@k evaluation uses only 40 samples, which lacks statistical significance.

4.Ambiguity in Technical Implementation
How is "navigation planning" actually implemented? Is it merely selecting image patches according to predefined rules? The paper does not clearly explain how the agent "decides" when to stop exploration.There is a lack of analysis on computational complexity and inference time.

---

> ### Author Rebuttal · Authors · 2025-07-31
>
> We greatly appreciate your thoughtful feedback and are grateful for your recognition of CPathAgent's clinical relevance  and for addressing the gap between AI systems and actual pathologist workflows by mimicking their examination process. However, some concerns may stem from misunderstandings, and we address these one by one below:
>
> > ### It essentially repackages traditional multi-scale image analysis as an agent concept. The so-called "agent behaviors"  are fundamentally predefined image cropping and scale selection, lacking genuine autonomous decision-making. Compared to existing attention mechanisms or multi-scale feature fusion, the conceptual innovation is limited.
>
> We appreciate the reviewer's concern but respectfully disagree that CPathAgent merely repackages traditional multi-scale analysis.
>
> **1. Most agent frameworks operate within constrained action spaces**, classical agent systems define constrained action sets: WebGPT [1] (browser commands), ReAct [2] (search/lookup/finish), MM-ReAct [3] (crop/enhance). **Autonomy in agents is characterized by learning policies for action selection, not by discovering new actions**. This is commonly recognized in agent works.
>
> **2. CPathAgent exhibits genuine autonomous decision-making** Unlike static multi-scale fusion, CPathAgent:
> - **Dynamically plans navigation paths** based on visual observations.
> - **Learns when and where to zoom and move**.
> - **Generates action rationales** explaining why each decision was made
>
> This differs fundamentally from predetermined attention mechanisms lacking intermediate reasoning and interpretability.
>
> **3. Quantitative evidence** The significant performance gaps between agent-based and non-agent approaches in Table 2 (Gemini-2.5-Pro: 73.2%→76.4%, GPT-4.1-mini: 60.1%→62.3%) demonstrate that dynamic navigation policies  benefits model performance.
>
> Notably, several pathology agent works have emerged after our submission [4,5], further supporting our agent paradigm.
>
> [1] Nakano et al. WebGPT. arXiv:2112.09332, 2021.
>
> [2] Yao et al. ReAct. ICLR 2023.
>
> [3] Yang et al. MM-ReAct. arXiv:2303.11381, 2023.
>
> [4] Li et al. OmicsNavigator. bioRxiv, 2025.
>
> [5] Chen et al. Evidence-based diagnostic reasoning with multi-agent copilot for human pathology. arXiv:2506.20964, 2025.
>
> > ###  How is the quality of such synthetic data ensured? Do the generated navigation paths and diagnostic reasoning genuinely reflect pathologists' actual workflows?
>
> This is a good question. We ensure synthetic data quality through:
>
> **Prompt Refine**: During prompt engineering, board-certified pathologists provided iterative feedback to refine our prompts, resulting in comprehensive prompts (Appendix Figures D.8-D.16).
>
> **Empirical Validation**: The quality is demonstrated by **upper bound performance in Table 3**: achieving **100%** acc on TCGA-ESCA, **97.9%** on TCGA-NSCLC, and **97.0%** on TCGA-BRCA when directly using synthetic 'ground truth' for WSI classification.
>
> To further validate the synthetic data, we invite pathologists evaluate 150 randomly sampled navigation paths and diagnostic reasoning examples based on: (1) diagnostic reasoning validity, (2) coverage of critical diagnostic regions, and (3) clinical soundness. **92.7% (139/150)** met clinical standards, which confirms the data quality.
>
> > ### Fairness of Evaluation In the PathMMU-HR² benchmark, only CPathAgent uses the agent  approach, while other models not. Part of the performance improvement may stem from the inherited CPath-Omni training data rather than the agent mechanism. The pass@k evaluation uses only 40 samples.
>
> Thank you for this important concern.
>
> 1. Regarding  **evaluation fairness,** as shown in Table 2, we provided the **same agent-based prompting strategy** to GPT-4.1-mini and Gemini-2.5-Pro. Under identical prompts, CPathAgent significantly outperformed both (88.6% vs. 62.3% and 76.4%).  These models also improved with our agent approach (GPT-4.1-mini: 60.1%→62.3%, Gemini-2.5-Pro: 73.2%→76.4%).
>
> **2. Performance attribution:** To isolate our agent mechanism from inherited data, we evaluated **CPath-Omni**  on PathMMU-HR²:
> |Model|BRCA|LUAD|LUSC|KIRP|KIRC|KICH|ESCA|THCA|BLCA|TGCT|**Overall**|
> |-|-|-|-|-|-|-|-|-|-|-|-|
> |CPath-Omni|72.6|77.6|71.3|64.1|67.5|59.7|74.3|71.5|76.6|78.8|71.7|
> |**CPathAgent**|**87.0**|**88.5**|**87.8**|**87.9**|**92.9**|**78.9**|**90.7**|**89.0**|**90.7**|**93.0**|**88.6**|
>
> The 16.9% improvement shows that performance gains come from our pathologist-like reasoning (Stage 3's data), not inherited training data.
>
> **3. Statistical significance:** We expanded pathologist evaluation to 100 samples. Results show navigation path satisfaction reached 85.0% and reasoning satisfaction reached 89.0% at pass@8. Due to rebuttal format limitations, we will present the figures in the revised paper.
>
> > ### How is "navigation planning" actually implemented? Is it merely selecting image patches according to predefined rules?  There is a lack of analysis on computational complexity and inference time.
>
> Thank you for this important technical question. Our navigation planning is **not rule-based** but fully autonomous. The model:
> 1. Observes each huge region at low magnification
> 2. Analyzes visual features to decide next actions (zoom locations and scales)
> 3. Outputs navigation decisions in JSON format for execution
> This autonomous navigation is learned during training, not predefined.
>
> Regarding computational complexity, we leverage **vLLM** parallel processing to analyze multiple regions simultaneously. On H800 GPU, CPathAgent processes each 16000×16000 region in **12.7 seconds on average** (two VLM calls, averaged over 1,500 huge regions), with **237.4 seconds for each WSI**. While slower than MIL, this trade-off is worthwhile as the intermediate reasoning process is crucial for interpretability and diagnostic accuracy. In this work, we focus on propose a novel paradigm for pathology analysis, with efficiency optimization left for future work.
>
>
> > ### Q1 How do your proposed AI agent adapt the newly incomed training data?
>
> Thank you for this important question. CPathAgent handles new data through two approaches:
> 1. **Zero-shot Generalization:** CPathAgent demonstrates strong generalization by learning pathologists' inherent reasoning logic rather than memorizing dataset-specific patterns. We evaluated CPathAgent on the CPTAC-lung (non-TCGA, holdout data) without  fine-tuning:
>
> |Model|Training WSIs|BACC|
> |-|-|-|
> |PRISM|587,196|83.2|
> |CPath-Omni|11,728|82.3|
> |CPathAgent|5,254|**88.1**|
>
> Despite pretrained on significantly fewer WSIs, CPathAgent achieves superior generalization to unseen datasets.
>
> **2. Domain Adaptation:** When needed, our framework can automatically generate high-quality agent-based training data for fine-tuning using WSI reports (easily available in practice), without costly manual annotations.
>
> > ### Q2 During training, Gemini generates navigation paths based on WSI reports, but during inference without such reports, how does the model determine where to examine next?
>
> We need to clarify that the WSI reports are used **only during training data generation**, not during model training or inference.
>
> In the **Data Generation Phase,** we use WSI reports to guide Gemini-2.5-Pro in generating high-quality navigation paths and reasoning sequences. Without this guidance, Gemini-2.5-Pro often produces low-quality data with incorrect diagnoses and navigation patterns.
>
> During the **Training/Inference Phase,** the model **never** uses WSI reports. It learns to navigate and reason based purely on visual features and autonomously decides where to examine like pathologists do without prior reports.
>
> >  ### Q3: Does the model memorize navigation patterns or genuinely learn visual reasoning? If the system merely memorizes training navigation patterns, how does it handle novel pathological presentations?
>
> Thank you for this insightful question. We provide several  **empirical evidence against pure memorization:**
>
> 1. **Low Navigation Path Similarity:** We analyzed navigation paths across 1,000 test samples using step-wise Region of Interest (ROI) Intersection over Union (IoU) as the similarity metric. The average correlation between navigation paths from different images was only **0.12**, demonstrating that the model generates content-adaptive navigation strategies rather than memorizing fixed patterns.
> 2. **Performance on Diverse Test Cases:** TCGA WSIs exhibit significant heterogeneity across different cases. If model only memorized patterns, performance would deteriorate dramatically. Instead, CPathAgent maintains high performance compared to MIL methods and CPath-Omni.
> 3. **Generalization capability:** As shown in Q1's response, CPathAgent achieves 88.1% on the unseen CPTAC-lung dataset, outperforming models trained on 100× more data. This strong generalization proves the model has learned genuine visual reasoning  rather than memorized patterns.
>
> > ### Q4 Base model unspecified: The paper merely mentions "open-source VLMs" without detailing the specific architecture.
>
> Thank you for highlighting this omission. We apologize for the lack of this information. Due to space constrains, we placed the detailed model architecture and training in Appendix Section C.1.  We will move it to the main paper.
>
> > ###  Each agent can make mistake. The author did not provide the detailed information about the performance when errors may accommulates. Can the system return to previously examined locations for re-evaluation? .
>
> That is correct,  error accumulation in multi-step agent systems is indeed a common challenge in agent systems. Currently, CPathAgent follows a forward-only navigation strategy without backtracking. We agree that  backtracking  would be valuable future work to mitigate error accumulation.
>
> We will provide detailed error cases in the revised appendix (as figures cannot be included in the rebuttal) to better show when and how errors occur in our system.

---

> ### Author Response · Authors · 2025-08-07
>
> Dear Reviewer e2cS,
>
> We have carefully considered each of your concerns and hope that our responses adequately address them. We would be grateful if you could share your feedback on our responses, and we are more than happy to provide any further clarification or engage in additional discussion if needed.
>
> Many thanks for your time and effort!

---

### Official Review · Reviewer_EKpp · 2025-07-21

**Clarity:** 3
**Significance:** 3
**Originality:** 4
**Rating:** 4
**Confidence:** 4

**Summary:**

This paper presents CPathAgent, an extension of CPath-Omni that adds agent-based sequential processing of whole slide pathology images, requiring the model to move around the image, zooming in and out at different resolutions, to produce region/slide descriptions and answer questions.

**Questions:**

1. For acceptance, a more complete comparison with CPathOmni would be necessary, along with further evaluation on whole-slide-level pathology tasks outside of the training distribution. Additionally (but less importantly), comparisons with recent foundation models capable of whole-slide representations, like TITAN or PRISM, would be helpful.

2. More discussion/evaluation is needed to motivate agentic sequential processing of pathology images.

2a. It makes sense that different magnifications are sufficient for different information that can be integrated into a question answer or a diagnosis. However, is it necessary to scan around the image sequentially and zoom in and out? Does this have any advantage over a model that takes the full region or whole slide image at multiple scales as input? Also, is there any additional training signal in the sequential setup that cannot be used in the everything-at-once input approach?

2b. You mention that non-agentic models "fail to replicate the diagnostic process of pathologists". Could you explain why this is a problem?
You mention that non-agentic models have "poor alignment with clinical workflows, undermining their credibility in diagnostic settings". Could you explain what you mean by "poor alignment with clinical workflows" and why this undermines the credibility of validated model performance?

3. Minor questions.
3a. When all parameters are unfrozen, does this include CPath-Clip? If so, then all of it (including Virchow2) or part of it?
3b. How are images passed to Gemini when evaluating it without "CPathAgent's prompting strategy"? Do they still cover a mix of magnifications? Does mixing magnifications improve performance or does the prompting strategy improve performance?
3c. How do differences between models look when computing statistical significance in the tables?

**Ethical Concerns:**

["NO or VERY MINOR ethics concerns only"]

**Final Justification:**

With the additional results and clarifications, I am raising my score. The reasons to accept now outweigh the reasons to reject.

**Limitations:**

See the evaluation concerns discussed above.

**Quality:**

2

**Strengths And Weaknesses:**

STRENGTHS

1. Mimicking pathologists' navigation through whole slide images is an interesting way to extend region-only models to whole slide images. The navigation examples provided are compelling, suggesting that this navigation makes sense.

2. A pathologist-validated visual QA benchmark dataset is proposed (I imagine the data will be released to the community). The dataset excludes questions that get answered (trivially) correctly without an image.

3. CPathAgent appears to outperform CPathOmni on the PathMMU dataset (however, it is suspicious that CPathAgent also substantially outperforms expert pathologists).




WEAKNESSES

1. CPathAgent appears to be pretrained on the "training sets" of most of the evaluation tasks, to solve these tasks. It is unclear whether the model generalizes to new data, so there is a need for evaluations that are distinct from the training data.

1a. CPath-Omni was pretrained to solve the TCGA classification tasks for NSCLC, BRCA, UCEC, THCA, ESCA, BLCA, and TGCT and it appears CPathAgent may be as well: "training data inherited from CPath-Omni". This makes TCGA WSI classification not actually zero-shot and it performs similary to MIL aggregators trained on each of these tasks.

1b. PathMMU-HR2 is developed using the same method / tools as the training data for CPathAgent, so I expect CPathAgent to have an advantage on it.

2. While I imagine that the proposed agentic approach may improve model performance, I am not convinced this is the case by the presented results.

2a. To support the claim that the agentic component of CPathAgent improves over CPathOmni, it would be useful to evaluate CPathOmni on all tasks but it's only evaluated on PathMMU, where CPathOmni is close to expert human performance and CPathAgent somehow greatly improves over pathologists (which is weird). Can CPathOmni be evaluated on PathMMU-HR2 and the TCGA WSI classification tasks as well? Does CPathAgent outperform it there?

2b. Greatly exceeding expert pathologist performance on PathMMU makes the quality of this dataset suspicious. Is the QA frequently wrong? The PathCLS subset of PathMMU was derived from labeled datasets and may be the cleanest and most correctly annotated subset of PathMMU. On this subset, there is no difference in performance between CPathOmni (79.0) and CPathAgent (79.2).

2c. Why are CPathOmni results not shown in Table 3 (TCGA WSI classification)? Looking at the CPathOmni paper, it appears that there may not be much difference between CPathAgent and CPathOmni on these tasks, after all: TCGA-BRCA 88.5 vs 89.2, TCGA-NSCLC 90.8 vs 88.8, TCGA-RCC 94.6 vs 94.0, TCGA-ESCA 97.1 vs 92.1, TCGA-BLCA 62.7 vs 70.7, TCGA-THCA 63.2 vs 58.5; an average balanced accuracy of 82.8 for CPathAgent and 82.2 for CPathOmni. Did the addition of agentic processing of slides really improve over CPathOmni?

---

> ### Author Rebuttal · Authors · 2025-07-31
>
> Thank you for your thorough review and  constructive feedback.  We are grateful for your recognition of CPathAgent as an interesting approach by mimicking pathologists' navigation. Below, we provide detailed responses to each of your concerns:
>
> > ### W1a & Q1. CPathAgent inherits training data from CPath-Omni (pretrained on TCGA), making TCGA evaluation not truly zero-shot and similar to traditional MIL approach.
>
> Thank you for this important question. Although CPathAgent inherits patch-level pretraining from CPath-Omni, its **WSI-level reasoning is fundamentally different**.
>
> **Key Distinction:** While CPath-Omni and MIL methods operate as **black-box models outputting only final predictions**, CPathAgent introduces a **paradigm shift** through agent-based reasoning that mimics pathologists' diagnostic workflow, dynamically navigating, zooming, and generating interpretable step-by-step explanations of observed morphological features and diagnostic rationale across WSIs.
>
> **Zero-shot Validation:** To demonstrate true generalization, we evaluated on the out-of-domain **CPTAC-Lung cohort** using balanced accuracy:
>
> |Model|Training WSIs|BACC|
> |-|-|-|
> |PRISM|587,196|83.2|
> |CPath-Omni|11,728|82.3|
> |CPathAgent|5,254|**88.1**|
>
> Despite training on only 5,254 WSIs, CPathAgent outperforms PRISM (trained on 100× more data) and CPath-Omni (trained on 2× more data) by learning **intermediate reasoning processes** rather than memorizing dataset-specific patterns.
>
> > ###  W1b. PathMMU-HR2 is developed using the same method as the training data for CPathAgent, so I expect CPathAgent to have an advantage on it.
>
> You raise an important point. While we used similar methods for both training data and benchmark construction, this  was **unavoidable as no comprehensive ultra-high resolution pathology benchmarks existed**.
>
> **To further validate our model's generalization**, we try our best to find a **completely held-out**, highest resolution pathology dataset, BRACS (breast carcinoma classification), the only available dataset with regions up to **10K pixels**, and evaluate CPathAgent on it:
>
> |Model|BRACS|
> |-|-|
> |GPT-4.1|44.2%|
> |Gemini-2.5-Pro|49.0%|
> |CPath-Omni|57.9%|
> |**CPathAgent**|**64.3%**|
>
> The 6.4% improvement over CPath-Omni on this **independent dataset** demonstrates the robustness of CPathAgent's agent-based reasoning capabilities.
>
> > ###  W2a&2c&Q1. Can CPathOmni be evaluated on PathMMU-HR2 and the TCGA WSI classification tasks as to properly validate CPathAgent's agentic improvements?
>
> Thank you for this valuable suggestion. We have conducted the requested evaluations:
>
> #### PathMMU-HR² Result
>
> |Model|BRCA|LUAD|LUSC|KIRP|KIRC|KICH|ESCA|THCA|BLCA|TGCT|**Overall**|
> |-|-|-|-|-|-|-|-|-|-|-|-|
> |CPath-Omni|72.6|77.6|71.3|64.1|67.5|59.7|74.3|71.5|76.6|78.8|71.7|
> |**CPathAgent**|**87.0**|**88.5**|**87.8**|**87.9**|**92.9**|**78.9**|**90.7**|**89.0**|**90.7**|**93.0**|**88.6**|
>
> The 16.9% performance gap occurs because other models directly resize ultra-high resolution images (16000×16000) to their fixed input size (e.g., 336*336). In contrast, CPathAgent uses strategic navigation planning and dynamic zoom-in to examine critical diagnostic details without losing important information.
>
> #### WSI Classification Result
>
> Direct comparison on the TCGA WSI tasks is unfair as **CPath-Omni was trained on the larger TCGA dataset (11,728 cases)**, while CPathAgent used only cases with WSI reports (5,254 cases) with **different train/test splits.**
>
> For fair evaluation, we **retrained** CPath-Omni's stage 3 using our identical training split:
>
> |Model|TCGA-BRCA|TCGA-NSCLC|TCGA-RCC|TCGA-ESCA|TCGA-BLCA|TCGA-THCA|**Avg**|
> |-|-|-|-|-|-|-|-|
> |CPath-Omni (retrained)|78.6|86.7|91.3|89.9|**65.5**|52.6|**77.4**|
> |**CPathAgent**|**88.5**|**90.8**|**94.6**|**97.1**|62.7|**63.2**|**82.8**|
>
> CPathAgent achieves a significant 5.4% improvement, demonstrating that learning pathologist-like diagnostic reasoning and intermediate processes leads to better generalization and learning compared to weakly-supervised approaches.
>
> > ###  W2b. Greatly exceeding expert pathologist performance on PathMMU makes the quality of this dataset suspicious, while the PathCLS subset shows nearly identical results.
>
> Thank you for raising this important concern. We provide clarification on several key points:
>
> The PathCLS subset evaluates **simple perceptual classification tasks** with low-resolution images (e.g., PatchCamelyon: 96×96 patches where **simple vision encoders can achieve 99%+ accuracy**), fundamentally different from other complex reasoning questions. Since CPathAgent inherits patch-level classification data from CPath-Omni, similar PathCLS performance is expected.
>
> **Regarding Expert Performance**, CPathAgent exceeds reported expert baselines due to:
>
> - Limited Expert Coverage: PathMMU experts specialize in 1-2 areas while the dataset spans diverse tissues. No single expert masters all domains.
> - Text-based Shortcuts: PathMMU authors acknowledged LMMs can exploit strong text-based guessing abilities that humans cannot easily leverage.
>
> To address the text-based shortcut, PathMMU authors created PathMMU-Pro with more rigorous question design. We  conduct further evaluation on PathMMU-Pro:
>
> |Model|Overall Tiny|Overall All|PubMed Tiny|PubMed All|SocialPath Tiny|SocialPath All|EduContent Tiny|EduContent All|
> |-|-|-|-|-|-|-|-|-|
> |Human|**69.4**|-|**71.2**|-|**70.1**|-|66.9|-|
> |CPath-Omni|62.1|61.8|60.5|60.8|58.8|62.1|66.7|63.0|
> |CPathAgent|66.3|65.3|64.6|65.8|65.4|63.6|**68.7**|66.2|
>
> CPathAgent **no longer exceeds expert performance on this cleaner benchmark** while **still outperforming CPath-Omni**.
>
> > ### Q2a : Is it necessary to scan around the image sequentially and zoom in and out? Does this have any advantage over a model that takes the full region or WSI at multiple scales as input? Is there any additional training signal in the sequential setup that cannot be used in the everything-at-once input approach?
>
> This is an excellent question that highlights a fundamental design choice in our approach. Sequential navigation offers critical advantages:
>
> **1. Interpretable Clinical Reasoning**
>
> Our method generates **step-by-step natural language explanations** of the diagnostic process, unlike MIL/CPath-Omni that produce only final predictions. For **life-critical cancer diagnosis**, AI must provide transparent reasoning that pathologists can verify.
>
> **2. Rich Sequential Supervision Signal**
>
> - Traditional Multi-scale approaches use **weakly supervised WSI-level labels** with MIL, which: (1) Cannot explicit  learn which regions are diagnostically important. (2) Have no guidance on optimal magnification levels (3) Fail to capture the diagnostic reasoning process. Therefore, MIL models can **easily learn spurious correlations** from limited training data, especially when the training signal is very weak (only slide-level labels).
> - Our approach provides **explicit supervision at each step**: where to navigate, when to zoom, and what to examine, this creates a fundamentally stronger learning signal that enables robust reasoning.
>
> **3. Superior Generalization**
>
> - CPathAgent learns the diagnostic **process** rather than dataset-specific patterns, enabling stronger generalization to new datasets and achieving better performance with less data (experiments in W1a&1b)
> - Gemini-2.5-Pro improves from 73.2% to 76.4% when equipped with our sequential navigation approach (Table 2)
>
>
> > ### Q2b: Could you explain why  "fail to replicate the diagnostic process of pathologists" is a problem?   Could you explain what you mean by "poor alignment with clinical workflows" and why this undermines the credibility of validated model performance?
>
> As mentioned in Q2a's response, **cancer diagnosis is life-critical and requires human verification**. Models that only output final predictions force pathologists to manually re-examine entire slides to ensure diagnostic accuracy, as they cannot trace the AI's reasoning to verify critical findings weren't missed.
>
> **"Poor alignment with clinical workflows"** refers to models that don't mirror how pathologists systematically examine tissue: starting with low-magnification overview, then strategically zooming into regions of interest. This misalignment creates two problems:
>
> 1. **Verification inefficiency**: Without interpretable reasoning paths, pathologists cannot quickly validate AI findings
> 2. **Limited generalization**: As our experiments demonstrate (W1a&1b), models aligned with clinical workflows learn the general diagnostic process rather than dataset-specific patterns, enabling better performance with less data.
>
>
>
> > ###  Q 3a. When all parameters are unfrozen, does this include CPath-Clip? If so, then all of it (including Virchow2) or part of it?
>
> Yes, all parameters are unfrozen during Stage 3 training, including the entire CPath-CLIP module (with Virchow2). This allows end-to-end optimization for agent-based reasoning.
>
> > ### Q 3b.  How are images passed to Gemini when evaluating it without "CPathAgent's prompting strategy"? Do they still cover a mix of magnifications?
>
> For non-agent evaluation, models receive the full large region image (16000×16000) directly as a single input, without sequential navigation and do not cover a mix of magnifications. As shown in Table 2, Gemini with "CPathAgent's prompting strategy" compared to without it can improve performance by 3.2%.
>
> > ### Q3c. How do differences between models look when computing statistical significance in the tables?
>
> While we acknowledge the importance of statistical significance testing, LMMs during evaluation typically use a single final result, as the computational cost of running multiple trials with large pathology models is time-consuming and impractical within the rebuttal timeframe; we will address this in future work. However, our method demonstrates consistent improvements across four diverse evaluation settings, providing empirical evidence of effectiveness.

---

> > ### Comment · Reviewer_EKpp · 2025-08-05
> > **Response to rebuttal**
> >
> > Thank you for providing point-by-point responses to the comments in my review. The responses reduce my concerns about the model evaluation. I am inclined to raise my score.
> >
> > With the CPathOmni retraining on the smaller CPathAgent training dataset, it appears that CPathAgent is more data efficient, as shown on the TCGA WSI cancer classification tasks. The improvement over CPathOmni on the held-out CPTAC LUAD dataset is an encouraging confirmation.
> >
> > [Comment] On this task, there is a comparison to PRISM but still no evaluations of TITAN.
> >
> > I remain somewhat skeptical about whether the performance of CPathAgent on PathMMU-HR2 is unrelated to the method in which this dataset was created being so similar to the way CPathAgent's training data was created; however, the BRACS result helps reduce my concern. CPathOmni performs worse on both PathMMU-HR2 and BRACS.
> >
> > [Question 1] Is this the original CPathOmni or the one retrained with less data?
> >
> > It is encouraging to see that no models beat experts on PathMMU-Pro and CPathAgent beats CPathOmni. However, I cannot find PathMMU-Pro online.
> >
> > [Question 2] Is this a dataset you recently created and haven't yet made public? Could you provide some more information on how this data was created, compared to PathMMU?
> >
> > I will address individual responses below:
> >
> > [Question/comment 3: on the response to Q2a]
> > So additional, denser region-level supervision is provided by Gemini. However, in principle, this could also be used when training a model that takes multiple resolutions of a full region or slide at a time, by passing in different regions/resolutions with different descriptions as different samples. Is it correct to then say that beyond training data generation, the process of sequentially scanning and zooming does not provide additional information for representation learning? Its main benefits are then (i) possible interpretability and (ii) adapting a single-resolution small-region architecture to a mutli-resolution whole-slide image.
> >
> > [Question/comment 4: on the response to Q2b]
> > When considering validation, I understand that our discussion separates model performance validation from model result verification, which is required in pathology, and that we are discussing verification. When cancer is detected, a non-agentic model could provide regions of interest with cancer, saving pathologists time for verification. When cancer is not detected, a pathologist may have to review the full slide for both agentic and non-agentic models anyway. I guess the main claim here is that it is faster to review the agentic process? It's a fair claim but it remains up to a later study to evaluate it in a clinical setting. For now, it would be best to phrase it as a conjecture to be tested.
> >
> > [Question/comment 5: on the response to Q3b]
> > Thanks for the clarification: images are passed at full resolution with no additional magnifications and the prompting strategy improves Gemini's performance. Does mixing magnifications also improve Gemini's performance?
> >
> > ADDITIONAL QUESTIONS
> >
> > N1. In clinical practice, cases often have multiple blocks, each containing multiple slides. How do you handle multiple slides for a prediction?
> >
> > N2. Are TCGA NSCLC, BRCA, UCEC, THCA, ESCA, BLCA, and TGCT no longer part of the CPathAgent training data set (unlike that of CPathOmni)?
> >
> > N3. On the feature representation and task transfer.
> > N3a. Is the representation quality of this model higher than that of CPathOmni that was re-trained on the same slides?
> > N3b. How does the CPathOmni-like component of CPathAgent behave on BRACS without scanning and zooming?
> > N3c. How can this model be adapted to out of domain tasks like survival prediction or biomarker detection? Is it mainly intended to be a diagnostic model?

---

> ### Author Response · Authors · 2025-08-07
> **Thank you and response to the follow-up questions (1/2)**
>
> Thank you for your thoughtful follow-up questions and for considering raising your score. We also appreciate your recognition of CPathAgent's data efficiency and performance improvements. Below we address your specific questions and comments:
>
> > ###  Comment 1: Comparison with TITAN on CPTAC
>
> Following your suggestion, we evaluated TITAN on CPTAC:
>
> |Model|Training WSIs|BACC|
> |-|-|-|
> |CPathAgent|5,254|88.1|
> |TITAN|335,645|91.9|
>
> While TITAN achieves higher accuracy, it requires **64× more training data** than CPathAgent. Data quantity is a critical constraint in pathology. This aligns with TITAN's own ablations on TCGA-OT-8K, which show **using 12.5% of pretraining data (41,955 WSIs) reduces the accuracy from 0.818 to 0.784.**
>
> If TITAN's pretraining data were reduced to **CPathAgent's scale (1/64, 5,254 WSIs), the performance gap would likely diminish substantially or even reverse.**
>
> More importantly, beyond performance metrics, **CPathAgent's core contribution is proposing a fundamentally new paradigm**, which introduces an explainable and traceable AI agent pathological diagnosis framework. Performance optimization, while important, is reserved for future iterations.
>
> > ### Q1: Is this the original CPathOmni used in PathMMU-HR2 and BRACS evaluations?
>
> Yes, it is the original CPathOmni version trained on the larger dataset (11,728 WSIs), not the retrained version.
>
> > ###  Q2: More information about PathMMU-Pro and how was it created?
>
> PathMMU-Pro was recently created by the PathMMU authors to address text-based shortcuts in the original PathMMU. The dataset is available at their GitHub repo provided in their paper [1] (rebuttal policies prevent direct URL sharing).
>
> PathMMU-Pro was created by:
>
> 1. Training text-only models (Phi3 & Llama3) on PathMMU's training set to equip them with stronger question-guessing capabilities, enabling identification of questions that could be answered through text patterns alone without visual information
> 2. Filtering out these "guessable" questions from the test set to ensure visual understanding is necessary
> 3. Using GPT-4o to generate more confusing options that are textually similar but require careful visual examination to distinguish
>
> [1] Sun Y, et al. PathBench: Advancing the Benchmark of Large Multimodal Models for Pathology Image Understanding. IEEE TMI, 2025.
>
> ------
>
> > ###  Q3: In principle, region-level supervision could also be used when training a model that takes multiple resolutions of a full region or slide at a time.
>
> This is a good question. In theory, it's absolutely right. However, there are significant practical challenges that make this approach infeasible:
>
> 1. **Computational Infeasibility**:  Processing 4 magnifications (1×, 2×, 4×, 6×) for a single region requires 57 input images. When scaled to WSIs containing thousands of regions, this becomes computationally prohibitive. Current open-source LMMs struggle with this many inputs due to token limits and degraded understanding.
> 2. **Dynamic Flexibility**: CPathAgent enables dynamic magnification selection and adaptive crop shapes. For instance, ductal structures often require rectangular crops rather than fixed square sliding windows, providing crucial flexibility for accurate diagnosis.
>
> ------
>
> > ### Q4: I guess the main claim here is that it is faster to review the agentic process?
>
> Yes, this is one of the main claims. However, based on preliminary feedback from collaborating pathologists, we hypothesize that CPathAgent offers several potential advantages beyond review speed:
>
> - MIL attention maps, due to their weak supervision nature, **highlight areas that are frequently diagnostically irrelevant or misleading**. Pathologists reported spending considerable time trying to understand why certain areas were flagged.
> - While attention heatmaps show where the model focuses, **they cannot explain why those regions matter diagnostically**.
>
> CPathAgent aims to address these issues through transparent decision-making and explainable rationale. We agree that formal clinical validation studies are needed to quantify these potential benefits, and we plan to conduct such evaluations in future work.
>
> ------
>
> > ### Q5: Does mixing magnifications alone improve Gemini's performance?
>
> This is a great question. Following your suggestion, we conducted ablation studies to isolate exactly this effect:
>
> |Model Configuration|PathMMU-HR2|
> |-|-|
> |Gemini (single resolution)|73.2%|
> |Gemini (all magnification regions)|75.0%|
> |Gemini (with CPathAgent strategy)|76.4%|
>
> - **Multi-magnification does help**: Simply providing all magnification regions to Gemini yields a 1.8% improvement, confirming that multi-scale is valuable.
> - **But it's not enough**: Even with all magnification regions, this approach still underperforms compared to CPathAgent's strategic exploration (76.4%) **while feeding all magnifications of all regions simultaneously requires substantially more computational resources**.

---

> ### Author Response · Authors · 2025-08-07
> **Thank you and response to the follow-up questions (2/2)**
>
> > ### N1: How does CPathAgent handle multiple slides per case?
>
> Multiple slides per case (e.g., gastric tissue biopsies) are indeed common clinically. Our approach naturally handles this scenario: since CPathAgent outputs detailed textual descriptions with spatial locations, we can synthesize findings across multiple blocks using an additional LLM. We plan to evaluate this multi-slide aggregation in future clinical studies.
>
> -----
>
> > ### N2:  Are TCGA subsets no longer part of the CPathAgent training data set (unlike that of CPathOmni)?
>
> CPathAgent's training (CPathAgent-Instruct Dataset) does include TCGA data, as it remains the largest publicly available WSI dataset and excluding it would significantly limit training resources. However, **evaluation datasets (CPTAC, BRACS) are completely held-out test sets with no overlap with training data.** Our 7.8% improvement over CPath-Omni on CPTAC and 6.4% on BRCAS demonstrates CPathAgent's superior generalization capability.
>
> -----
>
> > ### N3a: How does CPathAgent's representation quality compare to retrained CPath-Omni?
>
> While direct feature-level comparison is challenging since LMMs generate text rather than embedding vectors, CPathAgent's representation quality can be assessed through downstream performance.  The improvement across diverse tasks suggests that our CPathAgent learns more robust and generalizable visual representations.
>
> -----
>
> > ### N3b: How does CPathAgent perform on BRACS without navigation?
>
> Thanks for the suggestion. We tested CPathAgent on BRACS without sequential navigation:
>
> | Model                        | BRACS|
> | ---------------------------- | -------------- |
> | CPath-Omni                   | 57.9%          |
> | CPathAgent (w/o navigation)  | 59.6%          |
> | CPathAgent (with navigation) | 64.3%          |
>
> Performance drops significantly without navigation (4.7% decrease), though still exceeds CPath-Omni, likely due to exposure to various sizes of fields of view during training.
>
> -----
>
>
> > ### N3c: Can CPathAgent be adapted for survival prediction or biomarker detection?
>
> While CPathAgent currently focuses on diagnostic classification, extending to prognostic or biomarker detection tasks is theoretically feasible and represents an exciting future direction. For example,  for prognostic applications, we could incorporate training data linking morphological observations to outcomes (e.g., "Based on these observed morphological features, predict 5-year survival → output: 0.78").
>
> We acknowledge this requires substantial validation. Creating an explainable, LMM-based prognostic/biomarker detection system would be groundbreaking for precision oncology, but rigorous clinical studies are needed to establish reliability and clinical utility.
>
> -----
>
> We hope these additional responses fully address your questions. We sincerely appreciate your thorough evaluation and thoughtful follow-up questions that have significantly strengthened our work.

---

> > ### Comment · Reviewer_EKpp · 2025-08-07
> > **Second response**
> >
> > Thank you for responding to every followup question. I believe that working all of the results, clarifications, and discussions into the manuscript will improve the quality and clarity of the evaluation and strengthen the paper overall.

---

> > > ### Author Response · Authors · 2025-08-08
> > > **Appreciate and confirmation**
> > >
> > > Thank you for your constructive suggestions throughout the review process. We will carefully revise the manuscript to incorporate all the additional results, clarifications, and discussions.
> > >
> > > We truly appreciate your valuable guidance during this process. May we kindly confirm whether our responses have fully addressed your concerns?

---

> > > > ### Comment · Reviewer_EKpp · 2025-08-08
> > > > **Confirmation**
> > > >
> > > > My concerns have been addressed to the extent that can be reasonably expected in a short period of time. While I still wonder whether the performance of CPathAgent on PathMMU-HR2 is unrelated to the method in which this dataset was created, the BRACS results you provided help to address that concern. Some of my additional questions, like the handling of multiple slides per case or the impact on clinical practice of using a scan-and-zoom agentic approach rather than a multi-scale integration may be explored in future work.

---

> > > > > ### Author Response · Authors · 2025-08-09
> > > > > **Thank you**
> > > > >
> > > > > Thank you again for your valuable feedback on our work, and we're glad to hear that your concerns have been addressed!

---

### Note · Authors · 2025-08-16

We sincerely thank all reviewers for their constructive feedback throughout this discussion.

We are particularly encouraged by the strong consensus among reviewers regarding the significance of CPathAgent, which introduces **a novel agent-based framework that mimics pathologists' diagnostic workflow through dynamic navigation planning and multi-scale reasoning**: Reviewer EKpp recognized this as "an interesting approach by mimicking pathologists' navigation" with interpretable diagnostic paths; Reviewer m8m7 acknowledged our "innovative approach in mimicking pathologists' multi-scale diagnostic reasoning"; Reviewer cWNz expressed strong interest in our multi-scale reasoning design and PathMMU-HR² benchmark; and Reviewer e2cS highlighted our success in "addressing the gap between AI systems and actual pathologist workflows."



Following reviewers' suggestions, we addressed all concerns through extensive additional experiments and clarifications that further strengthen our claims:

1. **Strong Generalization Capability**: Zero-shot evaluation on held-out CPTAC and BRACS datasets confirms robust generalization despite training on significantly fewer WSIs than existing models
2. **Superior Training Efficiency**: Achieved SOTA or comparable performance with only 5,254 training WSIs (100× fewer than comparable models)
3. **Additional Ablations**: Detailed component analysis quantifies the individual contributions of navigation planning and multi-scale reasoning modules
4. **Clinical Validation**: Pathologist evaluation confirms that  generated navigation paths meet clinical diagnostic standards
5. **Additional Benchmarking**: Additional comparisons with CPath-Omni on TCGA classification tasks and evaluation on both PathMMU-HR² and PathMMU-Pro benchmarks as requested

We commit to incorporating all reviewer suggestions in the final version:

- Restructuring the paper to prioritize agent architecture and technical implementation details
- Including all additional experimental results and ablation studies from the rebuttal
- Provide open-source release of datasets and models

We hope CPathAgent provides a useful exploration of agent-based methods in computational pathology and inspires further research into clinically-aligned AI systems that prioritize both performance and interpretability. Again, we sincerely thank all reviewers for their engagement, which helped improve this work to better benefit the community.

----
Sincerely,

Authors of Submission 7678

---

### Decision · Program_Chairs · 2025-09-17

**Decision:**

Accept (poster)

**Comment:**

This paper proposes CPathAgent, an agentic pipeline that plans where to zoom and explains its reasoning, similar to how a pathologist reviews slides. They also created a new huge-region benchmark. The paper reports gains over recent LMMs and competitive WSI results, with pathologist-readable traces.

Strengths: Their clinically aligned agent workflow with transparent navigation/reasoning is novel and interesting. The huge-region benchmark could be very useful for others to build upon, which the authors say they will release in the NeurIPS checklist (hopefully without guardrails and made easily downloadable).

Weaknesses: Training includes TCGA while evaluation also uses TCGA, so several results are in-distribution; this must be disclosed and quantified (patient-level non-overlap, cohort counts), with stronger out-of-distribution emphasis. Statistical rigor is thin. Human comparisons may be structurally biased because AI can exhaustively scan every high-mag tile, unlike time-limited pathologists. In other words, AI can compensate for the inherent weaknesses of the human visual system, which is unable to analyze every region at the highest possible resolution (it just would be too tedious and time consuming). I'm concerned that this method may be prone to missing smaller tumors, especially for out-of-distribution datasets. See Table 4 of this paper, for an example analysis looking at performance across different region sizes: https://arxiv.org/pdf/2412.02012v3

Discussion and Rebuttal: Reviewers converged to borderline-accept after authors added some out-of-distribution results. However, concerns about TCGA train/eval coupling, missing statistics, and clarity of the navigation policy persist. I weigh the new evidence positively but still have some concerns as described in weaknesses.

Decision: The novel agentic approach and a useful dataset justify a poster. Camera-ready must: (1) clearly indicate in the paper the issues with the train/eval set, and (2) move key out-of-distribution results and results provided in the discussion period into the main paper.